# Sea surface height anomaly and geostrophic current velocity from altimetry measurements over the Arctic Ocean (2011–2020)

**Francesca Doglioni**[1]**, Robert Ricker**[1,2]**, Benjamin Rabe**[1]**, Alexander Barth**[3]**, Charles Troupin**[3]**, and Torsten Kanzow**[1,4]

[1]Alfred-Wegener-Institut Helmholtz-Zentrum für Polar- und Meeresforschung, Bremerhaven, Germany
[2]NORCE Norwegian Research Centre, Tromsø, Norway
[3]GeoHydrodynamics and Environment Research (GHER), University of Liège, Liège, Belgium
[4]Department 1 of Physics and Electrical Engineering, University of Bremen, Bremen, Germany

**Correspondence:** Francesca Doglioni (francesca.doglioni@awi.de)

**Abstract.** TS1 CE1 Satellite altimetry missions flying over the ice-covered Arctic Ocean have opened the possibility of further understanding changes in the ocean beneath the sea ice. This requires complex processing of satellite signals emerging from the sea surface in leads within the sea ice, with efforts to generate consistent Arctic-wide datasets of sea surface height ongoing. The aim of this paper is to provide and assess a novel gridded dataset of sea surface height anomaly and geostrophic velocity, which incorporates both the ice-covered and open ocean areas of the Arctic. Data from the CryoSat-2 mission in the period 2011–2020 were gridded at monthly intervals, up to 88° N, using the Data-Interpolating Variational Analysis (DIVA) method. To examine the robustness of our results, we compare our dataset to independent satellite data, mooring time series and Arctic-wide hydrographic observations. We find that our dataset is well correlated with independent satellite data at monthly timescales. Comparisons to in situ ocean observations show that our dataset provides reliable information on the variability of sea surface height and surface geostrophic currents over geographically diverse regions of the Arctic Ocean and different dynamical regimes and sea ice states. At all comparison sites we find agreement with in situ observed variability at seasonal to interannual timescales. Furthermore, we find that our geostrophic velocity fields can resolve the variability of boundary currents wider than about 50 km, a result relevant for studies of Arctic Ocean circulation. Additionally, large-scale seasonal features emerge. Sea surface height exhibits a wintertime Arctic-wide maximum, with the highest amplitude over the shelves. Also, we find a basin-wide seasonal acceleration of Arctic slope currents in winter. We suggest that this dataset can be used to study not only the large-scale sea surface height and circulation, but also the regionally confined boundary currents. The dataset is available in netCDF format from PANGAEA at https://doi.org/10.1594/PANGAEA.931869 (Doglioni et al., 2021d).

## 1 Introduction

Regionally enhanced atmospheric warming in the Arctic over the past century has been driving rapid changes at the sea surface. The reduction in the concentration and age of sea ice resulted in modified vertical momentum fluxes, which intensified ice and water drift, in turn enhancing sea ice drift and export. Evidence of basin-wide positive trends in sea ice drift, particularly strong in the summer season, has largely been found in satellite observations (Hakkinen et al., 2008; Spreen et al., 2011; Kwok et al., 2013; Kaur et al., 2018). In contrast to studies on ice drift, observational studies of ocean currents, including analysis of regional in situ data (e.g., McPhee, 2012), indirect calculation from wind and ice drift observation (Ma et al., 2017) or, only recently, satellite altimetry data (Armitage et al., 2017; Morison et al., 2021), give a more fragmentary picture of changes and intensification of surface ocean currents. The reason for this is that, in ice-covered regions, long-term observation of near-surface currents, either from in situ or satellite sensors, has been hindered until recent times by the very presence of ice.

Before the advent of satellite observations, the large-scale Arctic Ocean surface circulation (see a schematic in Fig. 1) was partially reconstructed from in situ observations and models, albeit with limitations in terms of spatial extent or processes represented. On the one hand, in situ observations of surface ocean currents are sparse due to the remoteness of the Arctic environment and the high risk of losing sensors in ice-covered areas (Haller et al., 2014). On the other hand, while numerical models allow for the study of basin-wide processes, they rely largely on theoretical formulation of physical processes, often constrained by insufficient in situ observations (Proshutinsky and Johnson, 1997; Jahn et al., 2010). Satellite-derived data then provided novel alternatives to tackle these issues. By accessing remote regions of the Arctic Ocean, satellite data proved to be a key component in constraining and assessing models, as pointed out by recent ocean reanalysis efforts by Nguyen et al. (2021), and can be used to infer ocean circulation below the ice. For instance, based on assumptions about the ice response to wind forcing (i.e., free drift), Kwok et al. (2013) used satellite sea ice drift observations to deduce near-surface ocean circulation. Beyond ice drift observations, satellite altimetry can provide a more direct way to observe near-surface ocean currents (Armitage et al., 2017). This is because altimetry-derived sea surface height can be used to compute the geostrophic velocity, a component of the ocean surface velocity that is dominant in the Arctic on spatial scales larger than 10 km (Nurser and Bacon, 2014) and timescales longer than a few days.

The first satellite altimetry missions over the Arctic Ocean, launched in the 1990s and at the turn of the 21st century, covered it only partially up to 82° N (e.g., ERS 1 and 2, Envisat) or flew over ice regions for limited periods of time (ICEsat-1). CryoSat-2 is currently the mission providing the most complete coverage and the longest life span, with observations up to 88° N since 2010 (Wingham et al., 2006). In the years to come, recently launched missions, such as Sentinel-3 and ICEsat-2, will provide an increasing amount of data from the Arctic Ocean. Despite the availability of data, methodologies for the processing of the signal coming from the ocean in ice-covered regions have taken much longer to develop. The observations were originally aimed at the study of the cryosphere (Laxon, 1994; Alexandrov et al., 2010; Ricker et al., 2014; Armitage and Davidson, 2014), with efforts towards the generation of altimetric datasets for oceanographic purposes being made later (e.g., Bouffard et al., 2017). For this reason, many available oceanographic datasets are limited either to the open ocean (Volkov and Pujol, 2012; Müller et al., 2019) or to the ice-covered ocean (Kwok and Morison, 2011, 2016; Mizobata et al., 2016).

Only in recent years have a few basin-wide, multi-annual, gridded datasets of sea surface height been generated at monthly timescales (Armitage et al., 2016; Rose et al., 2019; Prandi et al., 2021). These datasets play an important role in improving our understanding of the Arctic system as a whole and of its present and future change (Timmermans and Marshall, 2020). However, differences between independent gridded datasets are introduced by the altimeter signal processing (Ricker et al., 2014; Armitage and Davidson, 2014; Passaro et al., 2014), measurement corrections (Carrère et al., 2016; Ricker et al., 2016; Birol et al., 2017) and interpolation of observations onto regular grids. However, it is not well known how these products compare to each other or to what extent their spatial and temporal resolution is robust in ice-covered regions (e.g., signal-to-noise ratio). Sea surface height maps have been assessed mostly against tide gauge data at the periphery of the Arctic Ocean or in ice-covered regions against data from hydrographic profiles, which makes it difficult to evaluate the robustness of monthly estimates (Morison et al., 2012; Mizobata et al., 2016; Armitage et al., 2016; Morison et al., 2018; Rose et al., 2019; Morison et al., 2021; Prandi et al., 2021). Furthermore, so far only one study by Armitage et al. (2017) has provided and evaluated monthly maps of geostrophic velocities.

In this study we provide and assess a new Arctic-wide gridded dataset of sea surface height and geostrophic velocity, covering up to latitude 88° N at monthly resolution over the period 2011 to 2020. This dataset was obtained from CryoSat-2 observations covering both the ice-covered and ice-free Arctic Ocean. Our specific objectives are

– to document the methods used to produce the monthly fields of sea surface height and geostrophic velocity,

– to compare monthly sea surface height fields to an independent altimetry dataset, thereby suggesting methodological steps likely to introduce noise or biases in altimetry-gridded products at monthly resolution, and

– to assess this dataset through comparisons with in situ data, including multiyear mooring-based sea surface

height and current time series, from several regions of the Arctic Ocean with diverse geography and dynamical regimes.

This paper is structured as follows. In Sect. 2 we describe how altimetry-derived variables are commonly calculated, thereby defining the notation used in this work. In Sect. 3 we provide a description (e.g., sources, spatial and temporal coverage) of the altimetry data used to derive our monthly dataset: the independent altimetry and in situ datasets used for evaluation. In the Methods section (Sect. 4) we first describe the in situ data processing (Sect. 4.1) and then the derivation of monthly gridded sea surface height and geostrophic velocity from altimetry observations (Sect. 4.2, 4.3, 4.4). In Sect. 5 we present the monthly fields and their evaluation against independent altimetry measurements and in situ data. Comparing against in situ data, we identify the temporal and spatial scales on which they have the highest agreement. In the same section we also describe the seasonal cycle emerging from the final monthly maps. Lastly, in Sect. 6 we discuss the spatial and temporal resolution of our dataset and put the emerging features of the seasonal cycle in context with findings from other studies.

## 2 Ocean altimetry background

In oceanography, studying sea level variability is relevant to understanding underlying processes linked to steric and mass variations in the water column. These variations can be measured separately by means of in situ hydrographic profiles (steric) and ocean bottom pressure records (mass), though with limitations in terms of spatial and temporal coverage. An integrated measure of the spatial and temporal variability of these two components, known as dynamic ocean topography ($\eta$), can be derived over the global ocean from measurements of sea surface height ($h$), as obtained from satellite altimetry. In the following, we summarize how $\eta$ can be derived from altimetry measurements and introduce some notation relevant to satellite altimetry.

$h$ is the ocean height over a reference ellipsoid (e.g., WGS84, TOPEX/Poseidon) and is calculated by subtracting the measurement of the satellite range to the sea surface ($R$) from the satellite altitude $H$ over the ellipsoid:

$$h = H - (R + C), \tag{1}$$

where $C$ are corrections to the $R$ measurement. $\eta$ is then derived from $h$ by removing the geoid height ($G$), i.e., the static ocean height component given the Earth's gravitational field, as follows:

$$\eta(t) = h(t) - G. \tag{2}$$

The time-varying component of $\eta$, the sea surface height anomaly $\eta'$, is given by $h$ referenced to a long-term mean sea surface height $\langle h \rangle$:

$$\eta'(t) = h'(t) = h(t) - \langle h \rangle. \tag{3}$$

In order to compute the absolute geostrophic velocity, $\eta$ is reconstructed by adding the mean dynamic topography $\langle \eta \rangle$, the temporal mean of $\eta$. This is derived from $\langle h \rangle$ by removing $G$, as estimated via a geoid model (e.g., Rio et al., 2011; Farrell et al., 2012; Knudsen et al., 2019; Mulet et al., 2021).

$\eta$ is used to derive geostrophic velocities at the sea surface. Geostrophic velocities result from the balance of the pressure gradient force and the Coriolis force, valid in the Arctic on spatial scales larger than a few kilometers (Nurser and Bacon, 2014) and timescales longer than a few days. The two components can be expressed as

$$\begin{cases} u_g = -\frac{g}{f\,R_E} \frac{\partial \eta}{\partial \theta}, \\ v_g = \frac{g}{f\,R_E \cos(\theta)} \frac{\partial \eta}{\partial \phi}, \end{cases} \tag{4}$$

where $\theta$ and $\phi$ are latitude and longitude converted to radian angles, $R_E$ is the Earth's radius, $g$ is the gravitational acceleration and $f = 2\Omega \sin(\theta)$ is the Coriolis parameter.

The nomenclature introduced in this section will be used below to describe the datasets used and the ones resulting from the present analysis.

## 3 Data

### 3.1 CryoSat-2 sea surface height in ice-covered and ice-free regions

The monthly gridded dataset generated in this study is based on two sets of $\eta'$ observations along the satellite ground track (projection of its orbit at the ground), one over ice-covered and a second over ice-free areas. Observations from the European Space Agency (ESA) CryoSat-2 mission (ESA level L2, Bouzinac, 2012) were selected between 60 and 88° N over the period 2011–2020. For ice-covered areas, down to ice concentration 15 %, we use the Alfred Wegener Institute (AWI) dataset (data version 2.4, Hendricks et al., 2021), available at ftp://ftp.awi.de/sea_ice/projects/cryoawi_ssh (last access: TS4). The AWI dataset does not provide estimates below 15 % ice concentration, since the retrieval algorithm is optimized for ice-covered areas, while uncertainties increase in areas with low ice concentration (Ricker et al., 2014). The dataset includes year-round data (including summer), with an along-track resolution of approximately 300 m. In this dataset, radar echoes from the surface (waveforms) are classified into sea ice and open water. Then, sea surface elevations from openings in the sea ice cover (i.e., leads) are retrieved using the retracking algorithm described by Ricker et al. (2014). The processing includes waveforms in the synthetic aperture radar (SAR) and the interferometric SAR (SARIn) modes (ESA level-L1b dataset; see the areas covered by each altimeter mode at http://cryosat.mssl.ucl.ac.uk/qa/mode.php, last access: TS5). Over the open ocean, up to ice concentration 15 %, we use data archived in the Radar Altimetry Database System, with an along-track resolution of 7 km (RADS, Scharroo et al., 2013; Scharroo,

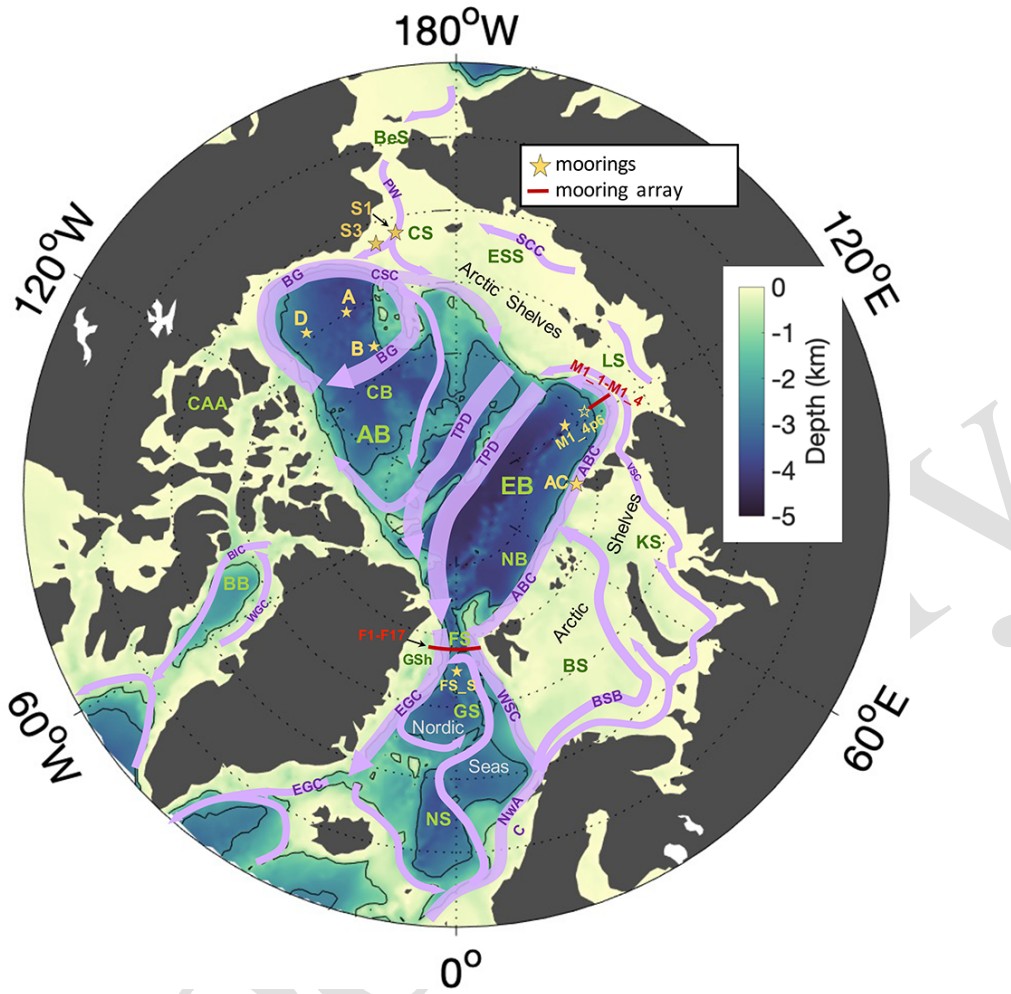

**Figure 1.** Arctic Ocean map and bathymetry (IBCAO, Jakobsson et al., 2012) with the main sub-regions (green acronyms) and the mean surface circulation pathways (purple arrows and abbreviations). Locations of moorings used for validation are indicated with yellow stars and red dotted lines: at the Laptev Sea continental slope, the empty star indicates where the bottom pressure data are taken. Depth contours are drawn at 1000 and 2500 m depths. Regions: Nordic Seas: Greenland Sea (GS), Norwegian Sea (NS); Arctic shelves: Barents Sea (BS), Kara Sea (KS), Laptev Sea (LS), East Siberian Sea (ESS), Chukchi Sea (CS), Greenland Shelf (GSh); Arctic deep basins: Amerasian Basin (AB), Canada Basin (CB), Eurasian Basin (EB), Nansen Basin (NB); Baffin Bay (BB); Canadian Arctic Archipelago (CAA); Fram Strait (FS); Bering Strait (BeS). Currents: West Spitsbergen Current (WSC); Norwegian Atlantic Current (NwAC); Barents Sea Branch (BSB); Vilkitsky Strait Current (VSC); Arctic Boundary Current (ABC); Siberian Coastal Current (SCC); Pacific Water inflow (PW); Chukchi Slope Current (CSC); Beaufort Gyre (BG); TransPolar Drift (TPD); East Greenland Current (EGC); West Greenland Current (WGC); Baffin Island Current (BIC). TS3

2018), available at http://rads.tudelft.nl/rads/rads.shtml (last access: TS6 ). The merged along-track dataset, as processed in this work (see Sect. 4.2), is available in Doglioni et al. (2021d).

5    All $\eta'$ observations are referenced to the global DTU15MSS mean sea surface (Technical University of Denmark, updated from the DTU13MSS described in Andersen et al., 2015), which uses multi-mission altimeter data including the satellites Envisat, ICEsat and CryoSat-2. To recon-

10 struct $\eta$ (Sect. 4.4), we added our final gridded $\eta'$ to the mean dynamic topography DTU17MDT (Knudsen et al., 2019),

which is the DTU15MSS minus the OGMOC geoid model, both referenced to the T/P ellipsoid (P. TS7 Knudsen, personal communication, 8 September 2022).

## 3.2   Datasets used for comparisons

We use independent satellite and in situ datasets to evaluate the final monthly fields of altimetry-derived $\eta'$ and $(u_g, v_g)$. These datasets are described below, and the locations of the moorings are indicated in Fig. 1.

### 3.2.1 Sea surface height

Monthly $\eta'$ fields were compared to an independent satellite gridded dataset over the entire Arctic. This dataset is described by Armitage et al. (2016) and will be hereafter referred to as the CPOM DOT (Centre for Polar Observation and Modelling Dynamic Ocean Topography, available at http://www.cpom.ucl.ac.uk/dynamic_topography, last access: TS8 ). The CPOM DOT is a regional Arctic dataset spanning the years 2003–2014, derived from sea surface height observations (relying on the satellite missions Envisat and CryoSat-2) and a geoid model (GOCO03s). Monthly fields are provided on a $0.75° \times 0.25°$ longitude–latitude grid, up to a latitude of 82° N. The CPOM DOT was compared to the interpolated $\eta'$ fields at grid points south of 82° N for the overlap period between January 2011 and December 2014. Both datasets were referred to their own temporal average over this period.

We further used several sources of in situ steric height (the height component due to changes in density) plus ocean bottom pressure equivalent height (related to changes in water mass) as ground truth to (i) correct instrumental biases in the along-track $\eta'$ and (ii) evaluate the spatial and temporal variability of the $\eta'$ fields.

In a first step we used steric height from hydrographic profiles collected in the Arctic Deep Basins, plus ocean bottom pressure from the Gravity Recovery and Climate Experiment satellite (GRACE), to correct an instrumental offset existing between the along-track AWI and RADS $\eta'$ observations (Sect. 4.2.1). The hydrographic profiles cover the period 2011–2014 and include data from various platforms, among them ships and autonomous drifting buoys (observations listed in Rabe et al., 2014, extended to 2014 using the sources listed in Solomon et al., 2021, in their Table 2). Steric height was computed following Eq. (7). Ocean bottom pressure is included in the GRACE release 6 data as provided by the Jet Propulsion Laboratory (data are available online at https://podaac.jpl.nasa.gov/dataset/TELLUS_GRAC_L3_JPL_RL06_LND_v03, last access: TS9 ).

Then, we assessed whether the offset applied as a correction to the AWI and RADS datasets did not bias the natural sea surface slope induced by geostrophic currents. We evaluated the correction in the Fram Strait, where the Eastern Greenland Current flows in a region of transition from ice-covered to ice-free areas. To this end, we compared zonal cross sections of the strait from our final $\eta$ fields to in situ steric height, based on hydrographic sections in the Fram Strait plus GRACE data (Sect. 5.2.1). The hydrographic sections were taken at 78° 50′ N from ship-based conductivity–temperature–depth (CTD) between late June and early July in 2011 and 2012 (expeditions ARK-XXVI/1 and ARK-XXVII/1 aboard RV *Polarstern*; von Appen et al., 2015). As for the hydrographic profiles, steric height was computed following Eq. (7) (Sect. 4.2.1).

Finally, we evaluated the temporal variability of the $\eta'$ fields by comparing them locally to CTD and McLane moored profiler (MMP) data from five seafloor moorings across the central Arctic (Table 1). The processing of temperature, salinity and ocean bottom pressure data from moorings is described in Sect. 4.1. Both mooring data and altimetry data from each location were referred to the temporal average over the time span covered by the mooring data. The moorings were located in the southern Fram Strait (FS_S), at the shelf break north of the Arctic Cape, the headland of Severnaya Zemlya (AC), down the continental slope north of the Laptev Sea (M1_4 and M1_6) and in the Beaufort Sea (A and D). FS_S was part of a meridional mooring array deployed by the AWI in the Fram Strait between 2016 and 2018. Data from the FS_S mooring are available in von Appen et al. (2019). The AC was one of seven moorings deployed between 2015 and 2018 within the context of the German–Russian project Changing Arctic Transpolar System (CATS). Moorings M1_4 and M1_6 were part of a six-mooring array deployed in the Laptev Sea continental slope between 2013 and 2015 within the Nansen and Amundsen Basins Observations System II project (NABOS-II). Steric height and bottom pressure equivalent height were calculated from moorings M1_6 and M1_4, respectively, given that not all measurements were available from a single mooring. Hereafter, the combination of data from the two moorings is indicated as M1_4p6. Data from the M1_4p6 mooring are available from the Arctic Data Center (Polyakov, 2016, 2019; Polyakov and Rembert, 2019). Data at moorings A and D cover the period 2011–2018 and were collected and made available by the Beaufort Gyre Exploration Program (BGEP) based at the Woods Hole Oceanographic Institution, in collaboration with researchers from Fisheries and Oceans Canada at the Institute of Ocean Sciences (https://www2.whoi.edu/site/beaufortgyre/, last access: TS10 ). Furthermore, we compared our $\eta$ monthly fields to monthly averages of the hydrographic profiles from the Arctic Deep Basins described above.

### 3.2.2 Velocity

We used measurements of near-surface velocity from a total of 19 moorings to evaluate monthly geostrophic velocity in four different regions within the Arctic. The validation points include eastern and western Arctic circulation regimes, the central Arctic Ocean, the Arctic shelf seas and the main exchange gateways of the Arctic. Data from two mooring lines in the Fram Strait and down the continental slope of the Laptev Sea were used to assess how well our final geostrophic fields resolve strong and narrow slope currents. Data from three moorings in the Beaufort Sea were used to evaluate our geostrophic fields in an open ocean region, characterized by weak and broad currents. Data from the Chukchi Sea served to evaluate how our dataset performs in a shallow shelf sea.

**Table 1.** Names, locations, monthly data availability and temperature/salinity sensor depth for the seafloor moorings used as a comparison dataset to validate altimetry-derived $\eta'$ (refer to Fig. 9).

| Name | Longitude | Latitude | No. of months (years) | T/S sensor depth (m) |
|------|-----------|----------|-----------------------|----------------------|
| FS_S | 0° E | 78°10′ N | 23 (2016–2018) | 49/231/729 |
| AC | 94°51′ E | 82°13′ N | 34 (2013–2018) | 50/131/196/293/593/1448 |
| M1_4p6 | 125°42′ E | 78°28′–81°9′ N | 24 (2013–2015) | 26/42/53, MMP profiler 70-760 |
| A | 150°1′ E | 75°0′ N | 57 (2011–2017) | MMP profiler 50-2001 |
| D | 139°59′ E | 74°0′ N | 88 (2011–2018) | MMP profiler 50-2001 |

**Table 2.** Names, locations, monthly data availability and averaging depth ranges for the seafloor moorings used as a comparison dataset to validate altimetry-derived geostrophic velocity; moorings are located across the Fram Strait (first 17 rows), across the Laptev Sea continental slope (following 4 rows), in the Beaufort Sea (following 3 rows) and in the eastern Chukchi Sea (last 2 rows). Variable locations indicate the relocation of the moorings in some years; in the third column, values in parentheses indicate the years of data availability. Data from mooring records longer than 24 months (in bold) were used to compute the correlation with altimetry.

| Name | Longitude | Latitude | No. of months (years) | Depth range (m) |
|------|-----------|----------|-----------------------|-----------------|
| Fram Strait | | | | |
| F1 | 8°40′ E | 78°50′ N | 7 (2015) | 75 |
| **F2** | 8°20′ E | 78°49′–79°00′ N | **42** (2011–2012, 2015–2018) | 75 |
| **F3** | 8°00′ E | 78°50′–79°00′ N | **73** (2011–2018) | 75 |
| **F4** | 7°01′ E | 78°50′–79°00′ N | **71** (2011–2018) | 75 |
| **F5** | 5°40′–6°01′ E | 78°50′–79°00′ N | **73** (2011–2018) | 75 |
| **F6** | 4°20′–5°00′ E | 78°50′–79°00′ N | **34** (2015–2018) | 75 |
| **F7** | 4°00′–4°05′ E | 78°50′ N | **38** (2012–2015) | 75 |
| **F8** | 2°45′–2°48′ E | 78°50′ N | **25** (2012–2014) | 75 |
| **F15** | 1°35′–1°36′ E | 78°50′ N | **42** (2011–2014) | 75 |
| **F16** | 0°00′–0°26′ E | 78°50′ N | **70** (2011–2014, 2016–2018) | 75 |
| F9 | 0°49′ W | 78°50′ N | 21 (2011–2012, 2014) | 75 |
| **F10** | 2°03′–1°59′ W | 78°50′ N | **68** (2011–2016) | 75 |
| F11 | 3°04′ W | 78°48′ N | 9 (2011–2012) | 75 |
| F12 | 4°01′–3°59′ W | 78°48′ N | 13 (2011–2012) | 75 |
| F13 | 5°00′ W | 78°50′ N | 20 (2011–2012) | 75 |
| F14 | 6°30′ W | 78°49′ N | 12 (2011–2012) | 75 |
| F17 | 8°7′ W | 78°50′ N | 13 (2011–2012) | 75 |
| Laptev Sea | | | | |
| **M1_1** | 125°48′–125°50′ E | 77°04′ N | **62** (2013–2018) | 20–50 |
| **M1_2** | 125°48′ E | 77°10′ N | **60** (2013–2018) | 20–50 |
| **M1_3** | 125°48′ E | 77°39′ N | **61** (2013–2018) | 20–50 |
| **M1_4** | 125°54′–125°58′ E | 78°28′ N | **61** (2013–2018) | 20–50 |
| Beaufort Sea | | | | |
| **A** | 150°1′ W | 75°0′ N | **82** (2011–2012, 2013–2018) | 20–40 |
| **B** | 150°2′ W | 77°59′ N | **83** (2011–2016, 2018) | 20–40 |
| **D** | 139°59′ W | 74°0′ N | **74** (2011–2014, 2015–2018) | 20–40 |
| Chukchi Sea | | | | |
| **S1** | −167°15′ E | 71°10′ N | **37** (2011–2014) | 35 |
| **S3** | −164°43′ E | 71°14′ N | **37** (2011–2014) | 35 |

In the Fram Strait, we employed 10 out of 17 moorings from the array located along a zonal section at $78°50'$ N, between the longitudes $9°$ W and $8°$ E, maintained since 1997 by the AWI (moorings F1–F10 and F15/F16; Beszczynska-Möller et al., 2012) and the Norwegian Polar Institute (NPI, moorings F11–F14 and F17; de Steur et al., 2009). Velocity measurements were acquired by acoustic Doppler current profilers (ADCPs) and current meters (CMs). We performed the comparison using the time series recorded by the shallower CM (75 m) and by the ADCP bin nominally closest to the CM sensor depth. The mooring data are available through PANGAEA (von Appen et al., 2019; von Appen, 2019). For the Laptev Sea, data were used from four moorings deployed in a meridional transect along the $126°$ E meridian within the context of the NABOS-II project (moorings M1_1 to M1_4). All four moorings provide records spanning 5 years between 2013 and 2018 (data are available from the Arctic Data Center in Polyakov, 2016, 2019; Polyakov and Rembert, 2019). In the Beaufort Sea, ADCP data from BGEP moorings A, B and D were used, covering the period 2011–2018 (available at https://www2.whoi.edu/site/beaufortgyre/, last access: TS11 ). In the Chukchi Sea we used ADCP data from the two moorings S1 and S3 over the period 2011–2014, processed by ASL Environmental Sciences and available from the NOAA National Centers for Environmental Information (Mudge et al., 2017).

At the two mooring arrays, we compared the $(u_g, v_g)$ component normal to the mooring line, linearly interpolated to the mooring locations ($v_n$), to monthly averages of the in situ measured velocities normal to the transects ($v_{ni}$). In the Beaufort Sea and the Chukchi Sea, we compared the speed and bearing of velocity from altimetry and moorings. The comparison was limited to those mooring locations where more than 24 months of in situ data were available at the time of manuscript preparation (Table 2). ADCP velocity measurements from the Laptev Sea continental slope, the Beaufort Sea and the Chukchi Sea were averaged in the depth range 20–50 m in order to capture the geostrophic flow at the surface while excluding the surface Ekman layer (McPhee, 1992; Cole et al., 2014). In the Chukchi Sea currents were processed and archived at three depths, of which only one was within in the 20–50 m range (Mudge et al., 2015); however, it has been shown that currents at this location are mostly barotropic (Fang et al., 2020).

## 4 Methods

In this section we describe the steps followed to derive monthly fields of $\eta'$ and geostrophic velocity $(u_g, v_g)$ from along-track satellite measurements. Furthermore, we provide details on the processing of in situ hydrographic data used for comparison.

### 4.1 Steric height and bottom pressure from mooring data

Time series of the in situ steric height anomaly ($\eta'_S$) and the bottom pressure equivalent height anomaly ($\eta'_P$) were computed from mooring-based measurements of water density and ocean bottom pressure. The relationship between $\eta'$ and the time anomaly of (i) the vertical density profile ($\rho'(z)$) and (ii) the ocean bottom pressure ($P'_b$) is derived by integration of the hydrostatic balance from the sea surface down to the bottom depth, $D$:

$$P'_b = \rho_0 g \eta' + g \int_{-D}^{0} \rho'(z)\mathrm{d}z, \tag{5}$$

where $g$ is the gravitational acceleration and $\rho_0$ is a reference ocean water density, set to $1028\,\mathrm{kg\,m^{-3}}$. Based on this relation, we defined $\eta'_S$ and $\eta'_P$ at the mooring sites FS_S, AC and M1_4p6 as

$$\begin{cases} \eta'_S = -\frac{1}{\rho_0} \int_{-D}^{0} \rho'(z)\mathrm{d}z, \\ \eta'_P = \frac{P'_b}{\rho_0 g}. \end{cases} \tag{6}$$

Vertical density profiles were obtained from temperature and salinity profiles using the Fofonoff and Millard (1983) formula for density. In turn, temperature and salinity profiles were obtained from moored-sensor data by linear interpolation on a regular pressure grid (2 dbar CE2 ) between the shallowest and deepest measurements (see Table 1). Near the surface, data were extrapolated assuming temperature and salinity to be constant and equal to the uppermost measurement. Below the deepest measurement, we assumed the density anomalies to be zero and did not perform extrapolation to the bottom. In the above procedure we made assumptions about the vertical density profile necessary to reconstruct the total steric variability from discrete measurements. First, we applied a conservative approach in the deep part of the water column by neglecting the temporal variability there. While this might have resulted in a slight underestimation of $\eta'_S$, it avoided propagating anomalies for several hundred meters to the bottom, where we do not expect much variability. Furthermore, linear interpolation of temperature and salinity between the discrete measurement levels might have introduced biases into $\eta_S$. Given that we are concerned here with temporal anomalies ($\eta'_S$), we tested how well different interpolation methods reconstructed the variability from a selection of more than 400 continuous CTD profiles from the Fram Strait. We found that linear interpolation was the optimal approach. This method, applied to vertically sub-sampled profiles, was able to reproduce a very large fraction of the total variability in the steric height (on average 88 %), larger than what was obtained with a more complex interpolation scheme like spline.

Ocean bottom pressure records $P'_b$ were de-tided by first performing a tidal analysis of the records using Matlab func-

tion `t_tide` (Pawlowicz et al., 2002) and then removing the resulting tidal time series. Linear trends were removed to account for instrumental drifts. The time series at FS_S exhibited large pressure anomalies, developing on timescales of several months, whose amplitude was at least 1 order of magnitude too large to be explained by changes in ocean currents. Therefore, we high-pass-filtered this time series with a cutoff frequency of 2 months. Despite the fact that this procedure discards part of the low-frequency variability, it has been shown that the coherence between satellite data of sea level and ocean bottom pressure is highest on timescales shorter than about 2 months (Quinn and Ponte, 2012). Furthermore, we note that we have also compared the filtered time series at the FS_S mooring with a filtered bottom pressure record from a mooring located 150 km apart, both at a depth of about 3000 m, which resulted in a high correlation coefficient. No other bottom pressure time series was affected.

## 4.2  Along-track sea surface height anomaly

We generated an Arctic-wide dataset of along-track $\eta'$ by merging the AWI and RADS $\eta'$ datasets. Inconsistencies between the two datasets were reduced by (i) creating a uniform along-track sampling, (ii) reducing biases due to different retracking algorithms and (iii) substituting geophysical corrections where two different corrections were used in the two source products. In this section we first give details about these methods and then present an estimate of the along-track $\eta'$ observational uncertainty.

### 4.2.1  Merging leads and open ocean data

Prior to merging the AWI and RADS datasets, we standardized their along-track sampling rates, which originally were 300 m and 7 km, respectively. With this aim, the AWI dataset was first smoothed by averaging over a 7 km along-track moving window and then linearly interpolated, following time, onto equally spaced locations (7 km) along the satellite tracks. Smoothing the AWI data along the tracks was beneficial to reduce noise, also in view of the computation of geostrophic velocity (see Eq. 10), given that the finite difference operator acts as a high-pass filter (e.g., Liu et al., 2012).

A step-like variation in the $\eta'$ observations at ocean–ice transitions appeared because different models are used to retrack radar signal returns in ice-covered and ice-free regions (Fig. 2a). This is commonly referred to as the "lead-open ocean bias" (Giles et al., 2012). Due to the technical nature of this bias, it is difficult to determine the true bias in the post-processing phase. This is why differences between leads and open ocean are usually corrected in terms of a simple offset (e.g., Giles et al., 2012; Armitage et al., 2016; Morison et al., 2018). To estimate the offset, we compared altimetry to independent in situ hydrography data, similarly to the approach taken by Morison et al. (2018). This approach gives the advantage that circulation features derived from spatial

$\eta$ differences at the transition between AWI and RADS data will be consistent with in situ hydrography.

A good proxy for altimetry-derived $\eta$ is the sum of hydrography-derived steric height ($h_S$) and GRACE-derived ocean bottom pressure ($h_P$, equivalent water thickness). We used hydrographic profiles in the Arctic Deep Basins (Fig. 2b) and compared those to the AWI and RADS along-track $\eta$ (given by $\eta = \eta' + \langle \eta \rangle$, where $\langle \eta \rangle$ is the DTU17MDT described in Knudsen et al., 2019). We computed $h_S$ as the vertical integral of the specific volume anomaly $\delta(p)$ relative to 400 db (Fofonoff and Millard, 1983):

$$h_S = g^{-1} \int_0^{400} \delta(p)\,\mathrm{d}p, \tag{7}$$

where  $\delta(p) = v(S, T, p) - v(35, 0, p)$  and  $v(S, T, p) = 1/\rho(S, T, p)$. The software used is from the seawater library for Matlab (Mathworks), Version 3.1 (Morgan and Pender, 2009). The depth range considered here captures changes in the polar mixed layer (Korhonen et al., 2013), which resides in the top 200 m across the Arctic and includes the main component of steric height variability up to sub-decadal timescales.

$\eta$ and $h_S + h_P$ were compared using all available data in the overlapping period 2011–2014. All $\eta$, $h_S$ and $h_P$ data points were bin-averaged on an equal area grid with a resolution of 25 km. At each bin, average $\eta$ values from the AWI and RADS datasets were compared separately to $h_S + h_P$. In Fig. 2b we show the result of this comparison. Both AWI and RADS data are linearly related to $h_S + h_P$, with a correlation coefficient of 0.98. This gave us confidence that the AWI and RADS datasets differed by a simple offset and that altimetry-derived $\eta$ patterns are consistent with in situ hydrography. We computed two separate offset values, for the AWI and RADS datasets, by taking the average difference between binned $\eta$ and binned $h_S + h_P$ in the ice-covered and ice-free regions, respectively. The two offsets amount to $-12.8$ and $-40.9$ cm. We corrected altimetry data by removing each offset from the respective along-track $\eta'$. After correcting for the two offsets, $\eta$ and $h_S + h_P$ had a root-mean-square deviation (RMSD) of 4–5 cm over a range of 70 cm.

### 4.2.2  Corrections

As a second step, we checked that all corrections applied to the satellite range $R$ (Eq. 1) were consistent between ice-covered and ice-free regions (Table 3 lists the products used here). Standard corrections (European Space Agency, 2016) were applied to both regions to account for (i) the reduction in satellite signal speed caused by the presence of the atmosphere (dry gases, water vapor, ions), (ii) the difference in reflection properties of wave troughs and crests at the sea surface (sea state bias correction, applied solely in the open ocean), and (iii) solid earth tides.

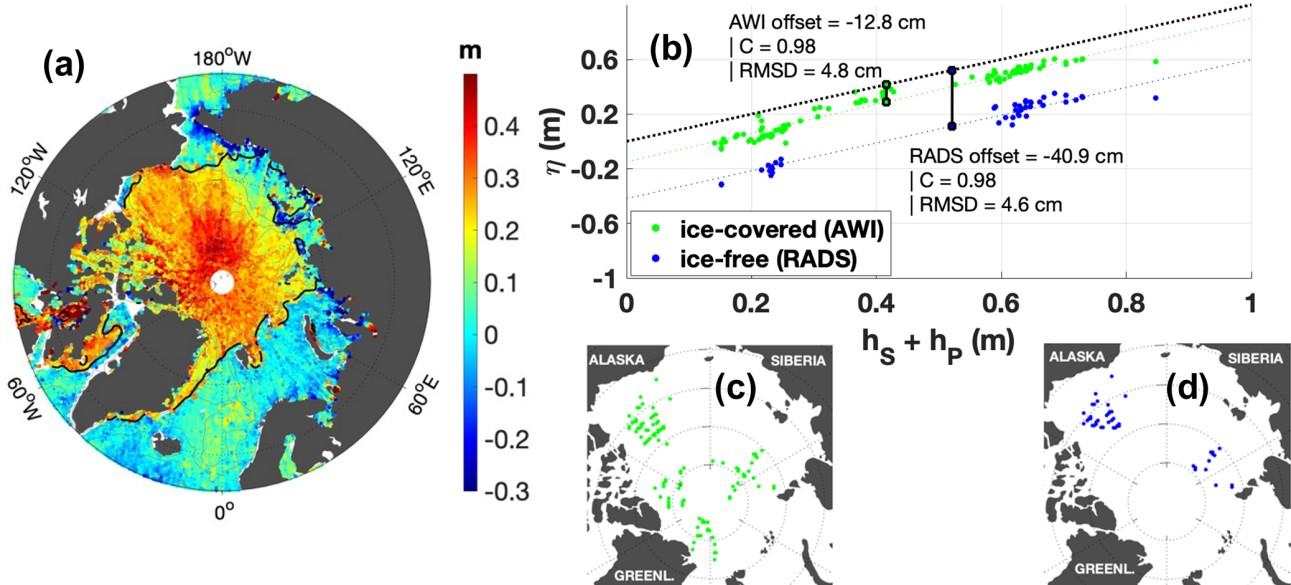

**Figure 2.** Characterization of the respective $\eta'$ bias over leads and open ocean. **(a)** Scatter plot of AWI (ice-covered) and RADS (ice-free) $\eta'$ observations for July 2015 prior to correcting the offset. The black solid line indicates the 15 % sea ice concentration as derived from the OSI SAF ice concentration products (archive OSI-401-b, available at ftp://osisaf.met.no/archive/ice/conc/, last access: TS12 ). **(b)** Steric height plus ocean bottom pressure ($h_S + h_P$) versus $\eta$ for the ice-covered altimetry data (AWI) and ice-free altimetry data (RADS). Vertical bars indicate the offset between the two altimetry datasets and $h_S + h_P$. Panels **(c)** and **(d)** show the grid points where $h_S + h_P$ data points overlap with along-track $\eta$ data points from the AWI (green, panel **c**) and RADS (blue, panel **d**).

**Table 3.** Altimetry corrections applied in this study. Abbreviations: ECMWF (European Centre for Medium-Range Weather Forecasts); CNES (Centre National d'Etudes Spatiales); MOG2D (Modèle d'ondes de gravité 2D); FES2014 (Finite Element Solution 2014); GDR-E (Geophysical Data Record, version E).

| Correction | Source | Reference |
|---|---|---|
| Dry troposphere | Derived from mean surface pressure, based on the ECMWF model | European Space Agency (2016) |
| Wet troposphere | Derived from mean surface pressure, based on the ECMWF model | European Space Agency (2016) |
| Ionosphere | Global Ionospheric Map, provided by CNES | Komjathy and Born (1999) |
| Dynamic atmosphere | Inverted Barometer + MOG2D barotropic model | Carrère et al. (2016) |
| Sea state bias (only open ocean) | Hybrid (mix between parametric and non-parametric techniques) | Scharroo and Lillibridge (2005) |
| Ocean tide | FES2014 | Lyard et al. (2021) |
| Solid earth tide | Cartwright model | Cartwright and Edden (1973) |
| Geocentric polar tide | Instantaneous Polar Location files (sourced from CNES) | Wahr (1985) |
| Orbit | GDR-E | European Space Agency (2016) |

Two further corrections are used to remove the high-frequency ocean variability due to ocean tides and the ocean response to atmospheric pressure and wind forcing. These corrections contribute to reducing the aliasing of sub-monthly temporal changes into spatial variability, which emerges in average fields as meridionally elongated patterns (meridional "trackiness", Stammer et al., 2000). In order to remove the most variability, we tested two products for each correction. First, to correct ocean tides, we used the FES2014 model (Lyard et al., 2021), a more recent version of the FES2004 model (provided by the ESA as standard correc-tion product; Lyard et al., 2006). FES2014 was previously found to perform better than FES2004 in the Arctic (Cancet et al., 2018) and has already been used to correct the most recent satellite altimetry products in this region (e.g., Rose et al., 2019; Prandi et al., 2021). Furthermore, in support of our choice, we found that the noise on the monthly fields, in areas of high tidal amplitude, was reduced by 20 % by using FES2014 with respect to FES2004 (Appendix A).

To correct the effect of atmospheric pressure and wind forcing, we used dynamic atmosphere correction (DAC, Car-rère et al., 2016). DAC is conventionally used today over the

global ocean because it better suppresses the high-frequency variability due to nonlocal forcing (Carrère and Lyard, 2003; Quinn and Ponte, 2012; Carrère et al., 2016). However, for ice-covered regions the ESA still suggests using an inverted barometer (IB) formula, which only accounts for the ocean response to local pressure forcing. This is because to date there has been little knowledge about which of the DAC and IB corrections performs better in ice-covered regions (e.g., Robbins et al., 2016). Studies from the last 2 decades have shown that the deviation of the ocean response from a simple IB response is larger at higher latitudes (e.g., Stammer et al., 2000; Vinogradova et al., 2007; Quinn and Ponte, 2012). In the Arctic, the effect of pressure and wind forcing is not only local, but also travels across the region in the form of mass waves (Fukumori et al., 1998; Peralta-Ferriz et al., 2011; Fukumori et al., 2015; Danielson et al., 2020). This indicates that it would be appropriate to apply DAC to both ice-covered and ice-free regions.

To support our choice of using DAC over IB, we looked at which of them reduced the standard deviation of the along-track $\eta'$ the most with respect to the uncorrected $\eta'$ (see Appendix B). Results showed that DAC outperforms IB in shallow shelf regions (particularly the East Siberian Sea and the Chukchi Sea, in agreement with findings by Piecuch et al., 2022) and that they perform equally well over the deep basins (Fig. B1). For instance, in the East Siberian Sea DAC reduced the uncorrected $\eta'$ standard deviation by 50 % at periods shorter than 20 d, in contrast to no reduction when applying a simple IB (see Table B1). The improvement in DAC with respect to IB over the shelves also appears in the $\eta'$ monthly grids, where meridionally oriented patterns of $\eta'$ are evidently reduced (Fig. B2).

### 4.2.3 Merged along-track dataset and uncertainty estimate

The final merged along-track dataset is composed of two sub-datasets, one for the ice-covered region and one for the ice-free region. The consistency between these two sub-datasets is indicated by their comparable Arctic-wide average standard deviation over the period 2011–2020, amounting to 11.1 and 10.4 cm, respectively.

The average monthly standard deviation and data point density, over the period 2011–2020, are shown in Fig. 3, both for the merged dataset and separately for the AWI and RADS datasets. The two datasets display consistent spatial and temporal variability in the overlap regions, with standard deviation largest in shallow areas throughout the year and enhanced in winter everywhere. The transition between the ice-covered and ice-free regions is generally smooth (Fig. 3a and b), except for increased standard deviation and decreased data density following the marginal ice zone in the Fram Strait. The distribution of data density shows that, both during summer and winter, more than about 50 observations per $100 \, \text{km}^2$ per month are available everywhere, except for the

region north of the Canadian Arctic Archipelago in winter, when the ocean is almost fully covered by pack ice.

Despite the smooth distribution in the average monthly statistics, we note that some residual large-scale sub-monthly variability persists in the data. Figure 3g shows, for instance, a decrease of $\sim 20 \, \text{cm}$ in $\eta'$ north of Greenland between the first and fourth weeks of July 2015. This suggests that, despite correcting high-frequency variability using DAC and a state-of-the-art ocean tidal correction, $\eta'$ is subject to residual large-scale variability on timescales shorter than a month. Constructing monthly maps based on sampling this large-scale, high-frequency variability at different times in different locations will artificially produce short wavelength patterns. A clear example of this pattern is shown in Appendix C, highlighting that residual high-frequency variability can result in representativity error on the monthly fields. We address this issue in the phase of interpolation (Sect. 4.3) and provide in Sect. 4.3.2 an estimate of the contribution of this unresolved variability to the error on the monthly $\eta'$ fields.

On top of the representativity error, several sources contribute to the uncertainty in the single along-track $\eta'$ observations. This uncertainty includes contributions from the altimeter measurement uncertainty, the waveform retracking method, the corrections and the orbit uncertainty. Given the difficulty in assessing the contribution of each of these sources, we provide here a comprehensive estimate of the observational uncertainty based on the absolute difference of the along-track $\eta'$ at satellite track crossovers (Fig. 4). We first defined crossovers as those pairs of $\eta'$ observations within a distance of 7 km. We excluded pairs belonging to the same satellite pass by verifying that they are separated by more than 1 h. We finally evaluated the absolute value of $\eta'$ differences at $\sim 7 \times 10^7$ crossovers, distributed within 100 km of the locations indicated in Fig. 4 (inset panel). In Fig. 4 we see that the crossover difference is small for short time differences and increases as crossovers are separated by a larger time difference. For crossovers very close in time, we expect the difference to approximate the observational uncertainty, while we expect it to increase with time due to additional variability. Therefore, we estimated the observational uncertainty as the average difference at crossovers separated by no more than 3 d, which is 3 cm.

This analysis provides additional information about the $\eta'$ decorrelation timescale. The $\eta'$ crossover difference increases with time above the uncertainty due to local variability. Figure 4 shows that variability increases very rapidly by about 3 cm in the first couple of weeks and then by a further 2 cm after 6 months, and then it decreases again by 2 cm after a full seasonal cycle. This indicates that, on timescales shorter than 1 year, $\eta'$ has a short decorrelation timescale below 1 month (in agreement with Landy et al., 2021) and a long decorrelation timescale of 6 months.

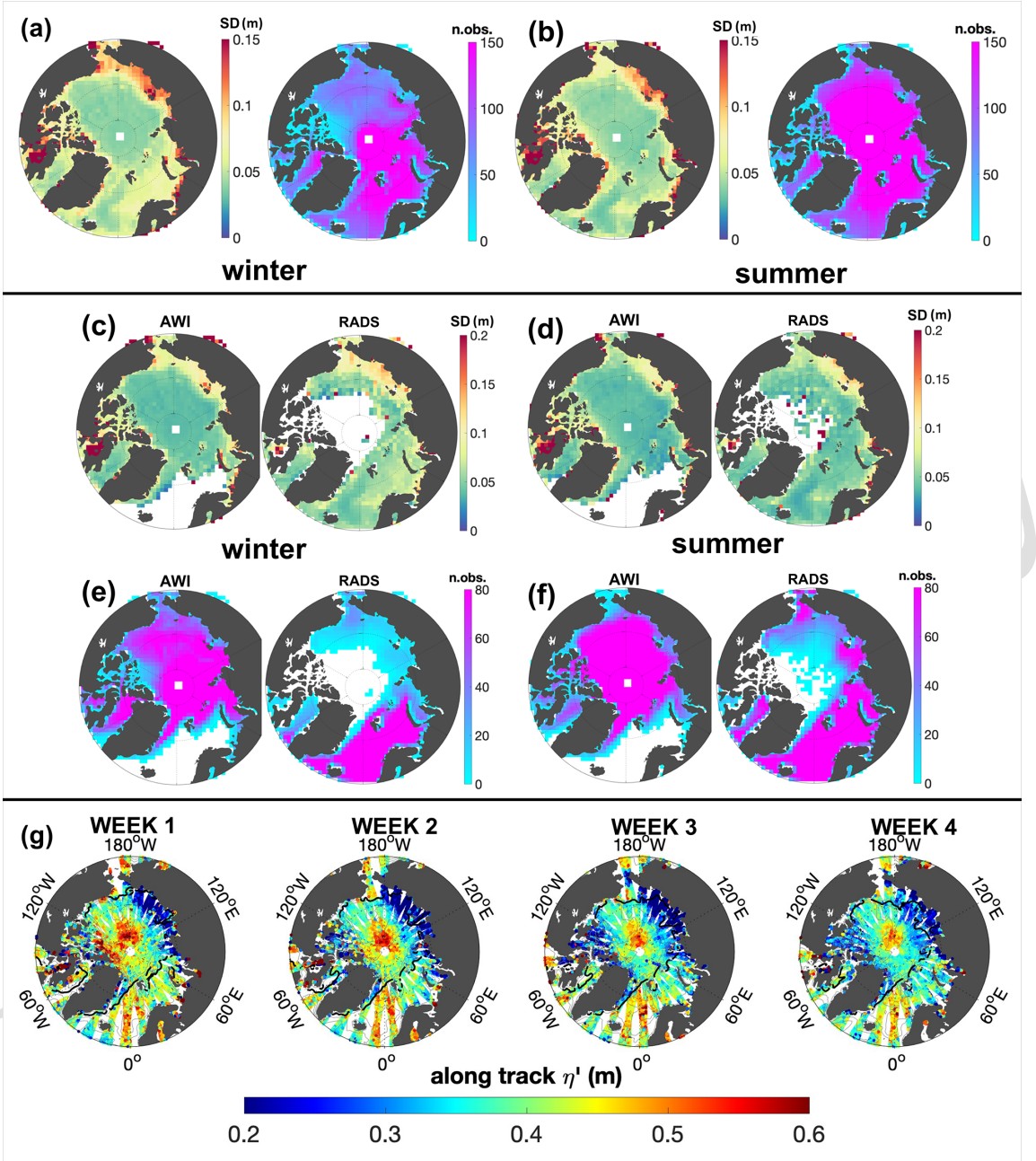

**Figure 3.** Average monthly statistics of the along-track $\eta'$ dataset over the ice-covered and ice-free Arctic Ocean in the period 2011–2020. Standard deviation (panels **a** left, **b** left, **c, d**) and the number of observations per $100\,\mathrm{km}^2$ per month (panels **a** right, **b** right, **e, f**) for the merged dataset **(a, b)** and separately the AWI and RADS datasets **(c–f)** are shown for the winter (October to April) and summer (May to September) seasons. **(g)** Example of weekly along-track data in the month of July 2015; the black solid line indicates the 15 % sea ice concentration as derived from the OSI SAF ice concentration products. Note the different color scales of panels **(a)** and **(b)** with respect to panels **(c)–(f)**.

## 4.3 Gridded sea surface height anomaly

We generated monthly $\eta'$ fields over the period 2011–2020 by interpolating the along-track data onto a longitude–latitude grid of resolution $0.75° \times 0.25°$ from 60 to 88° N. In Sect. 4.3.1 we provide details about the interpolation method used. In Sect. 4.3.2 we provide a global estimate of the standard error on the monthly $\eta'$ fields. Finally, based on the analysis of the error given in Sect. 4.3.2, in Sect. 4.3.3 we describe the steps taken in phase of interpolation to reduce the noise due to residual sub-monthly variability.

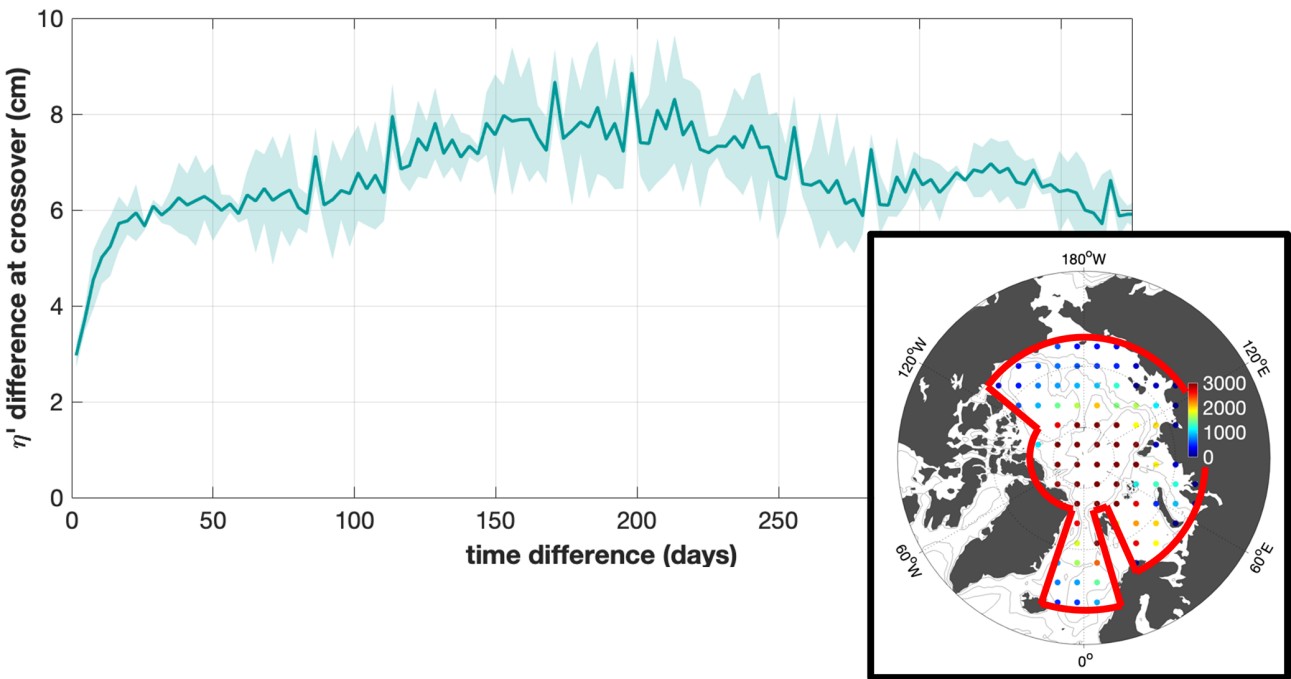

**Figure 4.** Absolute value of the $\eta'$ difference at crossovers between satellite tracks in a period of time of up to 1 year, computed using data inside the red line in the inset panel. The solid line in the main panel is the crossover difference averaged every 3 d; the shaded area shows the standard deviation of the crossover difference, averaged every half a day. Crossover differences were computed using data within 100 km around the locations indicated in the inset panel. The color of the dots in the inset panel indicates the number of crossovers found around that location.

### 4.3.1 Interpolation using the Data-Interpolating Variational Analysis

Along-track data were interpolated to obtain $\eta'$ fields on a regular latitude–longitude grid. We used the Data-Interpolating Variational Analysis (DIVA, Troupin et al., 2012; Barth et al., 2014), a tool based on a technique called the variational inverse method (VIM, Brasseur and Haus, 1991). DIVA has been successfully applied in the past by several studies (e.g., Tyberghein et al., 2012; Capet et al., 2014; Lenartz et al., 2017; Iona et al., 2018; Belgacem et al., 2021) to a variety of data types (e.g., temperature, salinity, chlorophyll concentration, nutrients, air pollutants), spatial and temporal extents, and regions (global ocean, Mediterranean Sea, Black Sea). We applied this method for the first time to altimetry observations in the Arctic Ocean.

Rixen et al. (2000) showed that the performance of the VIM is comparable to the widely used optimal interpolation technique (in its original formulation, Bretherton et al., 1976). DIVA offers advantages when treating large datasets in regions of complex topography. One advantage is that the VIM maintains low numerical cost when the number of data points is large compared to the grid points (Rixen et al., 2000). This was suitable for our case, with a number of data points in 1 month ($\sim 10^5$) 10 times larger than the number of grid points ($\sim 10^4$). Furthermore, DIVA allows us to naturally decouple basins that are not physically connected by

using a regularity constraint based on the gradient and Laplacian of the gridded field (Troupin et al., 2010).

A short description of the working principles of DIVA is given in the following. The optimal field in the VIM is found by minimizing a cost function (e.g., Brasseur and Haus, 1991; Troupin et al., 2012; Barth et al., 2014, 2021), which satisfies the basic requirements for the analysis field $\varphi$, such as its closeness to the data and its regularity (no abrupt changes). DIVA formalizes these principles in a cost function as follows.

$$J(\varphi) = \sum_{i=1}^{N} \mu \left[ d_i - \varphi(\mathbf{x}_i) \right]^2 + \int_{\Omega} \frac{1}{L^4} \varphi^2$$
$$+ \frac{2}{L^2} \nabla \varphi \cdot \nabla \varphi + (\nabla^2 \varphi)^2 \, \mathrm{d}\Omega \qquad (8)$$

In Eq. (8), the first term ensures the closeness of the analysis field to the data. This is achieved by globally minimizing the difference between $\varphi$ at the data locations $\mathbf{x}_i$ TS13 and the data themselves $d_i$, which are associated with a weight $\mu$. The second term generates a smooth field over the domain $\Omega$ (Troupin et al., 2012), where $L$ defines the length scale over which the data should be propagated spatially. In general, the field $\varphi$ and the data $d_i$ should be understood as anomalies relative to a background estimate. The data weights $\mu$ are directly proportional to the signal-to-noise ratio $\lambda$ (ratio of the

error variance of the background estimate, $\sigma^2$, to the error variance of the observations, $\epsilon^2$) and inversely proportional to the square of the length scale $L$ (Brasseur et al., 1996):

$$\mu = 4\pi \frac{\lambda}{L^2}. \tag{9}$$

As explained further below, the interpretation of weights $\mu$ in terms of the signal-to-noise ratio allows DIVA to calculate error maps at a low computational cost.

The length scale $L$ is a parameter related to the distance over which ocean state variables decorrelate. In the Arctic Ocean, boundary currents can be as narrow as a few tens of kilometers (Beszczynska-Möller et al., 2012; Pnyushkov et al., 2015). Even though satellite altimetry provides a tool to investigate the surface expression of these dynamic features, maps of sea surface height in the Arctic are commonly smoothed over hundreds of kilometers (Kwok and Morison, 2016; Pujol et al., 2016; Armitage et al., 2016; Rose et al., 2019; Prandi et al., 2021). In order to retain the possibility of resolving Arctic boundary currents in our maps of geostrophic currents, we generated monthly maps using a length scale smaller than a hundred kilometers while relying on a background field derived using a large length scale. That is, we applied a two-step interpolation as follows. We first computed a background field using all $\eta'$ observations in the period 2011–2020, interpolated with a large length scale of 300 km. In a second step, we interpolated weekly subsets of the data relative to the background field using a short length scale of 50 km. Finally, as explained in Sect. 4.3.3, we obtained monthly maps by averaging four weekly fields. The scale used in the second step (50 km) defines the spatial scale beyond which we expect to resolve the temporal variations, as assessed and discussed in Sects. 5.2 and 6.3. This length scale ensured that we would have enough tie points for the interpolation (see Fig. 3a and b) while attempting to resolve scales shorter than in previous works. From Fig. 3a we can see that the least constrained region is the ice-covered ocean north of the Canadian Arctic Archipelago, where in winter there are on average fewer than 50 data points per month per 100 km$^2$.

The signal-to-noise ratio $\lambda$ is to be interpreted as the ratio between the fraction of data variance that is representative of the final analysis field ($\sigma^2$) and the fraction that is to be considered noise ($\epsilon^2$). The latter might in general include the observational error as well as representativity errors (e.g., instantaneous measurements are not a good representation of a long-term mean). One possible way to give an estimate of $\lambda$ is the generalized cross-validation technique (Troupin et al., 2010). However, this technique has led past studies to an overestimation of $\lambda$ when applied to non-independent data (Troupin et al., 2010), in particular in applications where averaged fields were created (Troupin et al., 2012; Lauvset et al., 2016; Belgacem et al., 2021). We estimated instead $\epsilon^2$ and $\sigma^2$ separately from $\eta'$ observations, based on the approximation that weekly data subsets were not subject to error of

representation (see Sect. 4.2.3 and 4.3.3). We thus considered the observational uncertainty, calculated in Sect. 4.2.3, to be the dominant source of noise over a period of 1 week, and hence took $\epsilon$ equal to 3 cm. Under the same assumption, we took $\sigma$ equal to 8.2 cm, estimated by taking the data signal $\sigma^2$ equal to the spatial variance of weekly data subsets, averaged in the period 2011–2020. The signal-to-noise ratio $\lambda$, defined by the ratio of $\sigma^2$ over $\epsilon^2$, was therefore 7.5. This estimate lies in the range of values ($\lambda \sim 1$–10) used in previous studies applying DIVA to generate averaged fields (Troupin et al., 2010, 2012; Tyberghein et al., 2012; Lauvset et al., 2016; Iona et al., 2018; Watelet et al., 2020; Belgacem et al., 2021). Furthermore, we noted that the standard deviation of our analyzed CE3 $\eta'$ fields changed by only a small fraction when varying $\lambda$ in the range of 1–10.

Along with the gridded fields, DIVA has the capability to provide associated error maps using several different methods, each having different computational costs. A review of the methods is provided by Beckers et al. (2014). Among these, we selected the clever poor man's estimate due to its fast calculation (CPME, Beckers et al., 2014). The CPME speeds up calculations by circumventing the extraction of the data covariance matrix, which is never explicitly computed in DIVA. The CPME takes advantage of the fact that the absolute interpolation error scaled by the variance of the background field can be derived with a good approximation by applying the DIVA analysis to a vector of unit values (Beckers et al., 2014). We thus generated maps of relative error via the CPME, given as a fraction of the variance of the background field. These maps allow the user to assess the data coverage given by the distribution of the data in space, scaled by the length scale $L$ and the signal-to-noise ratio $\lambda$.

### 4.3.2 Error in monthly fields

The standard error in the monthly $\eta'$ fields comprises a component arising from the observational uncertainty and another arising from the representativity error due to unresolved sub-monthly variability. We provide here an average estimate of these two contributions over the area shown in the inset panel of Fig. 4, computed as follows.

The component deriving from the observational uncertainty was obtained for each month as the uncertainty estimate of an individual measurement, derived from the crossover analysis (i.e., 3 cm, Sect. 4.2.3), divided by the square root of the average number of data points per cell per month. This component of the standard error, averaged over the period 2011–2020, amounts to 1.7 cm. The monthly component stemming from the sub-monthly variability was first calculated at each grid point as the standard deviation of the four weekly $\eta'$ values divided by the square root of 4. To verify that the weekly interpolated fields were statistically independent, we calculated the integral timescale of $\eta'$ (Emery and Thomson, 2001) from the time series of weekly values between 2011 and 2020, high-pass-filtered with a cutoff of 2

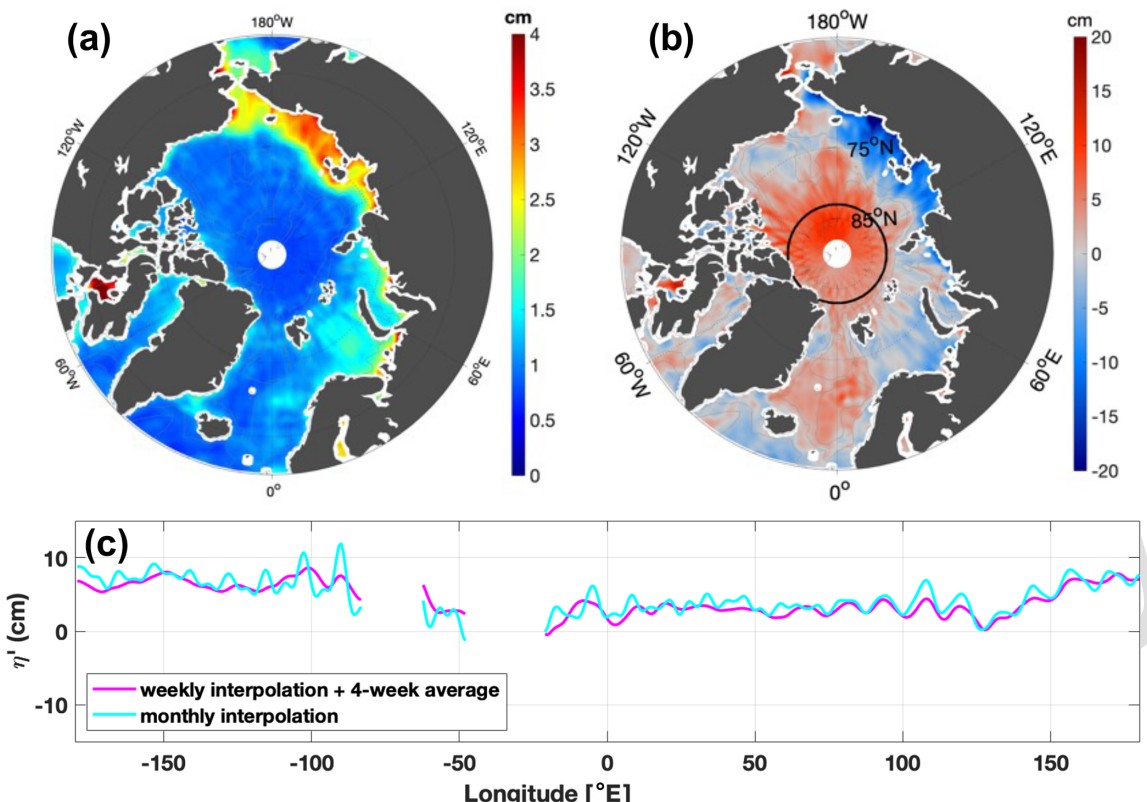

**Figure 5.** Residual sub-monthly variability in the $\eta'$ gridded field. **(a)** The sub-monthly contribution to the standard error on monthly $\eta'$ maps, computed from weekly maps, averaged over the period 2011–2020. **(b)** The July 2015 monthly gridded $\eta'$ field obtained by interpolating monthly data input. **(c)** $\eta'$ along a latitude (83° N) circle; $\eta'$ obtained from weekly interpolation plus averaging (Fig. 6a) and from monthly interpolation **(b)** are shown with magenta and cyan lines, respectively. Bathymetry contours are drawn at 100, 1000 and 2500 m depths.

months to exclude longer decorrelation timescales. Across the whole Arctic we found an integral timescale of about 1 week, in agreement with results by Landy et al. (2021), supporting the hypothesis of statistically independent weekly fields. The monthly average standard error yielded by this approach is 1.1 cm over the period 2011–2020. The time average distribution of this contribution is displayed in Fig. 5a, which shows values of 1–4 cm in areas shallower than 100 m, with peak values of more than 3 cm in the East Siberian Sea. We assumed that the observational and sub-monthly contributions to the error are independent and computed the total error by adding them in quadrature. This amounts to 2 cm, which is a conservative estimate of the total standard error on monthly averages over the period 2011–2020.

### 4.3.3   Minimization of sub-monthly variability

As seen in Sect. 4.2.3, the residual sub-monthly variability produces marked meridional trackiness if the interpolation is performed on a monthly set of $\eta'$ observations (see also Appendix C). To further reduce the sub-monthly variability, we performed the interpolation on weekly data subsets instead. Monthly $\eta'$ maps were obtained as the average of four

weekly maps. Furthermore, the analysis of the $\eta'$ decorrelation timescales presented in Sect. 4.3.2 showed that weekly estimates are statistically independent. Therefore, the associated interpolation error was computed by adding in quadrature four weekly error maps. By comparing Fig. 5b with Fig. 6a, one can appreciate how trackiness is reduced in a given month over the entire Arctic. In Fig. 5c we show in detail the $\eta'$ profile along a latitude circle as an example of the trackiness reduction obtained thanks to this approach. The field displayed in Fig. 5a shows the contribution of the sub-monthly variability to the error on the monthly $\eta'$ fields, computed as explained in Sect. 4.3.2.

### 4.4   Gridded geostrophic velocity

Monthly $\eta$ fields were reconstructed by adding up the $\langle\eta\rangle$ DTU17MDT, the $\eta'$ background field over the period 2011–2020 and the gridded $\eta'$ maps resulting from the steps described above. Based on the $\eta$ fields, geostrophic velocity was computed on the output grid following Eq. (4), with partial derivatives approximated by finite differences. The components of velocity on the longitude–latitude grid at indices

$i, j$ are given by

$$\begin{cases} u_{g,ij} = -\frac{g}{f\,R_E}\,\frac{\eta_{i+1,j}-\eta_{i-1,j}}{\theta_{i+1,j}-\theta_{i-1,j}}, \\ v_{g,ij} = \frac{g}{f\,R_E}\,\frac{1}{\cos(\theta_{ij})}\,\frac{\eta_{i+1,j}-\eta_{i-1,j}}{\Phi_{i+1,j}-\Phi_{i-1,j}}, \end{cases} \quad (10)$$

where variables are defined as for Eq. (4).

## 5  Results

Here we first describe the characteristics of the monthly maps of $\eta'$ and geostrophic velocity $(u_g, v_g)$, then present the results of their comparison with independent datasets, and lastly display the most prominent aspects of the $\eta'$ and $(u_g, v_g)$ seasonal cycle.

### 5.1  Monthly fields of sea surface height anomaly and geostrophic velocity

As an example to describe the general characteristics of a given monthly map over the 2011–2020 period, here we present results from the month of July 2015. Figure 6 shows fields of $\eta'$, relative error (associated with the interpolation) and $(u_g, v_g)$ for July 2015. The description below makes reference to the Arctic Ocean sub-regions and surface circulation pathways presented in Fig. 1.

In the $\eta'$ monthly fields we generally find that there are extended regions of either positive or negative values. In Fig. 6a, for instance, $\eta'$ is positive in deep regions, i.e., in the Nordic Seas and across the Arctic Deep Basins, and negative over the shelf seas. $\eta'$ also varies within these regions, being, for instance, maximum ($\sim 10$ cm) north of 85° N and minimum in the East Siberian Sea. Superimposed on these large-scale patterns, residual meridional trackiness appears south of 80° N, especially in shallow areas, where the error related to the residual sub-monthly variability is highest (Fig. 5c).

The relative error for the month of July 2015 is on average 0.23, with a minimum below 0.2 around the North Pole and a maximum above 0.3 south of 70° N (Fig. 6b). The largest relative error values are found in regions with data gaps (see the weekly data distribution in Fig. 3a): (i) south of 75° N, where the distance between the satellite tracks increases considerably; (ii) in a zonal band around 80° N, where the weekly data distribution is not uniform due to the satellite orbit geometry; (iii) in regions covered by multiyear ice during winter months (Fig. 3a, right).

In Fig. 6c we present the geostrophic velocity field $(u_g, v_g)$, with background colors highlighting monthly speed anomalies relative to the 2011–2020 mean speed. The distribution of anomalies aligns well with known circulation pathways, such as slope currents found along steep bottom topography gradients, or large-scale current patterns like the Beaufort Gyre and the Transpolar Drift. For instance, speed anomalies displayed in Fig. 6c show that in July 2015 currents were weak around the Nordic Seas (East Greenland Current, West Spitsbergen Current and Norwegian Atlantic Current) and at the Laptev Sea continental slope (Arctic Boundary Current), while they were intensified in the westernmost branch of the Beaufort Gyre and in the Pacific Water inflow across the Bering Strait. This indicates that our dataset yields realistic variability over a large span of the Arctic Ocean. Still, there are confined areas where speed anomalies do not follow circulation pathways but rather appear along meridionally elongated stripes. These patterns result from gradients between residual $\eta'$ sub-monthly variability and do not correspond to real monthly velocity anomaly.

### 5.2  Comparison to independent datasets

We evaluated both $\eta'$ and $(u_g, v_g)$ fields against independent data in order to (i) test the robustness of the monthly $\eta'$ fields, both in ice-free and ice-covered regions, by comparison to the satellite-derived, gridded CPOM dataset, (ii) verify the spatial consistency of our $\eta'$ fields in the Fram Strait, a region of transition between ice-covered and ice-free ocean, and (iii) assess the agreement in time and space between our gridded $\eta'$ and $(u_g, v_g)$ fields and mooring-based data in seasonally ice-covered regions over a time span of a few years.

#### 5.2.1  Sea surface height

We first compared our gridded $\eta'$ fields with the CPOM DOT. In this instance we aimed at testing the robustness of the temporal variability of our monthly $\eta'$ fields over the entire Arctic. A comparison of Arctic regional products to independent altimetry products was previously either not done (Armitage et al., 2016; Kwok and Morison, 2016; Rose et al., 2019) or only used products that were not tailored to ice-covered regions (Prandi et al., 2021). Results show good agreement of our gridded $\eta'$ fields with the CPOM DOT over most of the Arctic domain, with a correlation between datasets above 0.7 for 85 % of the grid points (Fig. 7a). The comparison yields lower correlation values (0.3 to 0.7) along the Canadian and Greenland coasts (where the multiyear ice persists for most of the year) and in sparse areas of the central Arctic and in the Barents Sea. Only in less than 1 % of the domain is the correlation below 0.3 (Baffin Bay). The RMSD (Fig. 7b) exhibits low values (2 to 4 cm) over more than 80 % of the domain, including most of the regions with water depths greater than 100 m. The RMSD is high (7–8 cm) over the East Siberian Sea and Chukchi Sea, where the error due to sub-monthly variability is also the highest. These results seem to indicate that altimetry-derived month-to-month variability is generally robust in relation to the methodology applied, also in ice-covered regions, with a few exceptions that we will discuss in Sect. 6.

Secondly, we wanted to demonstrate that in the Fram Strait, a transition zone between ice-covered areas in the west and ice-free areas in the east, the spatial sea surface slope associated with the local ocean circulation is retained in our

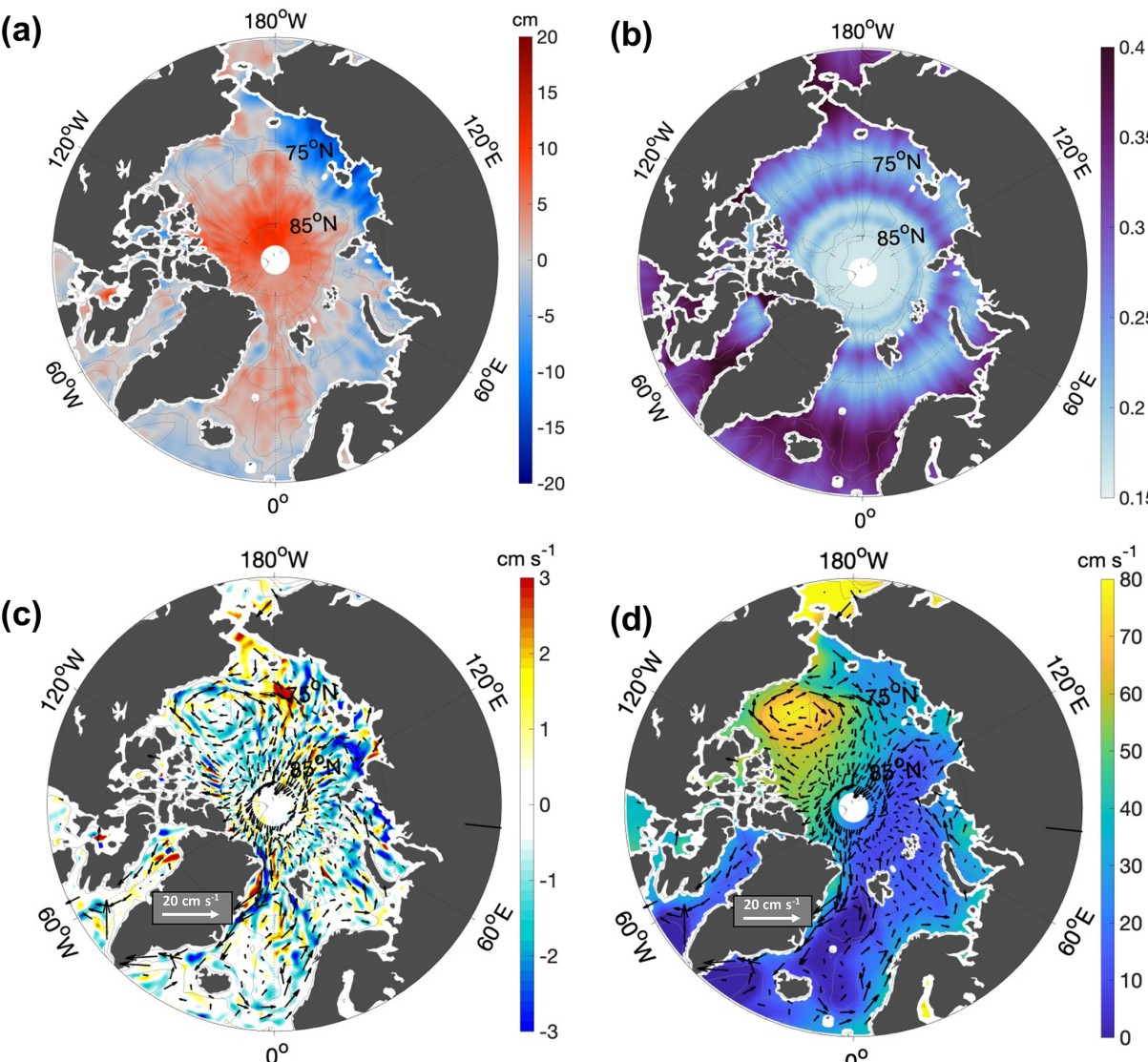

**Figure 6.** Example of monthly gridded fields included in the final data product for the month of July 2015. **(a)** The $\eta'$ field above the 2011–2020 background field. **(b)** Relative error field on the interpolated $\eta'$, given as a fraction of the variance of the background field. **(c)** $(u_g, v_g)$ field. Arrows in panel **(c)** represent the absolute $(u_g, v_g)$ field for the month of July 2015, whereas color highlights the anomaly of the monthly geostrophic speed $(V_g = \sqrt{u_g^2 + v_g^2})$ with respect to the mean geostrophic speed over the period 2011–2020. **(d)** Dynamic ocean topography ($\eta$, background color) and the associated geostrophic velocity field (as in panel **c**). Bathymetry contours are drawn at 100, 1000 and 2500 m depths.

$\eta$ fields (computed as described in Sect. 4.4). In order to do this, we carried out a comparison with independent hydrography data not used for the offset correction displayed in Fig. 2. In Fig. 8 we display two cross sections of altimetry-derived $\eta$ across the Fram Strait, in the months of June 2011 and June 2012, against dynamic height from ship-based CTD sections plus ocean bottom pressure from GRACE data. In the East Greenland Current (7 to 2° W), at the transition between ice-covered and ice-free regions in the western Fram Strait, the broad cross-shelf variation in $\eta$ is comparable to in situ data. We note though that the strong local gradients between 7 and 4° W, each spanning a distance of about 30–40 km, are not captured. This is likely due, on the one hand, to the 50 km length scale used to smooth altimetry data and on the other hand to the fact that profiles from the altimetry fields represent monthly averages, while those from in situ data represent a snapshot of hydrography over the course of a few days. Despite the above-mentioned differences, this comparison seems to indicate that the differential offset correction applied to altimetry data between ice-free and ice-

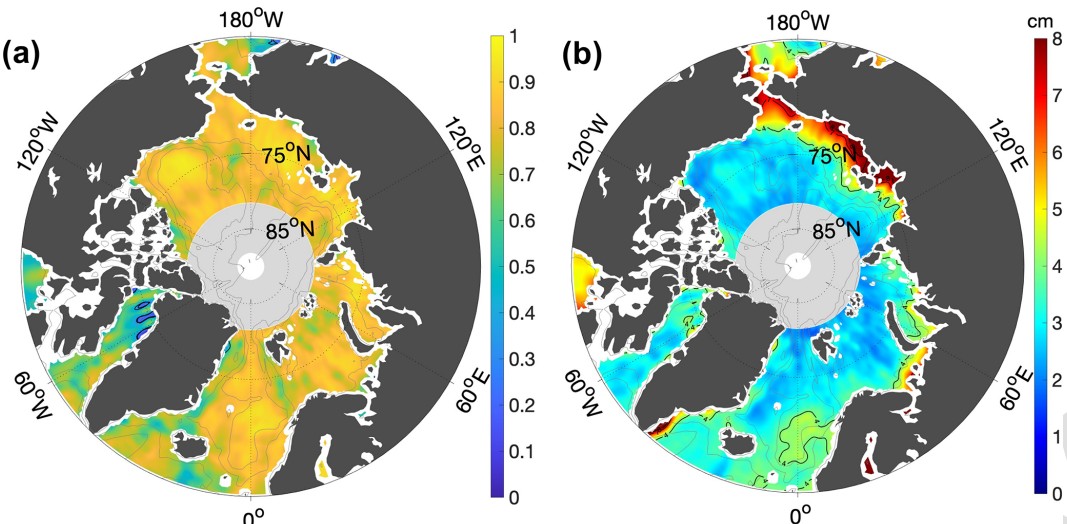

**Figure 7. (a)** Pearson's correlation coefficient and **(b)** RMSD between the gridded $\eta'$ fields as derived in this work and the CPOM DOT published by Armitage et al. (2016). Each dataset was referred to its own average over the period 2011–2014 before comparison. In panel **(a)**, correlation is $< 0.3$ and $p$ value is $> 0.05$ in the small areas in the Baffin Bay encircled by a thick black line. In panel **(b)**, thick black lines are contours of 4, 7 and 8 cm. The region shaded in grey north of $82°$ N is not included in the comparison because it is not covered by the CPOM DOT. Bathymetry contours (dotted lines) are drawn at 100, 1000 and 2500 m depths.

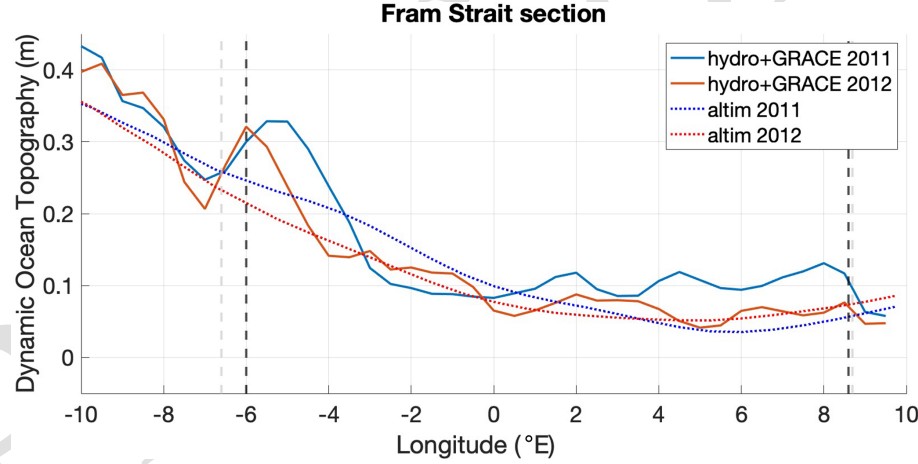

**Figure 8.** Cross sections of $\eta$ across the Fram Strait at $79°\,50'$ N in June 2011 and June 2012. Altimetry-derived $\eta$ is displayed against steric height $h_S$ from in situ hydrographic sections plus ocean bottom pressure $h_P$ from GRACE. Light grey and dark grey vertical dashed lines indicate the 300 and 400 m isobaths, respectively.

covered areas (shown in Fig. 2) has preserved the broad spatial sea surface slope associated with the East Greenland Current.

After having demonstrated the spatial consistency of our dataset, we now turn to the question to which degree the time variability in the gridded $\eta'$ fields is representative of independently observed variability. With this purpose, we compared in situ time series from five moorings at different locations in the Arctic Ocean to time series extracted locally from our $\eta'$ fields. Time series of $\eta'$ from altimetry and $\eta'_P + \eta'_S$ from mooring data (computed as described in

Sect. 4.1) are shown in Fig. 9. The correlation between the altimetry and mooring time series is higher than 0.5, with a $p$ value lower than or equal to 0.06 at all five sites. The correlation is highest at the M1_4p6 mooring, where in situ hydrography is measured up to 26 m below the sea surface. Sea surface height from altimetry and mooring follow roughly similar patterns, varying within a range of $\pm10$ cm over the comparison period at all the sites. The sea level at the moorings in the Eurasian Arctic (FS_S, AC and M1_4p6) is characterized by seasonal oscillations, with the signal amplitude decreasing during winter, starting in October, and increas-

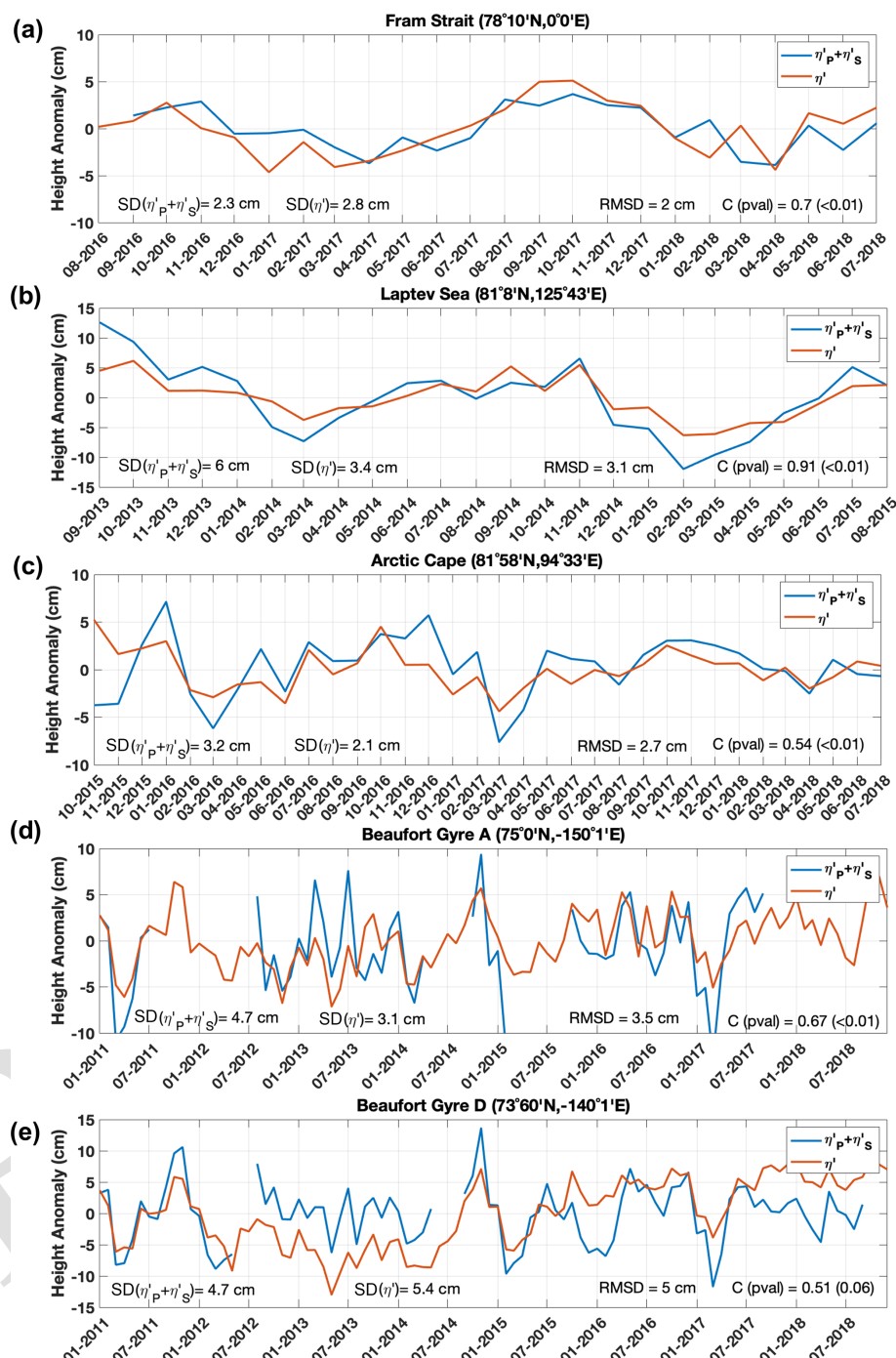

**Figure 9.** The sea level anomaly ($\eta'_P + \eta'_S$) derived from data at moorings **(a)** FS_S, **(b)** AC, **(c)** M1_4p6, **(d)** A and **(e)** D (blue line) is displayed against the $\eta'$ interpolated at the mooring location (red line). Standard deviations of $\eta'$ and $\eta'_P + \eta'_S$ are displayed in the bottom left corner and the RMSD and correlation coefficient in the bottom right corner (Pearson's correlation coefficient, where the $p$ value was computed using the effective number of degrees of freedom, Emery and Thomson, 2001).

ing during summer, starting in March. In the Beaufort Sea seasonality has a similar phase, though strong intra-seasonal and interannual variability is also present. At moorings A and D, altimetry and in situ data show agreement at interannual timescales. This is visible, for instance, in alternating years of a non-detectable seasonal cycle (2012, 2013, 2015, 2016) and a peaked seasonal cycle (2011, 2014). On the other hand, a trend between 2013 and 2018 is evident in the altimetry time series at mooring D but is not present in the in situ time series. At all the sites, particularly in the Beaufort Sea, short-

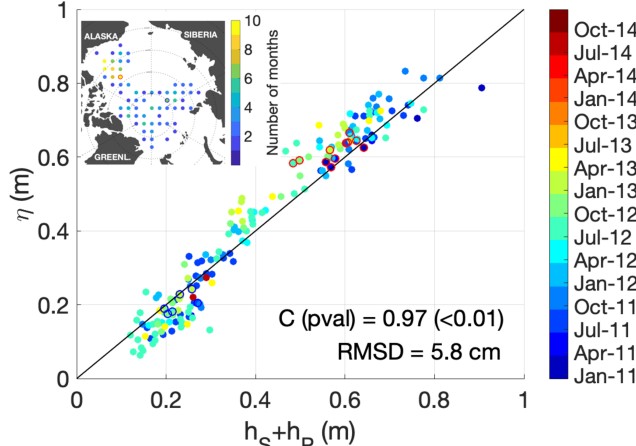

**Figure 10.** The steric height plus ocean bottom pressure ($h_S + h_P$) derived from hydrographic profiles in the Arctic Deep Basins and GRACE data are displayed against the gridded altimetry-derived $\eta$, each averaged in equal area grid cells with a resolution of 200 km. The red and blue circles indicate data points from the most populated grid cells in the Amerasian and Eurasian basins, respectively.

term variability appears in phase most of the time, though month-to-month variations are larger in mooring data than altimetry, as reflected in the relatively high RMSD between them.

Finally, we compared our gridded $\eta$ estimates to data from the Arctic Deep Basins, presented in Sect. 4.2.1, on a monthly basis. Each data source was spatially averaged for each month on the same equal area grid with a resolution of 200 km (Fig. 10). There is good agreement between the dynamic ocean topography estimated from the two methods, with a correlation coefficient of 0.97 and an RMSD of 5.8 cm over a range of about 70 cm. This indicates that the basin-scale gradients in sea surface height between the western and eastern Arctic Ocean are preserved in our $\eta$ maps. The spread accounts for different temporal and spatial coverage of in situ and satellite data within each cell. Despite this spread, when we isolate data points from the most populated grid cells in the Amerasian and Eurasian basins, we see that the temporal variability of in situ data is still reasonably represented by altimetry estimates.

### 5.2.2   Velocity

Satellite-derived maps of surface geostrophic velocity offer the advantage of a quasi-synoptic view of ocean surface currents and their variability. We evaluated this variability locally by comparison to mooring near-surface velocity. Given that the variability represented by the two data sources differs to some extent due to the different nature of the measurements and the spatio-temporal resolution, in our comparison we further assessed what the spatial and temporal scales are over which these two data sources provide consistent information on the underlying variability.

### Correlation and RMSD at mooring locations

The agreement of altimetry-derived and in situ velocities at mooring locations is summarized in Table 4. Hovmöller diagrams of velocity normal to the Fram Strait and Laptev Sea mooring lines are displayed in Figs. 11 and 12, while the comparison of the speed and bearing at moorings in the Beaufort Sea and the Chukchi Sea is shown in Fig. 13. In the Fram Strait, the correlation is significant ($p$ value $< 0.05$) and higher than 0.3 at moorings F3 to F5 across the continental slope in the eastern part of the strait. At these three moorings, both the mean $v_n$ and $v_{ni}$ are consistently positive and higher than or comparable to the corresponding standard deviation. The correlation is highest at mooring F3, the mooring with the longest continuous time series. Over the Laptev Sea continental slope, the correlation is highest at the M1_1 mooring in the uppermost part of the slope. At this mooring, $v_{ni}$ is on average 4 times larger than further down the slope. At the moorings located down the slope, the correlation is lower, still significant at mooring M1_4 but non-significant at moorings M1_2 and M1_3. In the Beaufort Sea, the mean currents' speed and their standard deviation are much lower than along the continental slopes, and the variability is dominated by month-to-month variations. The agreement is best at mooring B, located in the northern branch of the Beaufort Gyre. As already noted by Armitage et al. (2016), the current-bearing ADCP measurements at this mooring in the years 2011 to 2013 are offset around late summer, which might indicate a data bias related to different deployments; the in situ and altimetry-bearing estimates agree more closely after late summer 2014. At mooring A, closest to the center of the Beaufort Gyre, low correlation is associated with very weak mean currents ($< 2\,\mathrm{cm\,s^{-1}}$) and large oscillations in the currents' direction. Despite the low correlation coefficient, both data sources clearly identify a period, between 2013 and 2016, when the current bearing is consistently more stable and slowly rotating clockwise. Currents at the Chukchi Sea moorings S1 and S3 are faster than in the basin and correlation values higher. While at mooring S3 both current speed and bearing are well captured by altimetry, at mooring S1 altimetry shows an offset of about 40° clockwise.

### Spatial and temporal resolution

By examining the mean and standard deviation of velocity along the mooring lines, we note differences between gridded altimetry and in situ data in terms of spatial and temporal resolution. The mean $v_n$ shows low spatial variability and smooth transitions between nearby sites. Note that this variability is governed by the averaging scales underlying the DTU17MDT product. The scales captured by the DTU17MDT are defined by the resolution of the geoid model

**Table 4.** Comparison of velocity from altimetry and mooring data. Moorings from the two mooring lines are listed, from top to bottom, respectively, as westernmost to easternmost in the Fram Strait and southernmost to northernmost in the Laptev Sea continental slope. At these two arrays, the component normal to the array is compared (northward and eastward, respectively). In the Beaufort Sea, current speed and bearing are compared. The first two columns display Pearson's correlation coefficient and the RMSD; correlations with $p$ value $< 0.05$ are highlighted in bold ($p$ values were computed using the effective number of degrees of freedom, Emery and Thomson, 2001). The next four columns show the mean and standard deviation of the altimetry-derived and mooring velocity.

| | Correlation | RMSD (cm s$^{-1}$) | Mean altim. (cm s$^{-1}$) | Mean moor. (cm s$^{-1}$) | SD altim. (cm s$^{-1}$) | SD moor. (cm s$^{-1}$) |
|---|---|---|---|---|---|---|
| Fram Strait | | | | | | |
| F10 | 0.01 | 5.3 | −7.3 | −7.9 | 1.8 | 5.0 |
| F16 | **0.22** | 6.8 | −4.3 | 1.1 | 1.5 | 7.1 |
| F15 | 0.17 | 6.7 | −2.9 | −0.8 | 1.3 | 6.9 |
| F8 | −0.28 | 5.8 | −1.8 | 6.1 | 1.2 | 5.5 |
| F7 | −0.18 | 7.2 | −0.3 | −2.5 | 1.3 | 6.9 |
| F6 | 0.16 | 6.8 | 0.9 | −2.6 | 1.3 | 7.0 |
| F5 | **0.33** | 6.3 | 2.8 | 5.3 | 1.7 | 6.7 |
| F4 | **0.38** | 6.7 | 4.0 | 6.0 | 1.8 | 6.2 |
| F3 | **0.54** | 6.8 | 4.5 | 17.0 | 1.7 | 7.6 |
| F2 | 0.30 | 7.2 | 4.5 | 18.1 | 1.8 | 7.6 |
| Laptev Sea | | | | | | |
| M1_1 | **0.77** | 5.7 | 4.7 | 12.1 | 2.3 | 7.4 |
| M1_2 | 0.06 | 4.6 | 4.6 | 3.5 | 2.2 | 4.2 |
| M1_3 | 0.17 | 2.0 | 4.1 | 3.4 | 1.3 | 1.8 |
| M1_4 | **0.45** | 1.1 | 2.9 | 1.6 | 0.8 | 1.2 |
| Beaufort Sea | | | | | | |
| A speed (cm s$^{-1}$) | 0.03 | 2.0 | 1.5 | 2.1 | 1.1 | 1.7 |
| A bearing (°) | 0.12 | 144 | 310 | 255 | 73 | 103 |
| B speed (cm s$^{-1}$) | **0.53** | 2.3 | 3.5 | 3.6 | 0.9 | 2.6 |
| B bearing (°) | **0.26** | 76 | 83 | 100 | 24 | 68 |
| D speed (cm s$^{-1}$) | 0.18 | 1.7 | 3.1 | 2.5 | 1.1 | 1.4 |
| D bearing (°) | 0.24 | 51 | 166 | 151 | 26 | 50 |
| Chukchi Sea | | | | | | |
| S1 speed (cm s$^{-1}$) | **0.59** | 2.2 | 4.4 | 4.7 | 1.6 | 2.7 |
| S1 bearing (°) | 0.21 | 125 | 69 | 31 | 36 | 41 |
| S3 speed (cm s$^{-1}$) | **0.69** | 3.7 | 5 | 7.1 | 1.9 | 3.8 |
| S3 bearing (°) | **0.50** | 61 | 100 | 106 | 44 | 64 |

used to compute it. Previous studies, mentioning also the geoid model used by DTU17MDT, indicate that these scales are not smaller than 100 km (Gruber and Willberg, 2019; Bruinsma et al., 2014; Farrell et al., 2012). These large scales contrast with the high spatial variability of the $v_{ni}$ mean flow, which is derived by pointwise measurements. This is shown, for instance, by abrupt changes between moorings F15 and F8 (27 km apart) and F8 and F7 (25 km apart) or between M1_1 and M1_2 (11 km apart). The high spatial variability observed by the mooring data is ascribable to the small Arctic first baroclinic Rossby radius, which is below 10 km in the two study regions (Nurser and Bacon, 2014; von Appen et al., 2016; Pnyushkov et al., 2015). Despite the different spatial resolution of source data, in our comparison at the two mooring lines we observe that altimetry-derived geostrophic velocity capture transitions in the moored velocity from strong to weak mean flow occurring over distances of about 50–70 km. For instance, both altimetry and mooring data in the Fram Strait show a change from a strong mean flow at moorings F2–F3 to a weak mean flow at moorings F5–F6, within a distance of 50–60 km. At the Laptev Sea continental slope, where the in situ measured current intensity significantly decreases from mooring M1_1 to mooring M1_2, altimetry-derived currents only weaken significantly over a distance of about 70 km, at the position of mooring M1_3.

Furthermore, in the altimetry dataset the time variability associated with mesoscale processes is smoothed out due to

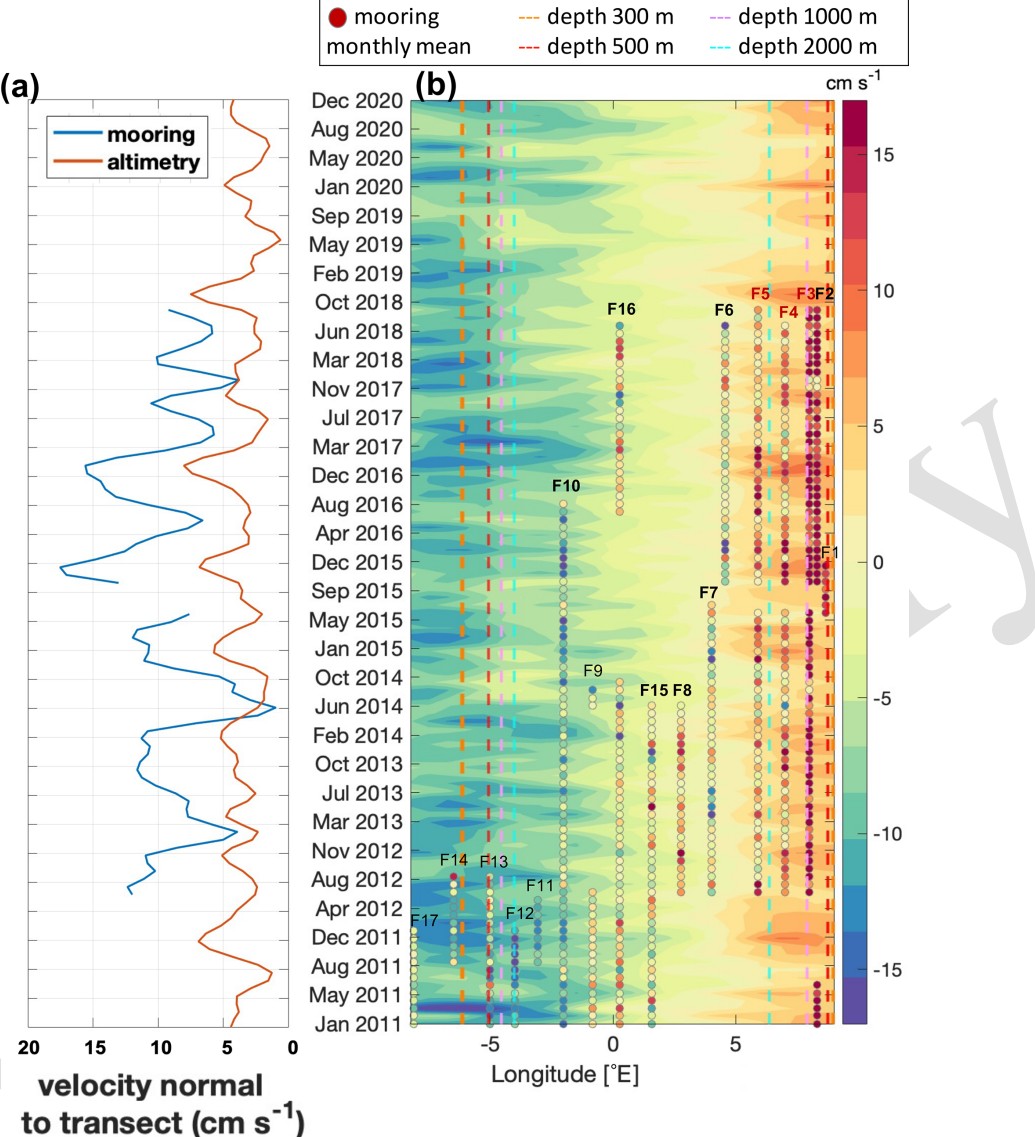

**Figure 11.** The altimetry-derived geostrophic velocity is shown against the in situ surface velocity at the mooring transects in the Fram Strait, along latitude 78° 50′ N (see Fig. 1). The component of the velocity normal to the transect is evaluated, and positive values represent northward velocity. **(a)** Longitudinal average of altimetry and in situ velocity across moorings indicated with red letters in panel **(b)** (corresponding to test 2; see Sect. 5.2.2); both time series have been filtered with a 4-month low-pass filter. **(b)** Hovmöller diagram representing the monthly temporal evolution of the altimetry-derived cross-transect geostrophic velocity. The circles represent monthly mean values of in situ cross-transect velocity. The mooring's names are displayed on top of each mooring's series; at moorings with bold letters, data covered a period longer than 24 months.

the 50 km decorrelation scale applied through the interpolation. This is reflected in the standard deviation of $v_n$, which is about 4 to 5 times smaller than that of $v_{ni}$ at most moorings. To establish the spatial scales over which altimetry-derived currents approximate best the temporal variability of in situ measured currents, we compared spatially averaged $v_n$ and $v_{ni}$ at the two mooring lines. We performed five tests, averaging data over sets of at least two moorings chosen among those closest to the shelf break (tests 1 to 5 in Table 5). In order to take into account the fact that time series of moorings closer to each other are less independent, we performed a weighted average of the $v_n$ and $v_{ni}$ time series. Each mooring was assigned a weight proportional to its distance to the two neighboring moorings (e.g., for mooring $j$, the weight is $w_j = \frac{d_{j,j-1} + d_{j,j+1}}{2}$, where $d$ is the distance) or to the one neighboring mooring (e.g., if $j$ is the first mooring in a set, its weight will be $w_j = d_{j,j+1}$).

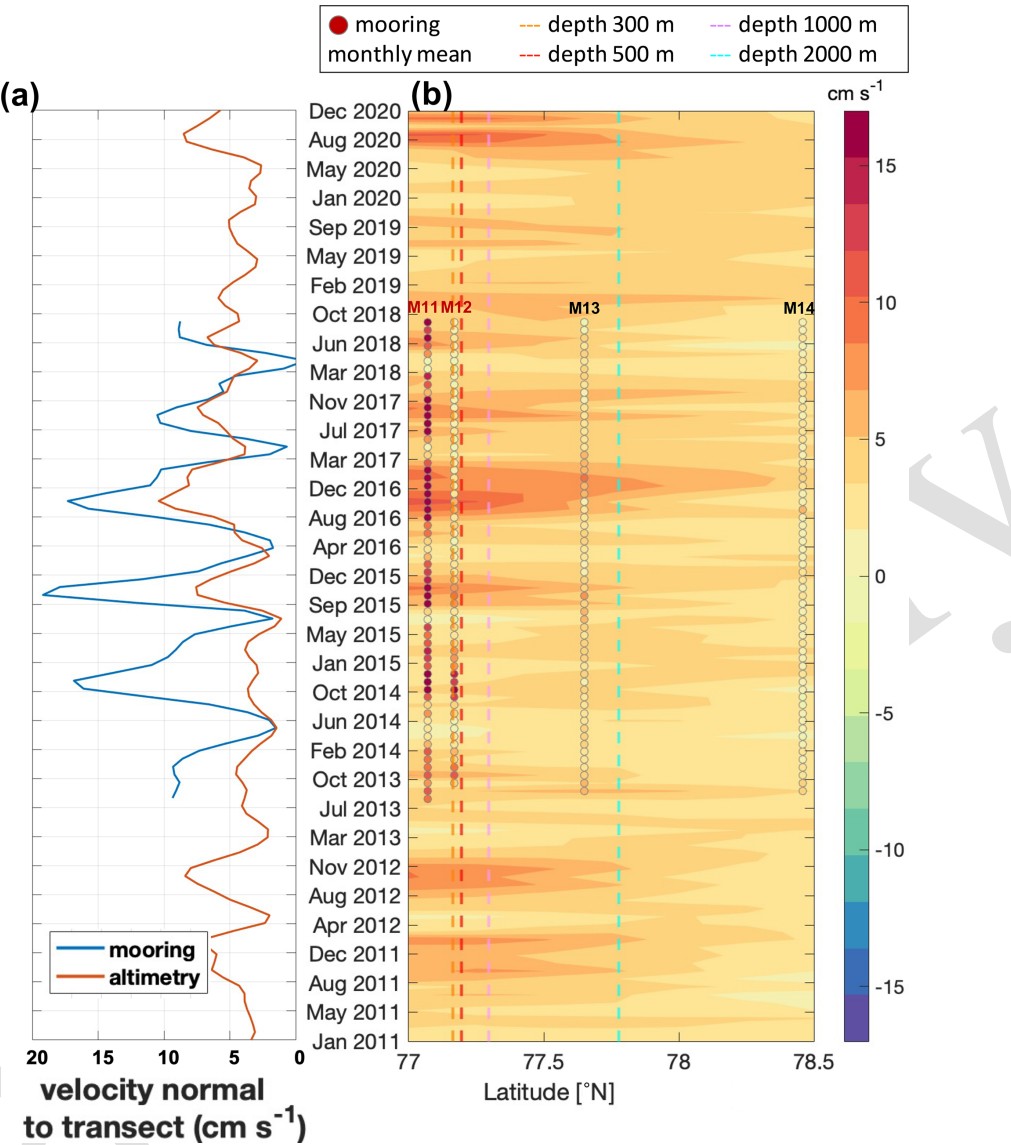

**Figure 12.** As in Fig. 11 but for velocities at the Laptev Sea continental slope, along longitude 126° 50′ E (see Fig. 1). The component of the velocity normal to the transect is positive eastward. The time series in panel **(a)** correspond to test 4 (see text).

**Table 5.** Comparison of spatially averaged altimetry and mooring velocity at the mooring lines. Each test (described in Sect. 5.2.2, "Spatial and temporal resolution") corresponds to the averaging of data from two or more moorings (names of moorings used in each test and cross-flow distance covered by them are indicated in the header). The first two rows show the Pearson correlation coefficient and RMSD between horizontally averaged $v_n$ and $v_{ni}$. The last two rows show correlations at frequencies lower and higher than 4 months. All correlations in this table have a $p$ value $< 0.01$, computed using the effective number of degrees of freedom (Emery and Thomson, 2001).

|  | Test 1 20 km F3, F4 | Test 2 45 km F3 to F5 | Test 3 85 km F3 to F7 | Test 4 11 km M1_1, M1_2 | Test 5 61 km M1_1 to M1_3 |
|---|---|---|---|---|---|
| Correlation | 0.55 | 0.62 | 0.49 | 0.61 | 0.36 |
| RMSD (cm s$^{-1}$) | 4.9 | 3.1 | 2.6 | 4.0 | 2.3 |
| Correlation 4 months' low pass | 0.63 | 0.68 | 0.61 | 0.62 | 0.37 |
| Correlation 4 months' high pass | 0.37 | 0.33 | 0.11 | 0.58 | 0.27 |

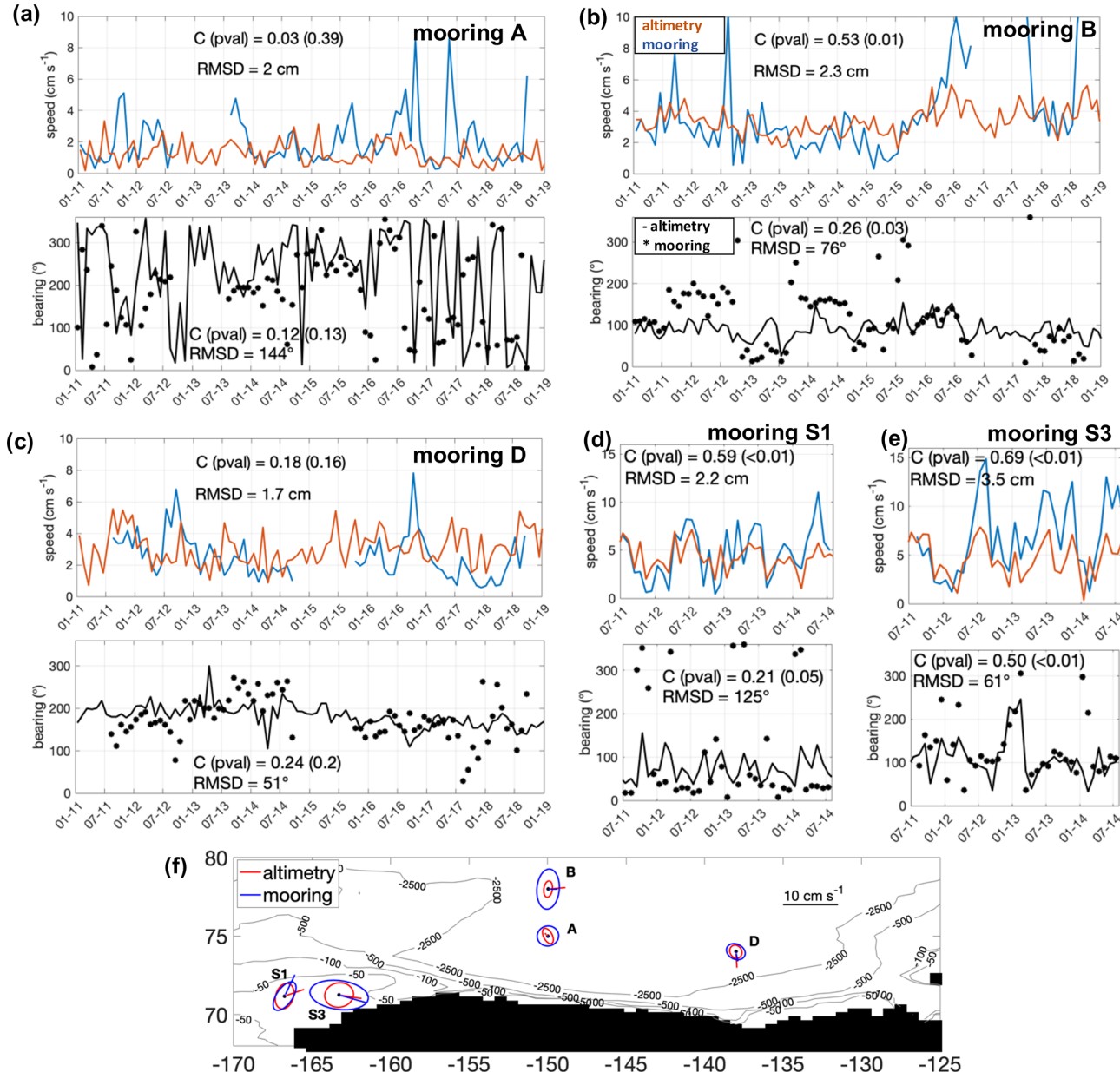

**Figure 13.** The altimetry-derived currents' speed and bearing are shown against the in situ measured ones at moorings A **(a)**, B **(b)** and D **(c)** in the Beaufort Sea and moorings S1 **(d)** and S3 **(e)** in the Chukchi Sea (see Fig. 1). **(f)** Mean currents and variance ellipses at each location.

In the Fram Strait, averaging over moorings F3 to F5 (test 2, spanning a distance of 45 km) yielded a correlation higher than that using data only from the F3 mooring (where the pointwise comparison was highest; compare Tables 4 and 5). Results from tests 1 and 3 yielded correlations comparable to that at F3. All three tests reduced the RMSD by about 2–3 cm with respect to that at F3. At the Laptev Sea continental slope, neither test 4 nor test 5 improved the correlation with respect to the comparison at the M1_1 mooring. This appears plausible, as visual inspection of the in situ observations reveals the slope current to be restricted to site M1_1

and not to extend out to M1_2 and beyond (Fig. 12b). This indicates that the spacing of the moorings is likely too wide to adequately resolve the scales of the slope current. Nevertheless, both tests 4 and 5 reduced the RMSD with respect to the value at M1_1 (2–4 cm lower).

Finally, we looked at the correlation between the spatially averaged $v_n$ and $v_{ni}$ in two frequency bands (Table 5), namely, seasonal to interannual (lower than 4 months) and intra-annual (higher than 4 months). In the seasonal-to-interannual frequency band, $v_n$ and $v_{ni}$ correlate better than or equally without filtering (Table 5), whereas in the intra-

seasonal frequency band the correlation worsens. The percentage of variance explained by each frequency band in each dataset was evaluated as

$$E = 100 \left( 1 - \frac{\mathrm{var}(x - x_F)}{\mathrm{var}(x)} \right), \tag{11}$$

where $x$ is the horizontally averaged $v_n$ or $v_{ni}$ time series (tests 1 to 5), and $x_F$ is the corresponding filtered time series. We find that seasonal-to-interannual frequencies explain most of the variability of the spatially averaged $v_n$ and $v_{ni}$. They constitute about 80 % of the total variability in the Fram Strait and about 90 % at the Laptev Sea continental slope.

In Figs. 11a and 12a we can see that, both in the Fram Strait and at the Laptev Sea continental slope, the current variability at timescales larger than 4 months is dominated by seasonal oscillations, which have similar characteristics in the altimetry and mooring data. The seasonal cycles of $v_n$ and $v_{ni}$ are in phase there, with peaks occurring in winter and troughs in early summer. Furthermore, $v_n$ and $v_{ni}$ show similarities in the interannual variability. For instance, in the Fram Strait both datasets feature a double-peaked seasonal oscillation in some years (e.g., winters 2013–2014, 2017–2018). At the Laptev Sea continental slope the seasonal cycle amplitude decreases in both datasets between 2016 and 2018. In the western Arctic (Fig. 13), seasonal oscillations are observed at the moorings in the Chukchi Sea, where altimetry and in situ data consistently show maximum speed in mid-summer and minimum speed in mid-winter. In contrast, a seasonal cycle is not clearly recognizable in the Beaufort Gyre currents at the locations of the A, B and D moorings.

In summary, the comparisons with moored observations suggest that the satellite-derived velocities can provide reliable information both on time mean properties and seasonal changes in the flow field on spatial scales exceeding 50–70 km.

## 5.3 Seasonal cycle

The seasonality of the Arctic sea level and surface currents has been studied in several previous works (e.g., Volkov et al., 2013; Armitage et al., 2016; Beszczynska-Möller et al., 2012; Baumann et al., 2018), giving us the opportunity to assess our dataset based on this literature. We defined the seasonal cycle of $\eta'$, following Volkov et al. (2013), as the harmonic least-square fit to $\eta'$ with a period of 1 year:

$$\eta'_{\mathrm{seas}} = A \cos \left[ 2\pi \left( \frac{t - \alpha}{P} \right) \right], \tag{12}$$

where $t$ is time, $P = 12$ months is the oscillation period, $A$ is the amplitude of the $\eta'$ seasonal cycle and $\alpha$ is its phase (i.e., month when the maximum occurs). We evaluated the fraction of variance explained by $\eta'_{\mathrm{seas}}$ at each grid point following Eq. (11), with $\eta'$ as $x$ and $\eta'_{\mathrm{seas}}$ as $x_F$.

In the following text CE4 we give an overview of the seasonal cycle observed in our product, with emphasis on the regions where it explains a high fraction of the total variability.

### 5.3.1 Sea surface height

The amplitude $A$ and the phase $\alpha$ of the $\eta'$ seasonal cycle are displayed in Fig. 14. The amplitude ranges between 1 and 8 cm, with values above 3 cm in the shallow shelf regions, in the southwestern Canada Basin and in the Nordic Seas (Fig. 14a). In these regions and in the Eurasian Basin, the seasonal cycle explains more than 20 % of the total variability. $\eta'_{\mathrm{seas}}$ is maximum in early winter across the Arctic Ocean, even though not uniformly (Fig. 14b). On the Eurasian side, we see a clear divide between deep and shallow regions, with $\eta'_{\mathrm{seas}}$ peaking earliest (September–October) in the Nordic Seas and the Eurasian Basin and later (November–December) all along the Eurasian shelves, from the Barents Sea to the East Siberian Sea. On the Amerasian side, $\eta'_{\mathrm{seas}}$ peaks earliest in the southwestern Canada Basin and later on the Chukchi Shelf.

In Fig. 14c we also display the monthly climatology of $\eta'$ observed in selected regions, computed as the January-to-December monthly averages over the years 2011–2020. We see that the harmonic fit is a good approximation of the climatology in most of these regions. One exception is the secondary peak in June–July exhibited by the climatology in the Canada Basin, the Eurasian Basin, the Laptev Sea and East Siberian Sea, and the northeastern Greenland Shelf.

### 5.3.2 Geostrophic velocity

Figure 15 shows the winter (January to March) and summer (June to August) average fields of $(u_g, v_g)$ over the period 2011–2020. Seasonal speed anomalies are most pronounced south of 80° N, namely, along the shelf edges, in some coastal regions, in the southern Canada Basin and in the Barents Sea. The strongest variation in current speed between summer and winter is about 3 cm s$^{-1}$. The time of the seasonal maximum of some of the main Arctic currents is shown in Table 6. From the comparison between summer and winter we observe a basin-wide, coherent seasonal acceleration of the Arctic slope currents in winter and a deceleration in summer. The speed of these slope currents peaks between September and April. That is, currents along the Nansen Basin shelf break, between the Fram Strait and the Lomonosov Ridge, peak in early winter (September to December), currents along the eastern shelf break of the Nordic Seas, in the Barents Sea and in the Baffin Bay peak in mid winter (November to February), and the East Greenland Current peaks in late winter (February to April). Seasonality is also recognizable in some currents not along the continental slopes, for instance, currents in the Kara Sea (peak between November and January), in the southern and western branches of the Beaufort Gyre (peaks in November–January

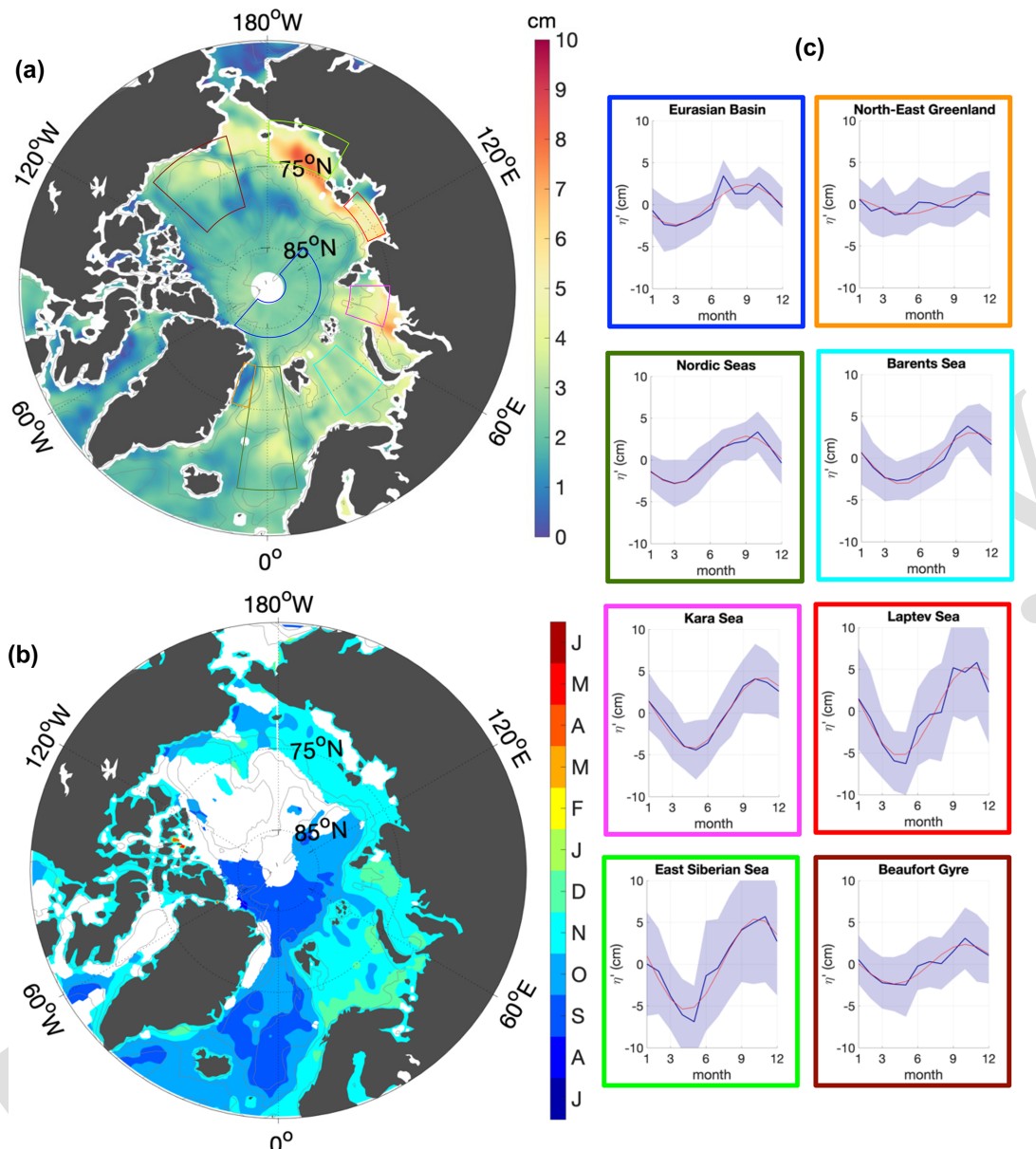

**Figure 14. (a)** Amplitude and **(b)** phase of the $\eta'$ annual harmonic oscillation between 2011 and 2018. Blanked areas in panel **(b)** are those areas where the seasonal cycle explains less than 20 % of the total variance. **(c)** Panels representing the $\eta'$ monthly climatology (blue line, with standard deviation as shading) and the $\eta'_{\mathrm{seas}}$ (red line) averaged over the areas marked in the map with the corresponding color. Bathymetry contours are drawn at 100, 1000 and 2500 m depth.

and March–May, respectively) and in the Chukchi Sea (peak in June–August).

## 6   Discussion

The dataset presented in this paper provides Arctic-wide monthly maps of sea surface height anomaly $\eta'$ up to 88° N, derived from CryoSat-2 altimetry observations, over the time span of 10 years. In addition, we also provide the associated geostrophic velocity $(u_g, v_g)$, which was not available be-

fore north of 82° N. Both sea surface height and geostrophic velocity were validated against independent data, including one satellite product and in situ data in both ice-covered and ice-free regions. The extensive validation, covering a large portion of the Arctic, provided a robust assessment of the capability of our satellite product to reveal realistic spatio-temporal variability in agreement with in situ observations. Furthermore, the comparison to an independent altimetry product allowed us to assess the consistency of the variability

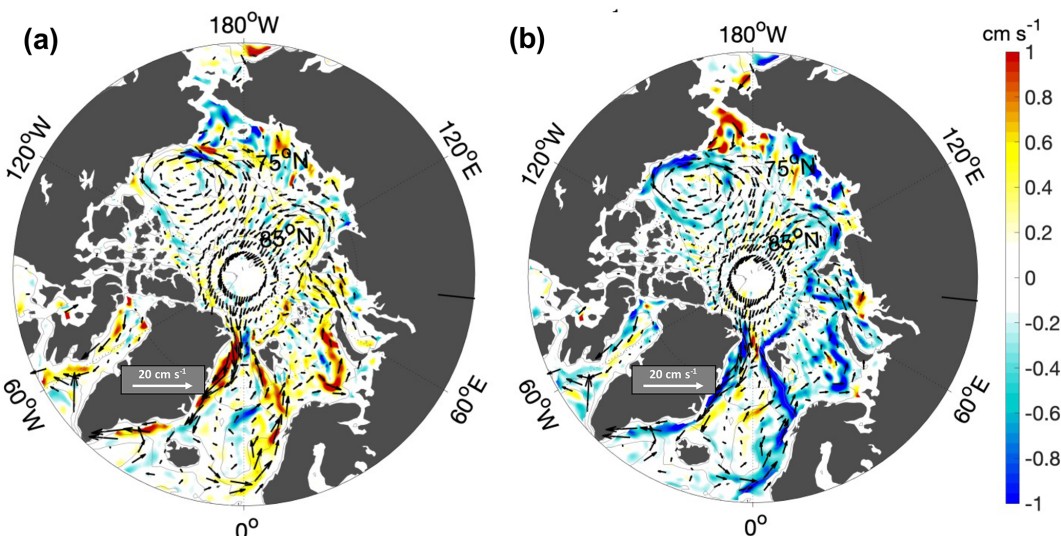

**Figure 15.** Average $(u_g, v_g)$ fields over the **(a)** winter months January–February–March and the **(b)** summer months June–July–August. Bathymetry contours are drawn at 100, 1000 and 2500 m depth. Arrows and colors are to be interpreted as described for Fig. 6c.

at monthly to interannual timescales between independently derived products.

In the following, we use results from the validation to discuss the following points. First, our multiyear, Arctic-wide comparison of monthly $\eta'$ fields against an independent altimetry product revealed isolated sites with low correlation across datasets, despite the general agreement. Thus, we discuss whether this is related to the methods used. Then, we discuss our results from the comparison to in situ data in terms of the spatial and temporal resolution of our altimetry dataset and the underlying dynamic regimes. Finally, we place our findings on the seasonal cycle of sea surface height and geostrophic flow in the context of the previous literature.

## 6.1 Impact of methodology

The comparison of our dataset with the CPOM DOT (Sect. 5.2.1) yielded a correlation higher than 0.7 over 85 % of the domain. This indicates that month-to-month variability is generally robust in relation to the methodology applied, an encouraging result that has not yet emerged from previous studies. However, correlation coefficients are lower in some regions, with non-negligible differences between the datasets there. Many data sources and processing steps, i.e., just as many sources of uncertainty, are taken to generate monthly gridded sea surface height. As a starting point to support future product development, in the following we discuss what the methodological steps are that may generate the largest differences between these two datasets.

In the first place, source data used for the two products in ice-covered areas (ellipsoidal heights from CryoSat-2) have been derived by applying different algorithms for the processing of satellite waveforms. Regional differences in the

monthly fields might thus have occurred due to different data densities. For instance, in our comparison the correlation is low in some areas of the ice-covered Arctic, where leads are detected based on surface classification techniques. These differ substantially between studies, depending on the parameters considered or the statistical techniques applied, and have to date been a source of uncertainty (Dettmering et al., 2018). More conservative techniques might be used to discard observations and reduce uncertainty. This results however in low data density in the central and western Arctic, where the most compact multiyear ice is located and leads are sparse (Willmes and Heinemann, 2016). Furthermore, generating data over the marginal ice zone still represents a challenge to overcome. This is because neither ocean-type retrackers nor ice-type retrackers are well suited to processing altimetry waveforms there, resulting in noisy or unusable data (Quartly et al., 2019). It is perhaps not surprising then that our comparison shows correlation values lower than 0.7 in open ocean areas of the central Arctic and the Baffin Bay, where large patches of low ice concentration form at the end of summer.

Secondly, different approaches were used in this study and in Armitage et al. (2016) to reduce unresolved sub-monthly variability in along-track data. On the one hand, we analyzed the sea level variability on sub-monthly timescales, finding that in the Arctic this variability can yield substantial noise in the monthly gridded fields, especially in the shelf regions (Sect. 4.3.2). To reduce this noise we took two steps. First, we aimed at removing the highest possible fraction of high-frequency variability (due to ocean tides and the ocean response to pressure plus wind) by using up-to-date corrections (FES2014 and DAC, respectively). Second, we applied the DIVA analysis to weekly rather than monthly data input

(Sect. 4.3.3). On the other hand, no dedicated analysis of this source of noise was made in Armitage et al. (2016), where relatively old corrections for tide- and wind-related high-frequency variability were used (FES2004 and IB, respectively). In their study, a generic approach is used to reduce spatial noise, which consists in bin-averaging the along-track data over longitude–latitude grid cells with a resolution of $2° \times 0.5°$. These different approaches are most likely responsible for the differences between the two datasets in regions where the sub-monthly variability is strongest (Fig. 5c). For example, the two datasets have the highest RMSD in the East Siberian Sea and Chukchi Sea regions, where we found that DAC yielded the most improvements over the IB (Appendix B). Furthermore, relatively low correlation values are shown in the Barents Sea and the Baffin Bay, two regions of strong tidal variability where the tidal model FES2014 performs better than the previous version FES2004 (Appendix A) and in general better than most of the models available for the Arctic Ocean (Cancet et al., 2018).

Finally, this study and the study by Armitage et al. (2016) applied different methods to grid the data into monthly estimates. In this work, we used a two-step gridding method which, in a first step, provides a background field as a backup field and, in a second step, grids the data into monthly fields using a decorrelation radius of 50 km. The gridding method applied in Armitage et al. (2016) instead does not rely on a background field, but rather smooths the previously binned data with a Gaussian convolution filter of radius 100 km. In the first place, these two different approaches provide different results when the interpolation is not well constrained by data, for instance, as mentioned above, in regions of very compact ice or in the marginal ice zone. Furthermore, in the two cases data are gridded using different decorrelation radii, which sets the actual dataset resolution. This therefore introduces a difference in the resolution between the two datasets, regardless of the chosen grid.

## 6.2 Pointwise comparison between satellite altimetry retrievals and in situ data

Pointwise comparison with independent in situ mooring-based time series of sea surface height was used to assess the time variability of our altimetry product in three separate regions of the central Arctic, i.e., the Fram Strait, the Nansen Basin and the Beaufort Sea (Fig. 9). Results showed that altimetry and in situ data yield roughly consistent temporal patterns, exhibiting variability on similar timescales. For instance, a seasonal signal is visible at all sites with a common peak in fall, more clearly defined in the Eurasian Arctic and more variable in intensity in the Beaufort Sea, and month-to-month variability is enhanced in the Beaufort Sea. Correlation is significant at all sites, with coefficients ranging between 0.5 and 0.9. The RMSD between altimetry and open ocean mooring observations (2–5 cm) was consistent with other studies comparing altimetry to in situ observations. For instance, studies comparing altimetry data with tide gauges found RMSD values in the range of 2 to 12 cm across the Arctic (Volkov and Pujol, 2012; Armitage et al., 2016; Rose et al., 2019). A similar result was obtained via comparison of altimetry with steric height from hydrographic profiles in the Arctic Deep Basins (Kwok and Morison, 2011).

Despite the broad agreement between altimetry- and mooring-derived sea surface height observations from the open ocean (Fig. 9), correlations were lower than or comparable to previous studies which compared altimetry to near-shore tide gauge measurements (Volkov and Pujol, 2012; Armitage et al., 2016; Rose et al., 2019). This can be expected for a few reasons. First, while tide gauges measure sea surface height, directly comparable to altimetry, estimates of sea surface height from mooring data include uncertainty resulting from limited vertical resolution. This agrees with our results, showing that altimetry correlates best with mooring data at the site with the most continuous and extended vertical sampling (M_4p6). Secondly, we expect tide gauge measurements to correlate better with altimetry given that sea surface height variability near the coast shows larger amplitudes than in the open ocean (see Fig. 14).

Altimetry-derived geostrophic velocity was compared to moored velocity at 19 moorings, including moorings located at important exchange gateways of the Arctic, i.e., in the Fram Strait and the Chukchi Sea. Results showed that the correlation is significant where variability on timescales of seasonal or longer is present. In contrast, large differences emerge on intra-seasonal timescales, especially in regions of weak mean currents (central Fram Strait, interior of the Eurasian Basin, Beaufort Sea). Another study by Armitage et al. (2017) compared altimetry-derived currents with moored current velocity from the interior of the Beaufort Sea. Correlation values in Armitage et al. (2017) were lower than or equal to 0.54, in line with our findings at most mooring sites, except for moorings M1_1, S1 and S3, which show correlation values larger than 0.6. The RMSD values of $1–2\,\mathrm{cm\,s^{-1}}$ over weak mean currents of $2–4\,\mathrm{cm\,s^{-1}}$ found in Armitage et al. (2017) also agree well with what we find in the same region.

## 6.3 Temporal and spatial resolution of altimetry-derived monthly estimates

The comparison between our altimetry-derived dataset and in situ data showed that agreement between these two data sources can be expected at scales of about 50–70 km and larger, both for sea surface height and surface circulation.

Large-scale patterns of altimetry-derived dynamic ocean topography are consistent with hydrography-based sea surface height in the central Arctic (Fig. 10). For instance, both data sources consistently show a decrease in sea surface height of about 70–80 cm from the Amerasian to Eurasian basins, which was also found in the comparisons carried out in Armitage et al. (2016) and Kwok and Morison (2016). Ad-

ditionally, comparing monthly profiles to snapshots from hydrographic sections in the Fram Strait (Fig. 8), we saw that the cyclonic shape in sea surface height characteristic of the Nordic Seas is well reproduced, with a minimum in the center of the strait. Furthermore, this comparison shows the continuity of the altimetry field across the ice edge, in the western part of the strait. On the other hand, we note that altimetry is unable to resolve gradients in sea surface height on short scales of about 30–40 km, which are captured by in situ profiles in the western part of the Fram Strait. This is consistent with the smoothing applied to the altimeter data in the gridding process, where a 50 km decorrelation radius was used.

The large spatial extent covered by the two mooring arrays allowed us to examine the agreement of altimetry and in situ velocity over different dynamic regimes and spatio-temporal scales. We found that correlation is highest in regions where the flow variability is dominated by steady currents (e.g., boundary currents) and lowest where it is dominated by non-stationary eddy activity. The change in correlation with dynamic regime can be explained by considering the different samplings of mesoscale activity by moorings and by altimetry. Mesoscale features are not resolved in our monthly altimetry fields because of the 50 km smoothing scale used in the interpolation. This is equivalent to about 10 times the local first-mode baroclinic Rossby radius (Nurser and Bacon, 2014; von Appen et al., 2016; Pnyushkov et al., 2015), which roughly sets the horizontal scale of mesoscale eddies. For this reason we also see that the correlation coefficient improves when time series are low-pass-filtered to retain only the seasonal and longer timescales, thereby suppressing the effect of mesoscale eddies (test results in Table 5).

We find evidence of different correlations in connection with the dynamic regime at both mooring lines and in the western Arctic. In the Fram Strait, altimetry and in situ data show the highest correlation on the shore and continental slope east of 5° E, within the West Spitsbergen Current, with the maximum correlation in the core, non-eddying part of the current (mooring F3, Beszczynska-Möller et al., 2012). In the Laptev Sea the correlation is highest at mooring M1_1, close to the shelf break, where the Arctic Boundary Current is strongest (Aksenov et al., 2011; Baumann et al., 2018). In contrast, in both regions the correlation breaks down where mean currents are slow and the mesoscale activity is enhanced. That is, the correlation is low and non-significant at moorings in the central Fram Strait, where the surface circulation is dominated by westward eddy propagation (von Appen et al., 2016; Hattermann et al., 2016). The comparison of temporally filtered time series in this region (test 3 in Table 5) clearly shows that the strongest decrease in correlation happens on intra-seasonal timescales, while the correlation on longer timescales remains stable. Similarly, correlation was low in the offshore part of the Laptev Sea continental slope, where current speed is low and eddy activity increases (Pnyushkov et al., 2015, 2018; Baumann et al., 2018). Our comparison with data from the moorings in the Beau-

fort Sea and the Chukchi Sea also supports the above results. While there is significant correlation of altimetry with data from within the relatively strong Pacific Water inflow in the Chukchi Sea (Woodgate et al., 2005; Fang et al., 2020), low and generally non-significant correlation is shown with data from the weak flow of the central Beaufort Gyre. In particular, the correlation is lowest at the two moorings located in the southern portion of the Beaufort Gyre, where the highest concentration of eddies is found (Zhao et al., 2016).

We thus used in situ surface velocities to evaluate the effective spatial and temporal resolution of altimetry-derived monthly currents. Looking at the mean spatial variability, we found that altimetry captures transitions from strong to weak currents occurring over distances of 50–70 km. Accordingly, spatially averaged velocity generally has a higher temporal correlation than velocity at a single mooring. For instance, in the region of the West Spitsbergen Current, the correlation is higher when averaging over about 50 km relative to about 20 km (compare tests 1 and 2 in Table 5). This indicates that the boundary current variability as observed by our altimetry-derived velocity agrees most closely with the in situ observed variability when both are averaged across at least 50 km. On the other hand, slightly lower correlation values are obtained when averaging data further into the central Fram Strait (about 80 km; see test 3 in Table 5), due to the different dynamic regime. There, eddies are a source of variability on intra-seasonal timescales, which is not resolved by our altimetry maps and biases the large-scale average velocity from moorings. By low-pass-filtering velocities with a cutoff of 4 months, we found, indeed, that the correlation between altimetry and in situ data is increased both in the Fram Strait and at the Laptev Sea continental slope.

The considerations above suggest that our maps of monthly geostrophic velocities for the Arctic Ocean can resolve seasonal to interannual variability of boundary currents wider than about 50 km. The current that we analyzed more in detail in this respect is the West Spitsbergen Current, which had not been shown to be resolved using altimetry before this study (Armitage et al., 2017). We suggest however that this result is relevant also for studies that wish to investigate other relatively narrow slope current systems of the Arctic Ocean, for instance, the Arctic Boundary Current (Baumann et al., 2018; Pérez Hernández et al., 2019) or the Chukchi Slope Current (Min et al., 2019). We do not however resolve mesoscale variability at intra-seasonal timescales. Past studies have shown that multi-altimeter integration is necessary over a large part of the global ocean to resolve mesoscale activity (e.g., Pujol et al., 2010). In a recent study, Prandi et al. (2021) combined altimeter data from three satellites flying over the Arctic Ocean, covering a time span of 3 years. Using tide gauge data as a reference signal, they estimate that the improvement in the resolution of the mapped sea surface height from a single altimeter product to a combined one is on average from 3 to 1.5 months. This indicates that future efforts to increase the temporal resolution

of gridded altimetry products should be directed towards the integration of data from more than one satellite. This comes however at the expense of the duration of the time series, which is limited in the Arctic region by relatively short overlap periods of satellite activity.

## 6.4 Seasonality

The sea surface height seasonal cycle is driven by changes in the steric component (due to vertical buoyancy fluxes and advection) and the mass component (due to water accumulation or release, precipitation, evaporation, and river runoff). Previous studies identified the seasonal cycle as the dominant component of the sea surface height variability in the Arctic (e.g., Volkov et al., 2013; Armitage et al., 2016; Müller et al., 2019). Our results confirm these findings, showing that this variability explains a fraction higher than 20 % of the total variability in large areas of the Arctic, including the Arctic Shelves, the Nordic Seas, the Eurasian Basin and part of the Canada Basin. Additionally, from monthly time series of altimetry-derived and in situ geostrophic velocity, we found that the variability of boundary currents at seasonal to interannual timescales dominates over intra-seasonal variability.

Large-scale features emerge in the seasonal cycle of $\eta'$ and $(u_g, v_g)$. First, $\eta'$ has a seasonal maximum in winter, between September and December, over most of the Arctic. Furthermore, we found that the amplitude of the seasonal cycle of $\eta'$ as well as the fraction of variability explained are higher over the shelf regions than in open ocean regions of the Arctic interior. Lastly, we found that geostrophic currents consistently strengthen along the continental slopes in winter and weaken in summer. These features find support in the literature. The wintertime occurrence of the $\eta'$ seasonal maximum is in agreement with previous studies of steric height seasonality from in situ data. For instance, from hydrographic profiles, the steric height was found to peak between September and November in the Greenland and Norwegian seas (Siegismund et al., 2007), in the central Barents Sea (Volkov et al., 2013) and in the Canada Basin (Proshutinsky et al., 2009). In addition, the secondary peak appearing from the $\eta'$ climatology in most of the Arctic interior (Fig. 14c) is in agreement with the late summer peak of ocean mass found by Peralta-Ferriz and Morison (2010) from GRACE data. Overall, both the Arctic-wide occurrence of the winter maximum and the decoupling of shallow and deep regions agree well with the first two empirical orthogonal functions of sea surface height derived by Bulczak et al. (2015) and Armitage et al. (2016): a basin-wide oscillation with a wintertime maximum and an anti-phase oscillation between shelf regions and deep basins. Finally, the strengthening of boundary currents in winter was documented for several regions by previous studies based on in situ data, satellite data and model output (Table 6). Exceptions to the wintertime peak are however also observed, for instance, in the Pacific Water inflow. Both our dataset and the past literature reveal that currents there are weaker in winter, when stronger winds oppose the flow driven by the Pacific pressure head into the Arctic (Woodgate et al., 2005; Peralta-Ferriz and Woodgate, 2017). Our dataset is thus able to describe the seasonality of sea surface height and geostrophic currents across the Arctic, consistent with previous studies.

## 7 Data availability

The final monthly maps of the sea surface height anomaly and geostrophic velocity (2011–2020) are available at https://doi.org/10.1594/PANGAEA.931869 (Doglioni et al., 2021d). This data file also includes, as auxiliary fields, (i) the relative error on the sea surface height, (ii) the mean dynamic topography for the period 2011–2020, (iii) the along-track sea surface height anomaly derived as described in Sect. 4.2 of this paper and (iv) its monthly binned values over the cells of the output grid.

The time series of steric height and bottom pressure equivalent height at moorings FS_S, AC and M1_4p6 and moorings A and D, as processed in this work, are available at https://doi.org/10.1594/PANGAEA.931871 (Doglioni et al., 2021c), https://doi.org/10.1594/PANGAEA.931878 (Doglioni et al., 2021a), https://doi.org/10.1594/PANGAEA.931875 (Doglioni et al., 2021b) and https://doi.org/10.1594/PANGAEA.949695 (Doglioni et al., 2022), respectively.

## 8 Conclusions

With this work we aim to contribute to basin-scale observational studies of the Arctic Ocean circulation by providing a new Arctic-wide gridded product of the satellite-derived sea surface height anomaly $(\eta')$ and geostrophic velocity $(u_g, v_g)$. We present monthly maps of $\eta'$ and $(u_g, v_g)$, spanning the years 2011 to 2020, covering both the ice-free and ice-covered parts of the ocean. We believe that this dataset can be used to study variability with spatial scales above 50 km, at seasonal to interannual timescales. Furthermore, both the gridded and along-track data provided with this dataset offer a valuable tool for constraining and evaluating new ocean reanalysis products for the Arctic (e.g., Nguyen et al., 2021; Fukumori et al., 2021).

We find that sub-monthly variability in the Arctic Ocean, due to tides and the response to wind and pressure, is a source of noise in the $\eta'$ monthly fields. We reduced this noise by (i) applying up-to-date altimetry corrections and (ii) averaging four weekly interpolated maps. The comparison of our dataset with the independent altimetry dataset CPOM DOT at monthly timescales yields a correlation coefficient higher than 0.7 over most of the Arctic, indicating that altimetry-derived sea surface height variability is relatively robust with respect to the methodology applied. Isolated areas of lower agreement are attributable to differences in the data coverage in ice-covered regions, in the approach used to correct

**Table 6.** Time of seasonal maximum occurrence in the currents of the Arctic Ocean in the results of this study. The abbreviations of the currents correspond to those indicated in Fig. 1, and slope currents are marked in bold. The third column indicates previous studies that find seasonality in agreement with our results.

| Current | Time of seasonal maximum | Other studies |
| --- | --- | --- |
| **WSC (and NwASC)** | November to February | Beszczynska-Möller et al. (2012), von Appen et al. (2016) |
| **BSB** | November to February | Schauer et al. (2002) |
| VSC | November to December | Janout et al. (2015) |
| **ABC** | October to January (western Nansen Basin) | Pérez Hernández et al. (2019) |
| | September to December (Laptev Sea continental slope) | Baumann et al. (2018) |
| BG | October to January (southern branch) | Proshutinsky et al. (2009), Armitage et al. (2017) |
| **CSC** | August to October | Min et al. (2019) |
| PW | June to August (central–eastern Chukchi Sea) | Woodgate et al. (2005) |
| **EGC** | February to April | Bacon et al. (2014), Le Bras et al. (2018), de Steur et al. (2018) |

sub-monthly variability and in the interpolation method, including a different spatial decorrelation scale.

The comparison of altimetry-derived monthly fields with in situ data shows that agreement between these two data sources can be expected at scales exceeding roughly 50 km, both for sea surface height and surface circulation patterns. Altimetry-derived temporal variability in sea surface height shows agreement with mooring data at seasonal and longer timescales, while differences persist at monthly timescales. The agreement between velocities varies depending on the underlying nature and scale of the variability, showing the highest correlation in regions where a stable flow (e.g., boundary currents) dominates the mesoscale eddy activity. For instance, within boundary currents the pointwise correlation coefficient between altimetry and moored velocity is highest close to the shelf break, both in the Fram Strait (0.54) and at the Laptev Sea continental slope (0.77). Furthermore, our results show that seasonal flow variability is also resolved in the ocean interior, away from boundary currents. In the western Arctic, correlation is relatively high, both within the strong Pacific Water inflow in the Chukchi Sea (0.69) and at the moorings in the Beaufort Sea (0.53), although it is lower in the eddy-rich part of the basin.

Lastly, large-scale patterns emerge from a preliminary analysis of the seasonality: $\eta'$ exhibits a basin-wide coherent seasonal cycle, with a maximum between September and December and higher amplitude on the shelves; the $(u_g, v_g)$ features an intensification of the Arctic slope currents in winter and a weakening in summer. The agreement of these features with previous in situ based studies points to the important role that altimetry has in the Arctic Ocean, integrating individual mooring-inferred results into a basin-wide perspective.

ity has in the past been corrected using the standard tidal model FES2004 or equally performing models (e.g., Pujol et al., 2016; Mizobata et al., 2016; Armitage et al., 2016; Müller et al., 2019). Recent works (e.g., Rose et al., 2019; Prandi et al., 2021) have instead used new model versions with improved performance (Cancet et al., 2018) such as the FES2014 model (Lyard et al., 2021). In order to support our choice to use FES2014 over FES2004, we compared their performance by evaluating the difference in residual noise on the monthly maps due to unresolved sub-monthly variability (computed as in Sect. 4.3.2).

We display here in Fig. A1 the sub-monthly contribution to the standard error in two areas of high tidal amplitude, namely, the Barents Sea and the Baffin Bay. We note that, in both regions, FES2014 reduces the standard error of values by up to 0.3–0.5 cm with respect to FES2004 (Fig. A1c and f), which is about 20 % of its local value and 30 %–50 % of the average value over the whole Arctic. In agreement with these results, findings from Cancet et al. (2018), who compared the performances of several tidal models in the Arctic, show that differences in tidal amplitude and phase with respect to tide gauge data are much lower for FES2014 than for FES2004.

## Appendix A: Ocean tide correction

Following the European Space Agency indications (European Space Agency, 2016; Lyard et al., 2006), tidal variabil-

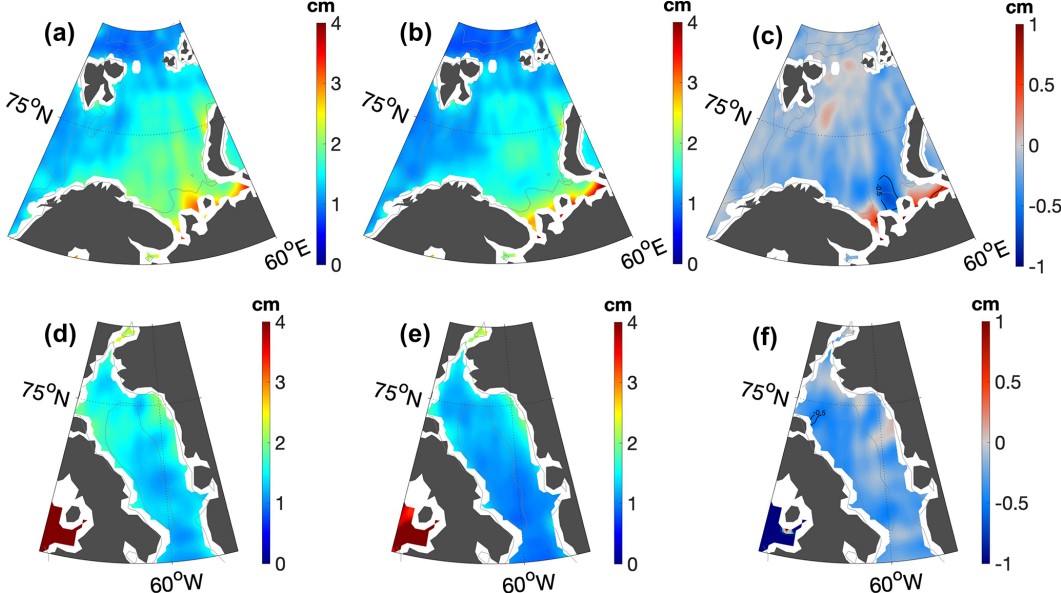

**Figure A1.** Comparison of performances of FES2004- and FES2014-based corrections. Sub-monthly contribution to the standard error on monthly $\eta'$ maps in the Barents Sea (**a–c**) and Baffin Bay (**d–f**) when using the tidal corrections FES2004 (**a, d**) and FES2014 (**b, e**). In panels (**c**) and (**f**) is shown the reduction in the error obtained with FES2014 with respect to FES2004.

## Appendix B: Dynamic atmospheric correction

DAC corrects the local and dynamic ocean response (waves) to pressure and wind changes and is derived from the sea surface height output of a barotropic model (Carrère and Lyard, 2003; Carrère et al., 2016). Up until the early 2000s, the effect of atmospheric pressure and winds on sea surface height had instead been corrected using an IB (e.g., Ponte and Gaspar, 1999; Carrère and Lyard, 2003). In the IB assumption, the sea surface height responds locally to changes in pressure, decreasing by approximately 1 cm for each increase in pressure of 1 mbar (atmospheric loading). Even though it has been shown that the IB is not always a good approximation of the ocean response, especially on timescales shorter than 20 d (Carrère and Lyard, 2003), little is known about the response in ice-covered regions (Robbins et al., 2016). Furthermore, recent results by Piecuch et al. (2022) suggest that deviations of the ocean response from a simple IB are particularly enhanced in the Arctic shelf regions with respect to the global average.

To establish whether DAC should also be used in ice-covered regions, we compared the reduction in altimetry standard deviation obtained by applying DAC with respect to IB in ice-covered regions of the Arctic Ocean. Figure B1a shows the binned difference in standard deviation applying the two corrections, where positive values indicate better performance of DAC over IB. DAC outperforms the IB in shallow shelf regions, and the two corrections perform equally well over the deep basins.

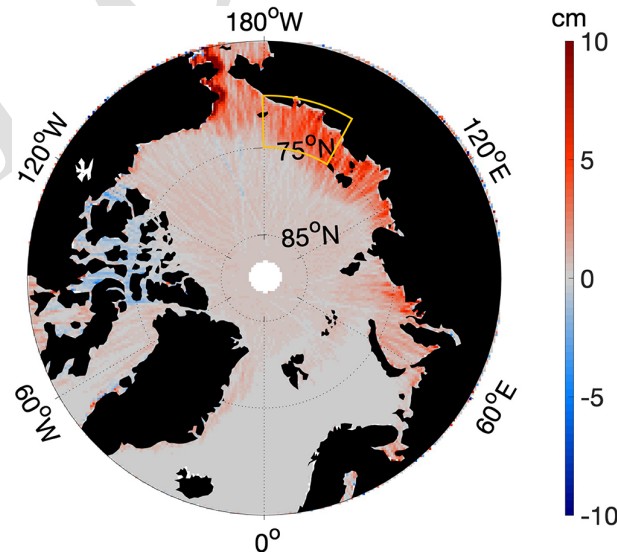

**Figure B1.** The along-track improvement in DAC correction, with respect to IB, in removing $\eta'$ high-frequency variability. Colors indicate the difference between the standard deviation of along-track $\eta'$ corrected with IB and corrected with DAC. The yellow square indicates the region of the East Siberian Sea where the frequency analysis was performed.

To understand which frequency bands have mostly contributed to this improvement, we took as an example the East Siberian Sea (yellow square indicated in Fig. B1a). We generated three time series of uncorrected $\eta'$, $\eta'$ corrected by IB and $\eta'$ corrected by DAC, averaged with a time step of 1 d

**Table B1.** Standard deviations of the three time series of along-track $\eta'$, averaged over the East Siberian Sea box (Fig. B1), using uncorrected $\eta'$, $\eta'$ corrected by IB and $\eta'$ corrected by DAC. For each year only ice-covered data are used, in the months November to July. Standard deviations are presented for the time series filtered in three different frequency bands.

| Standard deviation (cm) (uncorrected/IB/DAC) | $T > 20\,\mathrm{d}$ | $20\,\mathrm{d} > T > 5\,\mathrm{d}$ | $T < 5\,\mathrm{d}$ |
|---|---|---|---|
| 2011–2012 | 16.2/14.3/13.3 | 9.3/9.2/5.8 | 3.1/3.4/2.2 |
| 2012–2013 | 14.7/10.8/9.7 | 8.9/9.7/4.8 | 3.2/3.7/2.2 |
| 2013–2014 | 12.0/12.5/9.9 | 8.5/9.1/4.0 | 3.2/3.6/2.4 |
| 2014–2015 | 7.3/8.0/7.7 | 9.3/9.9/4.5 | 2.4/2.9/1.9 |
| 2015–2016 | 19.3/15.7/15.7 | 7.3/7.8/3.6 | 3.0/3.6/2.2 |
| 2016–2017 | 15.3/13.5/13.1 | 8.8/9.7/4.4 | 3.2/4.0/2.3 |
| 2017–2018 | 10.0/7.4/6.8 | 9.2/11.0/4.8 | 3.4/3.8/2.5 |

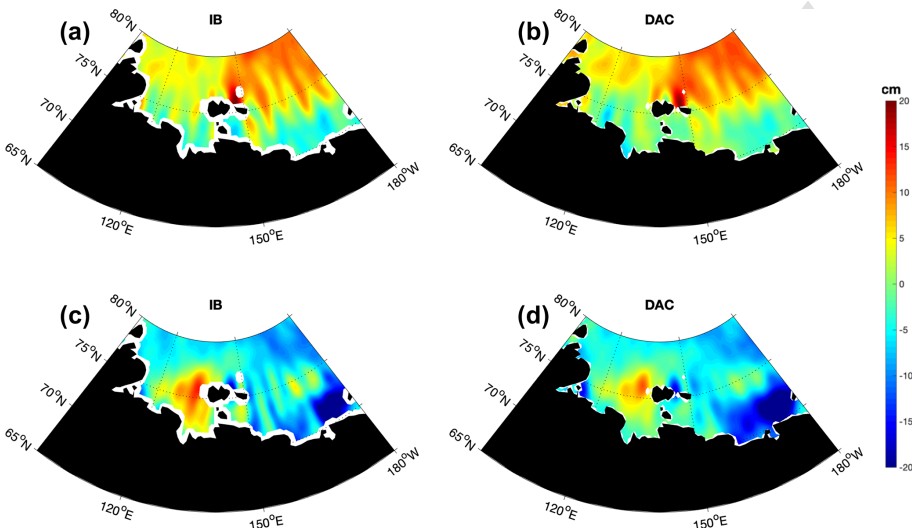

**Figure B2.** Effect of using correction DAC **(b, d)** instead of IB **(a, c)** on the monthly gridded $\eta'$ fields (see Sect. 4.3). Two examples are shown for the months of November 2014 **(a, b)** and November 2017 **(c, d)**.

over the indicated region. For each year we analyzed periods between November and July, which are the only months when data from leads are available. For each time series, we computed the standard deviation in frequency bands with periods $T > 20\,\mathrm{d}$, $5\,\mathrm{d} < T < 20\,\mathrm{d}$ and $T < 5\,\mathrm{d}$ (Table B1). Results show that DAC reduced the uncorrected $\eta'$ standard deviation by 50 % at periods shorter than 20 d, in contrast to no reduction when applying a simple IB.

Furthermore, standard deviation at periods between 20 and 5 d is larger than 60 % of the standard deviation at periods longer than 20 d, confirming that high-frequency variability represents a high portion of the total variability in the Arctic Ocean. The improvement in DAC with respect to IB over the shelves also appears in the $\eta'$ monthly grids, where meridionally oriented patterns of $\eta'$ are evidently reduced (two examples are given for the months of November 2014 and November 2017 in Fig. B2).

## Appendix C: Aliasing of residual sub-monthly variability

As stated in the main text, we performed the interpolation on weekly data subsets of observations of $\eta'$. Monthly maps were then obtained as the average of four weekly maps. The reasoning behind our approach is based on the fact that sea surface height in the Arctic exhibits large-scale, high-frequency (sub-monthly) variability, associated in part with the fast propagation of large-scale barotropic waves across the Arctic (Peralta-Ferriz et al., 2011; Fukumori et al., 2015; Danielson et al., 2020). This means that the variability is spatially coherent over hundreds of kilometers, yet it decorrelates quickly over time (e.g., weeks). Thus, measurements taken along tracks that are far away from each other yet within a few days of each other may still be able to resolve to some extent the spatial–temporal characteristics of the ocean variability. Instead, measurements taken along tracks that are close to each other yet taken 2 weeks apart from each other

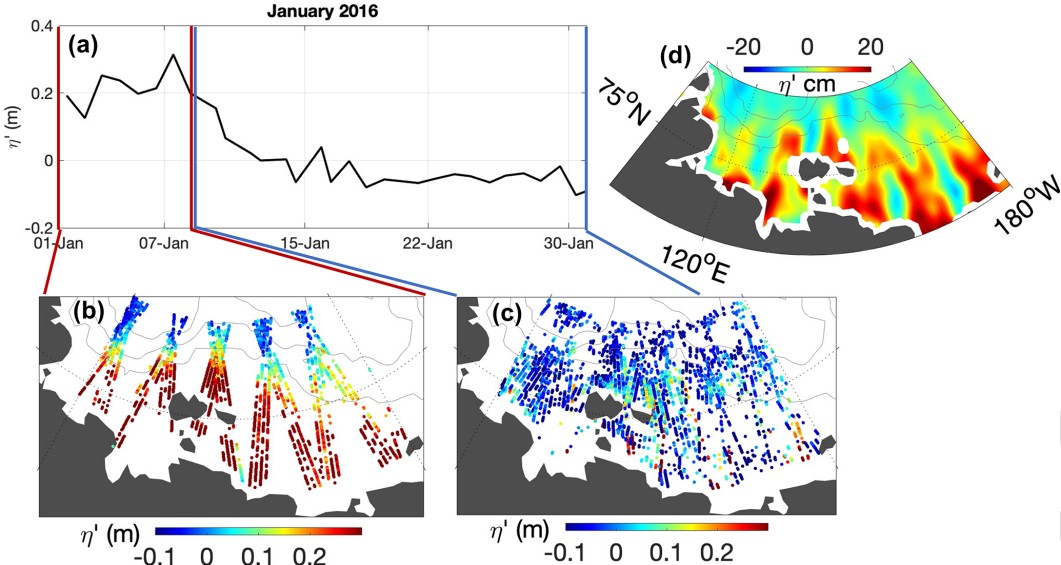

**Figure C1.** Trackiness introduced by sub-monthly variability. **(a)** Time series of average along-track $\eta'$ in the East Siberian Sea and Laptev Sea for the month of January 2016. **(b, c)** Scattered along-track $\eta'$ in the periods of **(b)** 1–8 January and **(c)** 9–31 January. **(d)** $\eta'$ field for the month of January 2016 if interpolation is performed on a monthly set of observations.

will create stripes (strong spatial sea surface height gradients) by not resolving the temporal variability. Since CryoSat-2 samples close-by regions at times separated by a large gap over the course of a month, trackiness will occur.

Therefore, constructing monthly maps based on sampling this large-scale, high-frequency variability at different times in different locations will artificially produce short wavelength patterns. We demonstrate this effect exemplarily in Fig. C1. One can clearly see how the sudden change in the large-scale sea surface height between the first and following weeks produces artificial stripes in the map when the monthly subset of data is interpolated.

**Author contributions.** FD processed the along-track data, performed the interpolation and the comparisons, and wrote most of the text. RR provided the along-track observations in the ice-covered regions and supported the processing of those. BR contributed to the processing of part of the in situ data and to the discussion and application of the interpolation method. AB and CT contributed to the application and description of the DIVA method. TK supervised the work and advised on the comparison with in situ data. All the authors contributed to the discussion of the results and to the improvement of the manuscript.

**Competing interests.** The contact author has declared that none of the authors has any competing interests.

**Disclaimer.** Publisher's note: Copernicus Publications remains neutral with regard to jurisdictional claims in published maps and institutional affiliations.

**Acknowledgements.** The processing of the CryoSat-2 sea surface height in ice-covered regions was funded by the German Ministry of Economic Affairs and Energy (grant no. 50EE1008); data from 2011 to 2018 were obtained from https://www.meereisportal. de (last access: TS14 , grant no. REKLIM-2013-04). In situ temperature and salinity data, ocean bottom pressure records and velocity data were collected from different sources. Data in the Eurasian Arctic are available in the framework of the Helmholtz Society's strategic investment FRontiers in Arctic marine Monitoring (FRAM) as well as the Nansen and Amundsen Basins Observations System II program (NABOS-II, NSF grant nos. AON-1203473, AON-1338948 and 1708427) and the joint Russian–German research project Changing Arctic Transpolar System (CATS). We are thankful to these projects for making publicly available quality-controlled in situ data. We thank expressly Vladimir Ivanov for leading the research expeditions of the CATS and NABOS-II projects and making the data collection possible. The data in the Beaufort Sea were collected and made available by the Beaufort Gyre Exploration Program based at the Woods Hole Oceanographic Institution (https://www2.whoi.edu/site/beaufortgyre/, last access: TS15 ) in collaboration with researchers from Fisheries and Oceans Canada at the Institute of Ocean Sciences. Data in the Chukchi Sea are available through the NOAA National Centers for Environmental Information (NCEI Accession 0164964). The work of Francesca Doglioni, Torsten Kanzow and Benjamin Rabe was part of the cooperative project REgional Atlantic Circulation and global changE (RACE) funded by the German Federal Ministry for Education and Research (BMBF), grant no. 03F0824E. The work of Francesca Doglioni is a contribution to the Helmholtz Climate Ini-

tiative REKLIM, a joint research project by the Helmholtz Association of German Research Centers (HGF). The work of Benjamin Rabe further contributed to the project Advective Pathways of nutrients and key Ecological substances in the Arctic (APEAR) (NE/R012865/1, NE/R012865/2, 03V01461), part of the Changing Arctic Ocean program jointly funded by the UKRI Natural Environment Research Council (NERC) and the BMBF. We were also funded by the Deutsche Forschungsgemeinschaft (DFG, German Research Foundation) through the Transregional Collaborative Research Centre TRR-172 "ArctiC Amplification: Climate Relevant Atmospheric and SurfaCe Processes, and Feedback Mechanisms (AC)3" (grant no. 268020496). We thank James Morison and five TS16 anonymous reviewers for their constructive comments that greatly helped to improve the manuscript and the associated datasets.

**Financial support.** This research has been supported by the NAME OF FUNDER (grant no. GRANT AGREEMENT NO). TS17

**Review statement.** This paper was edited by Giuseppe M. R. Manzella and reviewed by four anonymous referees.

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

## Remarks from the language copy-editor

CE1    Please note and confirm the addition of the cities to the affiliations above.
CE2    Should this be "db"?
CE3    US English spelling
CE4    "section(s)"?

## Remarks from the typesetter

TS1    The composition of Figs. 3, 5, 8, 9, 11 and 12 has been adjusted to our standards.
TS2    Please provide a shorter running title.
TS3    Thank you for sending the updated Figs. 1 and 6. We contacted the handling editor for approval and will replace the figures as soon as we have the editor's consent.
TS4    Please provide date of last access.
TS5    Please provide date of last access.
TS6    Please provide date of last access.
TS7    Please provide the full first name.
TS8    Please provide date of last access.
TS9    Please provide date of last access.
TS10    Please provide date of last access.
TS11    Please provide date of last access.
TS12    Please provide date of last access.
TS13    Please check throughout the text that all vectors are denoted by bold italics and matrices by bold roman.
TS14    Please provide date of last access.
TS15    Please provide date of last access.
TS16    Please check. Only four reviewers officially reviewed this article. Should "five" be changed to "four"?
TS17    Please note that there is funding information given in the acknowledgements, but you did not indicate any funding upon manuscript registration. Therefore, we were not able to complete the financial support statement. Please provide the missing information and double-check your acknowledgements to see whether repeated information can be removed from the acknowledgements. Thanks.
TS18    Please ensure that any data sets and software codes used in this work are properly cited in the text and included in this reference list. Thereby, please keep our reference style in mind, including creators, titles, publisher/repository, persistent identifier, and publication year. Regarding the publisher/repository, please add "[data set]" or "[code]" to the entry (e.g. Zenodo [code]).
TS19    Please provide the page range or article number.
TS20    Please provide date of last access.
TS21    Please provide the volume and page range/article number.
TS22    Please provide date of last access.
TS23    Please provide the page range or article number.
TS24    Please provide the volume and page range/article number.
TS25    Please provide the page range or article number.
TS26    Please provide the editors (if not authors), the publisher and a persistent identifier.
TS27    Please provide date of last access.
TS28    Please provide date of last access.
TS29    Please provide the page range or article number.
TS30    Please provide the page range or article number.
TS31    Please add [code] or [data set].
TS32    Please add name of repository and [code] or [data set].
TS33    Please add [code] or [data set].
TS34    Please provide date of last access.
TS35    Please provide date of last access.
TS36    Please provide the date and location of the symposium and a persistent identifier.
TS37    Please provide the editors (if not authors) and a persistent identifier.
TS38    Please provide the volume and page range/article number.
TS39    Please add [code] or [data set].

TS40    Please add [code] or [data set].
TS41    Please add [code] or [data set].