# Peer review of "Sea surface height anomaly and geostrophic current velocity from altimetry measurements over the Arctic Ocean (2011-2020)"

_Earth System Science Data, 2022_

## Author Comment (AC1)

**Authors' replies to
"Comment on essd-2022-111
Anonymous Referee #1"**

*Referee comment on "Sea surface height anomaly and geostrophic current velocity from altimetry measurements over the Arctic Ocean (2011–2020)" by Francesca Doglioni et al., Earth Syst. Sci. Data Discuss., https://doi.org/10.5194/essd-2022-111-RC1, 2022*
* * *
*The authors constructed a monthly mean sea level and dynamic ocean topography dataset using data from CryoSat-2 observations. The main differences from previous studies (but not all) are that the data construction focused on intra-month variability, the handling of data bias in the sea ice region and open water, and the validation using time series data from mooring system observations.*
*Correlation coefficients and RMSDs are calculated using the CPOM DOT as a comparison, but the differences with the CPOM DOT do not lead to the conclusion that the authors' product is truly better. In particular, the sensors in the mooring system that acquire insitu data are discrete in the vertical direction, and the steric height based on the linearly interpolated vertical profile is questionable. Also, the baroclinic Rossby radius is shorter than the L=300 km used in the first step of DIVA, which may be too much smoothing. Other comments are presented below.*

We are grateful to the reviewer for the constructive comments, which helped to improve the manuscript and the validation of our gridded product. Please find here few general remarks on the revision, and further below our elaboration on each comment.

In order to provide more evidence on the quality of the altimeter data used as source data and of our final gridded product, we added two pieces of analysis. First, we gave an overview on the statistics for the AWI and RADS along-track datasets and the merged dataset, furthermore including the latter in the file deposited by PANGAEA. Then, in order to extend the assessment of our gridded product to the western Arctic, we included comparisons with data from the Beaufort Gyre Observing System moorings, part of the Beaufort Gyre Exploration Program (BGEP[1], hereafter used as acronym for the moorings) and from two moorings in the central and eastern Chukchi Sea. We explored the possibility to compare to further data in the Chukchi Sea and Bering Strait but these time series were either too close to the coast to find good comparison points in the altimetry maps, or too short to have significant comparison, or very difficult to get at all.

We also revised our processing of in-situ data to evaluate possible errors deriving from it. We evaluated the reliability of our sea surface height from in-situ data in Fram Strait by comparing these data to continuous hydrographic profiles (for the steric height) and to a nearby bottom pressure recorder (for the bottom pressure equivalent height). We came to the conclusion that most of the variability is resolved, therefore we did not exclude these data from the validation. Detailed documentation is given below. We also found, though, a mistake in the computation of monthly means of in-situ data. When this was corrected, correlations between altimetry and in-situ sea surface height improved, with the greatest

improvement in the Fram Strait. Finally, we adjusted the averaging depth of in-situ velocity data in order to exclude the surface Ekman layer, with negligible impact on the results.

We clarify in our replies to your comments (10) and (12) what determines the actual resolution of our dataset. We explain that this is not related to the 300 km radius used in the first gridding step with DIVA, but to the 50 km radius used in the second and final step.

Regarding the comparison with the CPOM DOT, from our point of view, as discussed in our detailed replies, altimetry products for the Arctic Ocean are currently still at the stage of research. For this reason, it is rather useful to have a diversity of products, developed with different methodologies, that can be compared to understand how the quality of these datasets can still improve. We find this approach to be quite common, for instance, in the sea ice community for products of ice velocity, ice thickness, ice concentration.
* * *
*Major comments:*

*1) Mooring data1*

*The authors state that they are generating sea level data for the entire Arctic Ocean, but the only mooring system data used as validation are those installed in the Fram Strait and Laptev Sea. For example, WHOI has deployed BGOS mooring systems with MMP in the Canadian Basin. If time series data are important, then all available mooring system data should be used.*

Following your advice, we substantially extended our assessment of altimetry-derived sea surface height and geostrophic velocity fields to the western Arctic by comparing to additional mooring data in the Beaufort Sea and Chukchi Sea and hydrographic profiles in the Amerasian and Eurasian basins (see additions in sections 3, 5.2.1 and 6.2).

Regarding the assessment of sea surface height, first, we compared our sea surface height product to time series of the sum of steric height and bottom pressure equivalent height measured at the BGEP moorings A and D in the Canadian basin. Then, we compared our monthly fields to monthly estimates of the sum of steric height from in-situ hydrographic profiles, distributed in the deep basins, and bottom pressure equivalent height from GRACE. Geostrophic velocity was further compared to near-surface velocity from the BGEP moorings A, B and D and moorings S1 and S3 in the Chukchi Sea. In agreement with the comparisons previously done in the Eurasian Arctic, the additional comparisons show that, while there is reasonable agreement between our product and in-situ data at seasonal and longer time scales, significant differences are observed at monthly time scales.

We believe that these comparisons to data in western Arctic, in addition to the ones we conducted with mooring data at three locations in Fram Strait and the Eurasian Arctic (i.e., Arctic Cape and Laptev Sea; see Fig 9), provide an overview of the capability of our satellite product to reproduce in-situ measured sea level variability for a large portion of the Arctic.

Despite differences in the resolution and the nature of in-situ and remote sensing measurements, our gridded sea surface height significantly correlates with mooring data from the ice-covered Arctic and its boundaries, showing correlation coefficients ranging between 0.5 and 0.9. Furthermore, our work entails also the generation and assessment of geostrophic velocity, which now includes comparison with data from nineteen moorings and a discussion of its realism in the spatial and temporal patterns in regions of different dynamical regimes.

*2) Mooring data2*

*The authors calculate steric height using vertically discrete water temperature and salinity from mooring system observations. In the Arctic Ocean, where tilt pressure structures dominate, it is questionable whether the authors' method can correctly determine Steric height. Do the linearly interpolated vertical profiles of temperature and salinity reproduce the results of CTD observations made at the same time?*

We agree that linear interpolation of curved profiles will introduce a bias. However, given that the comparisons between mooring time series and altimetry were aimed at assessing the *temporal variability* of sea level, we tested here how the *variability* of the steric component is affected by the reconstruction of continuous vertical profiles with two different methods (linear and spline, see below). The results of this test showed that most of the variability is recovered using linear interpolation, and that we explained a higher portion of the variability by applying linear interpolation than splines. Following this analysis, we kept our method to compute steric height unchanged. We clarified in the manuscript in ll. 224-229 what is the uncertainty introduced by this approach.

We tested our approach using 432 continuous CTD profiles, deeper than 800 m, collected from the central Fram Strait over 16 years. Following the case of our comparison with mooring data in Fram Strait, we subsampled the profiles at depths of 50 m, 230 m and 720.

Fig R2 shows three series of steric height anomaly computed with different interpolation methods. We see there that the variability of steric height is still well represented by interpolated profiles. Linear interpolation performs better than spline interpolation, with 88% of explained variability. Even though we cannot assume that this result would be the same at all locations where we compared altimetry to in-situ data, the profiles in the Fram Strait were the most critical ones, with only 3 tie points. This suggests that vertical interpolation between discrete measurements can be applied with reasonable confidence as well at the Arctic Cape, in Laptev Sea and in the Beaufort Sea.

[Figure]

*Figure R 1: series of steric height anomaly computed from the 432 profiles using the continuous profiles (blue), linearly interpolated profiles (red, shifted by 0.2 m) and spline interpolated (yellow, shifted by 0.4 m) profiles.*

*3) Reference ellipsoid (line 82 "e.g., WGS84".)*

*From Figure 6d, I see that you used WGS84 instead of TP ellipsoid. Please specify somewhere that WGS84 was used in this study, not "e.g."*

Thank you for pointing that out. The description in section 2 is meant to be theoretical, therefore we changed now the sentence in ll. 82 to include the names of both ellipsoids. Regarding the DTU17MDT field used in this study, we added reference by personal communication with Per Knudsen (ll. 126-127), as we did not find written citable information on the reference ellipsoid used. However, when computing MDT it is usually ensured that the reference system for MSS and geoid is the same, so that the MDT difference will be in-sensitive to the choice. Furthermore, the difference between the T/P and the WGS84 ellipsoid over the Arctic Ocean consists basically of a spatially constant offset (Skourup et al. 2017), which has no impact on ocean dynamics.

*4) Sea ice concentration (line 107)*

*Isn't the 15% sea ice concentration a threshold to avoid so-called pseudo sea ice that misinterprets water vapor as sea ice, for example in the Bering Sea in summer, and doesn't it need to be 15% in the Arctic Ocean? Wouldn't the results be the same if this threshold were set to, say, 5%, 10%, or 20%? Isn't it usually the Waveform, for example Pulse Peakiness, that determines if it is sea ice or sea surface?*

We clarified in the text in ll. 112-114 why there is the need to choose a threshold between ice-covered and ice-free regions.

The 15% threshold is set for retracking of Cryosat-2 data over ice because below this ice concentration the ice is very sparse and the surface type classification (water, ice) is subject of high uncertainties in these areas (see Ricker et al. 2017, and Hendricks et al., 2021). The results would be very similar if a threshold of 20% was used, but in our case the 15% threshold fits well because it coincides with the uppermost limit in ice concentration set for the RADS dataset.

*5) ADCP (line 180)*

*ADCP velocity data is averaged in the upper 50 m. Shouldn't the velocities within the surface Ekman layer be excluded? Also, there must be a momentum flux due to sea ice movement, so shouldn't the surface still be excluded?*

Previous studies indicate that in the Arctic Ocean the Ekman layer extends approximately down to a depth of 20 m (e.g., Hunkins 1966; McPhee 1992; Cole et al. 2014; Peterson et al. 2017). In our previous manuscript version, we were already excluding data from the upper 10 m at the Laptev Sea continental slope. In the Fram Strait comparison, we used data from 75 m depth in order to avoid discontinuities in the timeseries when Current Meters were substituted with ADCPs in the later part of the time series. In order to account for the above comment, we adjusted the averaging depth at the Laptev Sea continental slope to the range 20-50 m, and included a comment about the averaging depth in ll. 200-201. The same averaging range was used for the ADCPs mounted on the BGEP moorings. This modification, however, did not lead any to significant changes in our results.

*6) High-pss filter(line 205)*

*Is there evidence that the high-pass filtered data is valid? If no, should it be excluded from the validation data?*

We did not exclude this timeseries from the validation data. The arguments presented here below to support this choice have been summarized in the manuscript at ll 234-238. While some variability is indeed missing in the in-situ data at this location, we can find strong evidence that the high pass filtered data is reliable (see below). We believe therefore that the in-situ timeseries at the mooring FS_S still includes a large fraction of the total variability. First, as we pointed out in our reply to your comment (2), the steric variability is still well represented at this location. Furthermore, Quinn and Ponte (2012) showed that the coherence between satellite observations of sea level and ocean bottom pressure is highest at timescales shorter than about 2 months, because of the presence of large wavelength – high frequency barotropic waves.

To test whether the high-pass filtered data is reliable, we considered the fact that high frequency variability in ocean bottom pressure decorrelates over large distances (e.g., Peralta-Ferriz et al., 2011). Therefore, we compared the time series at FS_S with the one from a bottom pressure sensor installed on another mooring 150 km away in the same basin (the bottom depth at the two moorings being 3012 m and 2778 m). After high-pass filtering the time series from the two moorings with various cut-off frequencies, we computed correlation and significance. The results are displayed in Table R1.

| cutoff | NO filtering | 4 months | 2 months | 1 month |
|---|---|---|---|---|
| **Correlation coefficient** | -0.54 | 0.13 | 0.57 | 0.73 |
| **p-value** | 0.88 | 0.16 | **<0.01** | **<0.01** |

*Table R 1: correlation between ocean bottom pressure at mooring FS_S and a nearby mooring (not used in manuscript) high-pass filtered with different cutoff frequencies.*

The correlation coefficient increases by filtering out long time scales and it is significant for thresholds below 4 months, with the best agreement on sub-monthly timescales. This gives us confidence that the high frequencies are not affected by instrumental noise and can be used to validate the month-to-month sea level variability due to mass contributions.

*7) Specific volume anomay*

*The reference depth the authors used is 400 dbar. Why? It is the depth of upper Atlantic Water. R. Kwok and other scientists used about 750 dbar (deepest depth of ITP).*

We thank the reviewer for pointing to this important issue. The main component of steric height changes on timescales up to a few years are in the so-called "Polar Mixed Layer" (PML, e.g. Korhonen et al., 2013), that part of the Arctic water column that may be seasonally mixed during ice formation and concurrent brine rejection. This layer is bounded underneath by the so-called "lower halocline", which is denoted approximately by the 34-isohaline (e.g. Korhonen et al., 2013). The lower halocline acts as a barrier between the PML and the deeper-lying warm Atlantic Water, which is largely isolated from surface influence in much of the Arctic basins. The lower halocline, and thus the Polar Mixed Layer, resides within the top 200 m across the Arctic. Hence, using the depth-range of 0-400 m should capture most of the steric height variability up to decadal timescales. This approach has been used in several studies, e.g. on Arctic freshwater (Rabe et al., 2011; 2014). Even though this range probably does not capture all multidecadal changes associated with changes in the Atlantic Water inflow, the dataset we're using to assess altimetry does not cover those timescales. Thus, a comparison between the steric height derived with this procedure and the altimetry data is appropriate.

Another reason to discard deeper data when studying variability is that much of the data over the time period under study here has been measured by WHOI-ITP (autonomous CTD profilers). These systems have to be quality-controlled by applying a conductivity correction. The only way to achieve that, for instruments that are usually not recovered, is to compare to historical data. The range used for that is 400-800 dbar (their profiles usually reach to a maximum of 760 dbar). Even though the historical reference data field is updated in time, it is not likely that variability deeper than about 400 m is captured by ITPs (see also Sumata et al., 2018). Furthermore, other studies assessing altimetry data used in the past steric height from hydrographic profiles integrated to a reference pressure shallower than the full ITP depth (760 db), e.g. 500 db, for instance Kwok et al. (2011), Morison et al. (2011), Armitage et al. (2016).

Following our reasoning above, we have modified the manuscript in ll. 265-267 to explain what component of the variability we expect to capture integrating over this depth range.

*8) Figure 2b*

*The INSET PANEL in Figure 2b is difficult to understand. How about color-coding the grid points by AWI and RADS?*

We changed Figure 2 by including two panels with color-coded grid points, green for AWI data and blue for RADS data.

*9) FIgure 4 inset panel*

*The color of the inset panel in Figure 4 indicates the number of crossovers, which basically increases as you go to higher latitudes since these are polar-orbiting satellites.*
*If this inset panel is independent and the color of each season indicates the difference of η at the crossover point, it is easy to understand where and in which season there is a difference.*

We are grateful for the suggestion but decided not to change the content of Figure 4. Even though we understand that there is an interest to know more about the error budget in the different seasons, our main aim in the inset of this figure is to show what the statistics for the error calculation is. To make this more explicit, we added one sentence in ll. 334-335 specifying the total number of crossovers used in this analysis. We will take however your comment in consideration for future investigation.

*10) Spatial resolution*

*Is the resolution setting just following the CPOM DOT, or if you want to differentiate yourself from CPOM, is there some strategy to change the resolution? At the moment, it is no different from CPOM except in the Siberian Sea.*

We consider that the dataset resolution is set by the data coverage and the decorrelation length used in the interpolation. In our case, the formulation of the DIVA gridding method is derived from a continuous equation, so the solution depends only on the decorrelation radius and the data density, regardless of the chosen grid, as long as this has a fine enough resolution. Even though the output grid for our dataset and the CPOM DOT has the same resolution, we used a radius of 50 km in the final gridding with DIVA while Armitage et al. (2016) smoothed the CPOM DOT with a larger radius of 100 km. This introduces a difference in the resolution between the two datasets. We additionally provide an analysis of the spatio-temporal resolution based on comparison with in situ data (see sections 5.2.2b and 6.3). From this analysis emerges, in summary, that our monthly geostrophic velocities can resolve seasonal to interannual variability of boundary currents wider than about 50 km.

Furthermore, our work differs from the what done in Armitage et al. (2016) in several aspects that go beyond the product resolution. We clarified these aspects in section 6.1 of the manuscript and used them to guide our discussion. A summary is provided in the list below:
- the source data (ellipsoidal heights from CryoSat-2) used in ice covered areas, which have been derived using different algorithms;

- the approach used to correct and/or minimise the impact of unresolved high frequency variability due to wind and tidal forcing and the estimate of its impact on the error;
- the interpolation method.

In addition, we extended the number of regions and dynamical regimes covered by the validation, which includes now both the eastern and western Arctic circulation regimes, the central Arctic Ocean, Arctic shelf seas and the main exchange gateways of the Arctic. This provides an improved analysis and validation of the realism in the spatial and temporal patterns of our dataset with respect to the work done by Armitage et al. 2016 and 2017.

Given that altimetric data products for the Arctic Ocean are currently still at the stage of open research, rather than development and improvement of a mature product, we think it is not appropriate to benchmark these products against each other. We rather think it is useful to compare them and find how they differ to understand potential uncertainties in the data. It is quite common in other areas of research, for instance in the sea ice community, to have a set of products (e.g., sea ice concentration or sea ice thickness), developed with different methodologies by different teams. Products can have different strengths also depending on the application. In future developments, methods and approaches from different products might find their way into a new, more mature DOT product.

*11) lines 358 & 360*

*I don't understand it because there is no detailed explanation of where the 4.2 cm and 8.2 cm came from.*

Thank you for pointing to this ambiguity. Regarding the 4.2 cm, this was a previous estimate of the error on the along-track data, which should have been updated in the current version to 3 cm (see ll. 339 of the revised manuscript); we corrected in this paragraph the number to 3 cm (ll. 403). The quantity 8.2 cm is instead our estimate of $\sigma$, which is then used to compute the signal to noise ratio used in DIVA (given by $\sigma^2/\varepsilon^2$). This estimate was derived by taking the data signal ($\sigma^2$) equal to the average spatial variance of weekly subsets of along-track data in the period 2011-2020. We agree that the formulation in the previous manuscript was not clear and rephrased this sentence to explain where this number comes from (ll. 404-405).

*12) Local gradients between 7W and 4W (line 462)*

*The authors used L=300 km, so they just applied too much smoothing. Why not take into account the bathymetry and reduce the value of L if there are steep velocity changes in the horizontal direction, for example?*

Regarding your first comment, we believe there is a misunderstanding about the meaning of the decorrelation scales used for the first and second interpolation steps, and would like to clarify that here below.

We used the decorrelation scale of L1=300 km to compute the background field (first step of interpolation). This step was necessary so that we could remove the mean field from the data and obtain anomalies relative to the period covered by the data. In the second step, where we apply the final gridding of the anomalies and obtain the final monthly fields, we used a decorrelation scale of L2=50 km. The scale used to compute the background field (L1) does not influence the smoothness of the final monthly fields, which is instead related to L2 and to the signal-to-noise ratio. The scales resolved in the dataset are further discussed in sections 5.2 and 6.3, where it results that the dataset does not resolve variability at scales shorter than L2=50 km (e.g. ll. 751-754). We added few sentences in the method section to clarify what above and furthermore highlight that a decorrelation scale of 50 km was chosen as the shortest radius possible based on the number of data points available to constrain the interpolation (ll. 389-394).

Regarding your second question, we agree that probably a spatially homogeneous decorrelation radius for the whole Arctic region is not the most accurate approximation. However, there is to date not enough knowledge to determine an appropriate alternative. We tested a built-in option in DIVA to apply anisotropic correlation length, where the anisotropy was based on a long-term average geostrophic velocity field from model output. This test however showed no significant change in the monthly fields. For this reason, and given that we do not know the accuracy of the model field used for anisotropy, we decided to exclude this feature in the final approach.

*13) Figure 9a*

*I guess relatively low correlation is due to incorrect steric height based on linear interpolation.*

In the previous version of the manuscript, we unfortunately made a mistake in computing the monthly averages of in situ SSH. This was actually in part the reason for the low correlations. As you can see in the updated manuscript, in Fig. 9a the correlation coefficient now reaches 0.7, which we do not consider as low. We agree on the fact that part of the variability is not captured by the interpolated in-situ data at this location, for the reasons discussed following your comments (2) an (3). We think, however, that data from this mooring still resolve most of the variability, one clear example being the steric height seasonal cycle. Therefore, as discussed under comment 2, we consider the interpolated mooring data a valid dataset in the context of our evaluation, given also the scarcity of data available in the Arctic Ocean.

*Minor comments:*

*Line 82 : e.g, --> e.g.,*

Changed

*Line 129: CPM --> CPOM*

Changed.

*Table 3: The table shows FES2014, but the caption describes it as FES2004.*

Changed.
* * *
References:

[1]Beaufort Gyre Exploration Program: https://www2.whoi.edu/site/beaufortgyre/

Armitage, T. W. K., S. Bacon, A. L. Ridout, S. F. Thomas, Y. Aksenov, and D. J. Wingham, 2016: Arctic sea surface height variability and change from satellite radar altimetry and GRACE, 2003-2014. *Journal of Geophysical Research: Oceans*, 121, 4303–4322, https://doi.org/10.1002/2015jc011579

Cole, S. T., M.-L. Timmermans, J. M. Toole, R. A. Krishfield, and F. T. Thwaites, 2014: Ekman Veering, Internal Waves, and Turbulence Observed under Arctic Sea Ice. *J Phys Oceanogr*, 44, 1306–1328, https://doi.org/10.1175/jpo-d-12-0191.1

Hunkins, K., 1966: Ekman drift currents in the Arctic Ocean. *Deep Sea Res Oceanogr Abstr*, 13, 607–620, https://doi.org/10.1016/0011-7471(66)90592-4

Korhonen, M., B. Rudels, M. Marnela, A. Wisotzki, and J. Zhao, 2013: Time and space variability of freshwater content, heat content and seasonal ice melt in the Arctic Ocean from 1991 to 2011. *Ocean Sci*, 9, 1015–1055, https://doi.org/10.5194/os-9-1015-2013

Kwok, R., and J. Morison, 2011: Dynamic topography of the ice-covered Arctic Ocean from ICESat. *Geophysical Research Letters*, 38, 1–6, https://doi.org/10.1029/2010gl046063

McPhee, M. G., 1992: Turbulent heat flux in the upper ocean under sea ice. *J Geophys Res Oceans*, 97, 5365–5379, https://doi.org/10.1029/92jc00239

Morison, J., R. Kwok, C. Peralta-Ferriz, M. Alkire, I. Rigor, R. Andersen, and M. Steele, 2011: Changing Arctic Ocean freshwater pathways. *Nature*, 481, 66–70, https://doi.org/10.1038/nature10705

Peterson, A. K., I. Fer, M. G. McPhee, and A. Randelhoff, 2017: Turbulent heat and momentum fluxes in the upper ocean under Arctic sea ice. *J Geophys Res Oceans*, 122, 1439–1456, https://doi.org/10.1002/2016jc012283

Rabe, B., M. Karcher, U. Schauer, J. M. Toole, R. Krishfield, S. Pisarev, F. Kauker, and T. Kikuchi, 2011: An assessment of Arctic Ocean freshwater content changes from the 1990s to the 2006–2008 period. *Deep-Sea Research Part I*, 58, 173–185, https://doi.org/doi:10.1016/j.dsr.2010.12.002

Rabe, B., and Coauthors, 2014: Arctic Ocean basin liquid freshwater storage trend 1992–2012. *Geophys Res Lett*, 41, 961–968, https://doi.org/10.1002/2013gl058121

Ricker, R., S. Hendricks, L. Kaleschke, X. Tian-Kunze, J. King, and C. Haas, 2017: A weekly Arctic sea-ice thickness data record from merged CryoSat-2 and SMOS satellite data. *The Cryosphere Discussions*, 11, 1607–1623, https://doi.org/10.5194/tc-11-1607-2017

Skourup, H., and Coauthors, 2017: An Assessment of State-of-the-Art Mean Sea Surface and Geoid Models of the Arctic Ocean: Implications for Sea Ice Freeboard Retrieval. *Journal of Geophysical Research: Oceans*, 122, 8593–8613, https://doi.org/10.1002/2017jc013176.

Sumata, H., and Coauthors, 2018: Decorrelation scales for Arctic Ocean hydrography – Part I: Amerasian Basin. *Ocean Science*, 14, 161–185, https://doi.org/10.5194/os-14-161-2018

---

## Author Comment (AC2)

**Authors' replies to
"Comment on essd-2022-111
Anonymous Referee #2"**

*Referee comment on "Sea surface height anomaly and geostrophic current velocity from altimetry measurements over the Arctic Ocean (2011–2020)" by Francesca Doglioni et al., Earth Syst. Sci. Data Discuss., https://doi.org/10.5194/essd-2022-111-RC2, 2022*
* * *
*In this work, the authors assess a new Arctic-wide gridded dataset of sea surface height and geostrophic velocity at monthly resolution during the period 2011 to 2020. This dataset was generated using Cryosat-2 observations from two products (RADS and AWI). The authors describe how the gridded altimetry-derived variables are produced and show results from comparisons against available in situ measurements (moorings) and an independent state-of-the-art data (altimetry). The seasonal cycle emerging from the final monthly maps is finally discussed.*

*Overall the paper is certainly of interest to the Arctic community. There is a need to have a unified data set for the ice-covered region and for the ice-free region that is validated properly. The authors provide a description of RADS and AWI products as well as how the two products are made consistent and homogenous before gridding. The new gridded data set is validated, but only in the Fram Strait and Laptev Sea.*

We are grateful to the reviewer for the constructive comments, which helped to improve the manuscript and the validation of our gridded product. We thank you also for your positive statement on the value of the publication of our manuscript and dataset, once the validation is reinforced. Please find here few general remarks on the revision, and further below our elaboration on each comment.

In order to provide more evidence on the quality of the altimeter data used as source data and of our final gridded product, we added two pieces of analysis. First, we gave an overview on the statistics for the AWI and RADS along-track datasets and the merged dataset, furthermore including the latter in the file deposited by PANGAEA. Then, in order to extend the assessment of our gridded product to the western Arctic, we included comparisons with data from the Beaufort Gyre Observing System moorings, part of the Beaufort Gyre Exploration Program (BGEP[1], hereafter used as acronym for the moorings) and from two moorings in the central and eastern Chukchi Sea. We explored the possibility to compare to further data in the Chukchi Sea and Bering Strait but these time series were either too close to the coast to find good comparison points in the altimetry maps, or too short to have significant comparison, or very difficult to get at all.

We also revised our processing of in-situ data to evaluate possible errors deriving from it. We evaluated the reliability of our sea surface height from in-situ data in Fram Strait by comparing these data to continuous hydrographic profiles (for the steric height) and to a nearby bottom pressure recorder (for the bottom pressure equivalent height). We came to

the conclusion that most of the variability is resolved, therefore we did not exclude these data from the validation. Detailed documentation is given below. We also found, though, a mistake in the computation of monthly means of in-situ data. When this was corrected, correlations between altimetry and in-situ sea surface height improved, with the greatest improvement in the Fram Strait. Finally, we adjusted the averaging depth of in-situ velocity data in order to exclude the surface Ekman layer, with negligible impact on the results.
* * *
*1) Overall the paper is well written with a clear rationale. My feeling is that the processing and validation part is rather limited. Altimeter data retrieval in the ice-covered region is very challenging due to presence of ice that perturbs radar echoes. It is stated that AWI product takes care using a customized processing. The reader expects more convincing analyses of RADS and AWI before their merging, with some statistics about their quality in the whole Arctic area. Authors only show one profile at certain date (Figure 3) that cannot be representative of a ten year period.*

Within the ice-covered regions, we retrieve the sea level from leads (openings in the ice cover, cracks). Because of their flat surface compared to sea ice, the received radar echo power is dominated by the reflections from open water, even if it covers just a small area within the satellite footprint. Therefore, the perturbation by sea ice is rather limited (Ricker et al., 2014).

However, we agree that providing an overview on the statistics relative to the AWI and RADS datasets can help the reader to evaluate the quality of the final product. For this reason, we added panels in Fig 3 to visualise average 2011-2020 statistics, namely the spatial distribution of monthly standard deviation and number of observations for AWI and RADS (separated into summer and winter seasons) and for the merged dataset. These maps are commented in ll. 308-315. Furthermore, we included the merged along track dataset, as processed in this work, in the final data file deposited in PANGAEA (Doglioni et al., 2021)

*2) I have also a remark about the velocity geostrophic computation at the surface. It is well known altimetry data are noisy and the accuracy of the slope estimate from along-track SLA is strongly impacted. It might dominate the errors in estimating ocean currents because a simple finite-difference of the along-track SLA acts as a high-pass filter. This effect can be mitigated, see Liu, Y., Weisberg, R.H., Vignudelli, S., Roblou, L. and Merz, C.R., 2012. Comparison of the X-TRACK altimetry estimated currents with moored ADCP and HF radar observations on the West Florida Shelf. Advances in Space Research, 50(8), pp.1085-1098.*

We agree that in general the computation of ocean currents from along-track data is impacted by the noise in the observations. In our case, this was most true for the AWI data in the ice-covered regions because they were originally provided at high frequency sampling (20 Hz), not smoothed. For this reason, in our procedure we do smooth and subsample the AWI data along the satellite tracks before using them (see ll. 247-248). As an addition, we mention now in the text (ll. 248-250) that smoothing is beneficial to reduce noise in the computation of geostrophic velocity as well.

In our work however, we compute geostrophic velocity from the interpolated maps of sea surface height, which consist in a two-dimensional, smoothed reconstruction of the fields from along track data. In this case, the error in the ocean currents is rather dominated by residual sub-monthly variability between neighbouring tracks, that is aliased into spatial variability when mapping monthly data (see appendix C). This is addressed in our work by (1) using up-to-date corrections for tides and the response to wind and atmospheric pressure and (2) interpolating the along-track data weekly and averaging four subsequent weekly fields (see explanation in section 4.3.2).

*3) Also, the direct comparison of the altimeter-derived geostrophic current velocities with the mooring real current velocities does not account for a wind-driven Ekman velocity component. Why ? is the wind contribution supposed negligible ?*

Previous studies indicate that in the Arctic Ocean the Ekman layer extends approximately down to a depth of 20 m (e.g., Hunkins 1966; McPhee 1992; Cole et al. 2014; Peterson et al. 2017). In our previous manuscript version, we were already excluding data from the upper 10 m at the Laptev Sea continental slope. In the Fram Strait comparison, we used data from 75 m depth in order to avoid discontinuities in the timeseries when Current Meters were substituted with ADCPs in the later part of the time series. In order to account for the above comment, we adjusted the averaging depth at the Laptev Sea continental slope to the range 20-50 m, and included a comment about the averaging depth in ll. 200-2001. This change, however, did not lead any to significant changes in our results.

*4) Having said that, I think the paper deserves publication after the authors reinforce the statistics of the two data sources (AWI and RADS) that are used to generate the new data set. If possible I also recommend to extend the validation to other sites in order to provide the reader with a more complete picture about the accuracy of the altimeter-derived sea level and velocities against in situ measurements.*

Following your advice, we integrated Fig. 3 to provide an overview on the average monthly statistics of the AWI and RADS datasets and the merged dataset, and commented the results therein. This addition supported our confidence in the quality of the input data and the consistency between the AWI and RADS sub-datasets.

Furthermore, we substantially extended our assessment of altimetry-derived sea surface height and geostrophic velocity fields to the western Arctic by comparing to additional mooring data in the Beaufort Sea and Chukchi Sea and hydrographic profiles in the Amerasian and Eurasian basins (see additions in sections 3, 5.2.1 and 6.2).

Regarding the assessment of sea surface height, first, we compared our sea surface height product to time series of the sum of steric height and bottom pressure equivalent height measured at the BGEP moorings A and D in the Canadian basin. Then, we compared our monthly fields to monthly estimates of the sum of steric height from in-situ hydrographic profiles, distributed in the deep basins, and bottom pressure equivalent height from GRACE. Geostrophic velocity was further compared to near-surface velocity from the BGEP moorings A, B and D and moorings S1 and S3 in the Chukchi Sea. In agreement with the

comparisons previously done in the Eurasian Arctic, the additional comparisons show that, while there is reasonable agreement between our product and in-situ data at seasonal and longer time scales, significant differences are observed at monthly time scales.

We believe that these comparisons to data in western Arctic, in addition to the ones we conducted with mooring data at three locations in Fram Strait and the Eurasian Arctic (i.e., Arctic Cape and Laptev Sea; see Fig 9), provide an overview of the capability of our satellite product to reproduce in-situ measured variability for a large portion of the Arctic. Despite differences in the resolution and the nature of in-situ and remote sensing measurements, our gridded sea surface height significantly correlates with mooring data from the ice-covered Arctic and its boundaries, showing correlation coefficients ranging between 0.5 and 0.9. Furthermore, our work entails also the generation and assessment of geostrophic velocity, which now includes comparison with data from nineteen moorings and a discussion of its realism in the spatial and temporal patterns in regions of different dynamical regimes.
* * *
References:

[1]Beaufort Gyre Exploration Projgram: https://www2.whoi.edu/site/beaufortgyre/

Cole, S. T., M.-L. Timmermans, J. M. Toole, R. A. Krishfield, and F. T. Thwaites, 2014: Ekman Veering, Internal Waves, and Turbulence Observed under Arctic Sea Ice. *J Phys Oceanogr*, 44, 1306–1328, https://doi.org/10.1175/jpo-d-12-0191.1

Doglioni, F., Ricker, R., Rabe, B., Barth, A., Troupin, C., and Kanzow, T., 2021: Pan-Arctic monthly maps of sea surface height anomaly and geostrophic velocity from the satellite altimetry Cryosat-2 mission, 2011-2020, https://doi.pangaea.de/10.1594/PANGAEA.931869

Hunkins, K., 1966: Ekman drift currents in the Arctic Ocean. *Deep Sea Res Oceanogr Abstr*, 13, 607–620, https://doi.org/10.1016/0011-7471(66)90592-4

McPhee, M. G., 1992: Turbulent heat flux in the upper ocean under sea ice. *J Geophys Res Oceans*, 97, 5365–5379, https://doi.org/10.1029/92jc00239

Peterson, A. K., I. Fer, M. G. McPhee, and A. Randelhoff, 2017: Turbulent heat and momentum fluxes in the upper ocean under Arctic sea ice. *J Geophys Res Oceans*, 122, 1439–1456, https://doi.org/10.1002/2016jc012283

Ricker, R., S. Hendricks, V. Helm, and M. Davidson, 2014: Sensitivity of CryoSat-2 Arctic sea-ice freeboard and thickness on radar-waveform interpretation. *The Cryosphere*, **8**, 1607–1622, https://doi.org/10.5194/tc-8-1607-2014

---

## Author Comment (AC3)

**Authors' replies to**
**Editor Comment on essd-2022-111**

*Editor comment on "Sea surface height anomaly and geostrophic current velocity from altimetry measurements over the Arctic Ocean (2011–2020)" by Francesca Doglioni et al., Earth Syst. Sci. Data Discuss., https://doi.org/10.5194/essd-2022-111-RC2, 2022*
* * *
*The paper can only be accepted after a major revision of the entire content. In particular I would like to emphasize the fact that satellite data must be validated by in situ data. As noted by one referee, there is only one point of verification: too little to ensure the validity of the results over the entire area under study. The methodology must be better explained and the advantages emphasized more precisely. Last thing: the spatial resolution. Considering the criticisms of the referees. Authors are allowed to review their manuscript which will undergo further re-evaluation*

We thank the editor for the opportunity to revise our work, which allowed us to improve the clarity of the manuscript itself and substantially extend the dataset validation. We have responded to all of the criticisms highlighted above. In the following, we give a brief overview over the modifications we implemented accordingly. Further details can be found in our replies to the referees.

**1. Improved information on data quality**

The main concern expressed by the two referees, and supported by your above comment, regards the data quality and validation. Specifically, the referees asked to include more information about the quality of the source data used to generate the maps, and to extend the validation of the final monthly maps.

To provide more information on the quality of the source data (AWI and RADS), we acted in two directions:

Modification 1.1: We provided in Fig 3 an overview of the AWI and RADS data density (number of data points per 100 km$^2$) and statistics accompanying the source data, which was commented in section 4.2.3;

Modification 1.2: We added the merged ice-covered and open-ocean along-track data, as processed in this work, to the final data file deposited in the data archive PANGAEA.

In this way we now provide clearer information on the quality of the source data in a 2-stage way, particularly addressing the comment by Referee#2 who mentioned that only one single satellite profile was shown. The manuscript is thus much improved in terms of transparency regarding the quality of the source data. Readers, who want a general overview of the source data characteristics will find this addressed by modification 1.1. Readers who prefer to carry out their own in-depth assessment, will be able to do this building on modification 1.2.

**2. Validation**

To improve the validation of the final monthly sea level maps, we extended the quality assessment of our gridded sea surface height and geostrophic velocity fields as follows.

Modification 2.1: the validation now includes mooring data in the Beaufort gyre (deep basin in the western Arctic). Sea surface height was compared to data from moorings part of the Beaufort Gyre Exploration Program (BGEP[1]). Geostrophic velocity was compared to velocity data from the BGEP moorings.

Modification 2.2: the validation now includes mooring data in Chukchi Sea (shelf region in the western Arctic). Geostrophic velocity was compared to velocity data from two moorings in the Chukchi Sea.

Modification 2.3: Validation of sea surface heigh fields now include basin scale, monthly comparisons to hydrographic profiles from the central Arctic basins.

With these modifications our validation includes both the eastern and western Arctic circulation regimes, the central Arctic Ocean, Arctic shelf seas and the main exchange gateways of the Arctic. The extension of the validation is a huge effort, and we argue, that all major aspects of the Arctic Ocean circulation are included now. Overall, the extended validation is consistent with our previous results, showing that the correlation with in-situ data is significant where variability on seasonal and longer time scales is present, while it is reduced in presence of intense eddy activity.

**3. Better explanation and advantages of methodology, including the dataset resolution**

The methodology used to process the in-situ data was revised and improved, furthermore differences / advantages of our methodology to previous approaches were pointed out, following comments from both referees.

Modification 3.1: We checked the processing of in-situ data for possible errors or approximations, which could lower the correlation between in-situ and altimetry data (see especially our replies to comments from Referee#1). This revision was indeed helpful to support the reliability of the in-situ sea surface height estimates. After the revision, the correlation coefficients between altimetry- and in-situ- sea surface height actually improved at all comparison sites, now ranging from 0.5 to 0.9.

Modification 3.2:
In our manuscript discussion we described more explicitly the differences and advantages of our work with respect to what done in Armitage et al. (2016) to derive to CPOM DOT: in section 6.1, we clarify the methodological steps in which the processing of our dataset differs from the CPOM DOT; in section 6.3, we discuss the spatial and temporal resolution of our dataset. The advantages of our work are briefly summarized in the following.
- We used an active approach to filter out unresolved high frequency ocean variability from the along-track satellite data: *i)* we demonstrated the improvement in the

correction provided by more recent models for wind- and tide-related variability (i.e., DAC versus IB and FES2014 versus FES2004), and finally applied those; *ii)* we derived the monthly maps as averages of the following four weekly interpolated maps. Through this two-steps approach, we also provided: an overview of Arctic sea-level-variability at sub-monthly timescales; an estimate of the contribution of this variability to the error on the monthly means.

- We used an interpolation radius of 50 km to grid our SLA fields, shorter than the smoothing radius of 100 km used for the CPOM DOT. Given that the solution provided by the DIVA gridding method is derived from a continuous equation, this scale is the one that impacts the effective resolution of the dataset, independently from the resolution of the output grid. Therefore, the resolution is improved with respect to the CPOM DOT.

- The analysis and validation of the realism in the spatial and temporal patterns of our dataset is extended with respect to the work done by Armitage et al. 2016 and 2017. This improvement consists of:

    1. Extended number of regions and dynamical regimes covered by the validation: sea surface height is compared to data from 5 moorings in the Fram Strait, Eurasian Basin and Beaufort Sea and more than 3000 hydrographic profiles distributed across the Amerasian basin, the Eurasian Basin and in the Fram Strait; geostrophic velocity is validated based on data from a total of 19 moorings from the Fram Strait, Eurasian Arctic, Beaufort Gyre and Chukchi Sea.

    2. More in-depth analysis of spatial and temporal patterns in velocity fields (see results in section 5.2.2b and discussion in section 6.3):

        ▪ Regarding the spatial resolution, our results show that the geostrophic velocity can capture transitions from strong to weak mean flow on scales roughly exceeding 50 km, which is consistent with the underlying smoothing of the altimeter data using a 50 km scale;
        ▪ Regarding the temporal resolution, we find that altimetry and in-situ velocity agree best in regions where the flow is dominated by steady currents, with dominant variability at seasonal and longer time scales. Within these regions, correlation coefficients are highest when data are averaged over a cross-flow distance of about 50 km.

        Consistently with the resolution set for our dataset in phase of interpolation, as a result of our validation we show to be able to resolve temporal variability of flow speed of the West Spitzbergen Current when spatially averaged over a 50 km, a narrow but important pathway for Atlantic Water into the Arctic. On the contrary, it is mentioned among the results of Armitage et al. (2017) that the smoothing applied to the CPOM DOT prevents properly resolving this current.

3. Evaluation of currents seasonality by literature review: In sections 5.3 and 6.4 we provide an overview of the seasonality emerging from our dataset and how that compares to literature results. This is indeed an added value to the direct comparison, as it shows the consistency of our results with previously published results, and highlights the usefulness of the dataset for studies of large-scale Arctic Ocean circulation.

**References**:

[1]Beaufort Gyre Exploration Program: https://www2.whoi.edu/site/beaufortgyre/

Armitage, T. W. K., S. Bacon, A. L. Ridout, S. F. Thomas, Y. Aksenov, and D. J. Wingham, 2016: Arctic sea surface height variability and change from satellite radar altimetry and GRACE, 2003-2014. *Journal of Geophysical Research: Oceans*, **121**, 4303–4322, https://doi.org/10.1002/2015jc011579

Armitage, T. W. K., S. Bacon, A. L. Ridout, A. A. Petty, S. Wolbach, and M. Tsamados, 2017: Arctic Ocean surface geostrophic circulation 2003–2014. *The Cryosphere*, **11**, 1767–1780, https://doi.org/10.5194/tc-11-1767-2017.

---

## Author Response (AR2)

*Reviews for "Sea surface height anomaly and geostrophic current velocity from altimetry measurements over the Arctic Ocean (2011-2020)" by Francesca Doglioni et al., Earth Syst. Sci. Data Discuss., https://doi.org/10.5194/essd-2022-111-RC1, 2022*

**Referee#1**
* * *
The authors created a new gridded, altimetry-based, monthly sea surface height and geostrophic velocity dataset in the Arctic Ocean over both sea-ice covered and sea-ice free regions. Compared with the existing gridded dataset of sea surface height and geostrophic velocity (Armitage et al., 2016, 2017), this dataset covers more recent period 2011-2020 from 2003-2014, and is processed with the newer tidal model FES2014 and different interpolation method to have better spatial resolution for boundary currents.

The manuscript is well written. The methods involved have been described in detail, and the validation is extensive. The dataset would be useful for many scientific communities studying the changes happening in the Arctic Ocean, especially the extension in time from 2003-2014 to 2011-2020. I recommend the paper accepted subject to minor revisions (review by editor)

Major comments:

Swapping Section 4.3.3 and Section 4.3.2 makes more sense to me. The 1-week decorrelation time scale (L438-439) presented in Section 4.3.3 provides a basis for how to estimate monthly error maps from weekly error maps. Therefore moving Section 4.3.3 before Section 4.3.2 makes more sense.
**We agree and swapped the two sections. Thank you for your suggestion.**

The manuscript does not mention ocean reanalysis products (e.g., ASTE by Nguyen, et al., 2021, doi:10.1029/2020MS002398) or recent modelling studies (e.g., Bacon et al., 2015, doi:10.1098/rsta.2014.0169). In particular, this new data product can be used for constraining new ocean reanalysis products. Comparing the new data product with some ocean reanalysis products in this manuscript may not be practically possible because of the space limit, but adding a few sentences about ocean reanalysis effort for the Arctic Ocean (e.g., in Introduction) and potential usefulness of the new altimetry data for the ocean reanalysis community (e.g., in Discussion) would be helpful.
**This is a very good point that we missed somehow to emphasise, thank you for suggesting reference literature. We included few sentences in this regard in the Introduction and Conclusions (as the Discussion was focused on the results from the dataset assessment).**

L302-308: Regarding the dynamic ocean response to air pressure forcing, a recently-published paper by Piecuch et al. (2022, JPO, https://doi.org/10.1175/JPO-D-22-0090.1) should be cited here and probably also in Appendix B. The East Siberian Sea where the improvement is large by using the DAC correction (Fig. B1) is also found by Piecuch et al. (2022).
**Thank you for pointing to this relevant literature. We included a citation to this paper both in this section and in appendix B.**

L385/L421: Monthly fields were calculated by averaging four weekly data (such as L385, L421). Did the authors do something extra (such as interpolation) to take care of possible misalignment of the average time of four weekly data and the center time of a calendar month?

*The weeks were defined based on the days of the month rather than the actual calendar week. Each month was divided in four parts as follows: day#1-day#8, day#9-day#15, day#16-day#23, day#24-LASTday. We believe that this has a negligible influence on our final results.*

L421-422: The sentence, especially "quadratic sum of four weekly error maps", needs to be reworded. The monthly error maps are probably square root of some averaging of the weekly error maps. It would also be nice to explicitly say something about the assumption made about how the weekly error maps are correlated. Are they independent, fully correlated, or something in between?

*Thank you for pointing that out the wrong wording, we meant to refer to the sum in quadrature, which is the square root of the sum of the squares. We modified this part of the sentence to "[…] the associated interpolation error was computed by adding in quadrature of four weekly error maps".*

*Before this sentence we also clarified that, as it results from the section about the error on the monthly fields, the weekly SLA fields are statistically independent, which allow us to treat the errors as independent and add them in quadrature. This will be in the worst case an estimate of the maximum error.*

Figure 1: Some of the texts in the figure are hard to read. Might better to change color/font to make them easy to read.

*We improved the readability of the above figure by changing colors and re-arranging text.*

Figure 6: Can the authors explain why there are ring-like patterns that are aligned with latitudinal circles?

*If we understand well, the reviewer refers to the pattern in Fig. 6b. This pattern emerges because the gaps left by the satellite orbit geometry in the weekly data distribution are larger in some latitude bands (visible in Fig. 3g, e.g., around 80° N or 67° N). Over the month these large gaps only shift in longitude but not in latitude. This makes so in those latitude bands one gets the largest error.*

Minor comments:

L44-45: Replace "methodological developments" with "methodologies" to avoid redundant "develop"(ments) in this sentence.
*Done.*

L73: Add (Sect. 4) after "section"
*Done.*

L80: Replace "to understand" with "to understanding"
*Done.*

L120: Remove "Remko"
*Done.*

L121: Replace "section" with "Sect."
*Done.*

L125: Replace "section" with "Sect." to be consistent throughout the manuscript
*Done.*

L136: GOCO3s should be GOCO03s
*Done.*

L138: Why not starting from January 2010?
*CryoSat-2 observations are available only starting November 2010, with the quality of the first few months being significantly lower that the rest of the mission. Therefore, our gridded product starts from January 2011.*

L155: Since L146-147 just talked about how steric height would be calculated, either remove "Steric height was computed following equation 7 (Sect. 4.2.1)" or change it to "Again, steric height was computed following Eq. 7 (Sect. 4.2.1)".
*Done.*

L161: AWI was defined above. so there is no need to define AWI again here. Just use AWI instead of Alfred Wegener Institute.
*Done.*

L169: Remove the comma after "the Arctic Data Center"
*Done.*

L172-173: Missing words in "monthly averages of the of the hydrographic profiles"
*Corrected.*

L186: Maybe "close", not "closer"?
*Changed to "closest".*

L196: moorings locations, instead of "moorings positions"
*Done.*

L213: Add what is rho' (e.g., density anomaly w.r.t. 1000 g/m3)
*In the introduction to this section, we specified that all prime variables in Eq. (5) and (6) refer to time anomalies. We removed the rho' in the sentence following Eq. (6) to avoid confusion, as this is already defined above.*

L216: Vertical density profiles
*Added.*

L223: avoided propagating
***Changed.***

L224-225: Not clear why linear interpolation of TS between discrete measurement levels might introduce "time-mean biases", especially why "time-mean biases"
***We agree that in general the bias introduced can be also a time-variable one. We rephrased this paragraph to clarify that we are only concerned with the accuracy in the temporal variability, reason why we tested the interpolation used as described in the text below this sentence (now at lines 225-229).***

L259: change (Fig. ??b) to (Fig. 1)
***Corrected.***

L346: Replace "interpolation of" with "interpolating"
***Done.***

L347: Replace "section 4.3.1" with "Sect. 4.3.1"
***Done.***

L347-349: Need to rewrite/reword the sentence "In Sect. 4.3.2 ... to obtain monthly fields", which is hard to read and understand.
***We rephrased the sentence, thank you.***

L352: Replace "Along track" with "Along-track"
***Done.***

L391: Replace "sections" with "Sects."
***Done.***

L392: Replace "Fig" with "Fig."
***Done.***

L402: Replace "sections" with "Sects."
***Done.***

L461-462: The two sentences are redundant. Remove one.
***Done.***

L530: Replace "section 4.2.1" with "Sect. 4.2.1"
***Done.***

L588: Take "into" account
***Corrected.***

L589-591: How is the weight defined, if averaging data sets over only two moorings? I believe the subscripts i & j-1 or i & j+1 means moorings i & j-1 or i & j+1. So either d (i,j-1) or d(j, j+1) is missing in the equation for the weight if there are only two moorings.

*Thanks for pointing this out, I slightly modified the text to make this point clearer. The "outermost" moorings in the average are weighted only based on the distance to the one nearby mooring (without dividing by 2). In this way, if only two moorings are used, they will have equal weight.*

L593: Use ";" instead of "," for "... was highest, compare ..."
*Changed.*

L598: Replace "4 an 5" with "4 and 5"
*Corrected.*

L608: Replace "Fig." with "Figs."
*Changed.*

L662: should "allow to assess" be "allow us to assess"?
*Changed.*

L683: Change "multi year ice" to "multi-year ice"
*Done.*

L606: Add a space before the left parenthesis "used(FES2004".
*Corrected.*

L765: Remove one "the"
*Corrected.*

L769: Make "non significant" one word "nonsignificant".
*Corrected.*

L776: Same as L769 to use "nonsignificant"
*Corrected.*

L777: Replace "cental" with "central"
*Done.*

L782: Maybe replace "the single moorings" with "at a single mooring"
*Corrected.*

L787: Maybe remove the second "which"?
*Done.*

L828: "currents there are weaker" sounds better than "currents are there weaker"
*Changed.*

L830: Replace "sArctic" with "Arctic"
*Done.*

L860: "2011-2018" or "2011-2020"?
***Thank you for pointing that out, I missed to correct it to "2011-2020". I changed it now.***

Appendix:

L908: Replace "larger then" with "larger than"
***Corrected.***

L925: Add a dot after "Fig" in " Fig C1" to make it "Fig. C1"
***Done.***

Reference:

L1006-1007: The second URL (https://www.sciencedirect.com/science/article/pii/S0273117718300309#f0005) is not needed.
***Corrected.***

Tables and Figures:

Table 1 Replace "position" with "locations"
***Changed.***

Table 2 Replace "position" in " Name, location, monthly ..." with "location". Same replace "positions" in "Variable positions indicate ..." with "locations".
***Changed.***

Figure 1: Probably need to rephrase "the bottom pressure data are taken from the empty star" to something like: the empty star is where the bottom pressure data are taken.
***Corrected.***

Figure 2: Rearrange panels b-d by moving panel b to top right and panels c & d to bottom right and change the caption " The two upper panels ..." accordingly. Also, change "prior correcting the offset" in the caption to "prior to correcting the offset".
***Changed.***

Figure 3: It seems "(per 100 km2)" only for number of observations, not for standard deviation. Probably moving "(per 100 km2)" (also add "per month") after "number of observations" in the second sentence makes more sense.
***Corrected.***

Figure A1: It appears some panels are incorrectly referenced. Replace "... correction FES2004 (a, d) and FES2014 (b, c). In panels (e) and (f)" with "... correction FES2004 (a, d) and FES2014 (b, e). In panels (c) and (f)"
***Corrected.***

**Referee#2**
* * *
The paper describes a new satellite-based Arctic-wide gridded dataset for monthly sea surface height anomaly and geostrophic velocity. The paper is in general well written. The methodology for generating the dataset is clear and robust. The new dataset is evaluated using both independent gridded satellite data and in-situ observations in different locations. The paper and the dataset are very relevant to the Arctic research community.

I am reviewing the revised manuscript. The major criticism of the early version, which I didn't review, was the limited validation (only at two locations) of the Arctic-wide gridded dataset. In response to the criticism, the authors substantially expanded the assessment to include all eastern and western Arctic circulation regimes, the central Arctic Ocean, Arctic shelf seas and the main exchange gateways of the Arctic. In other words, all major aspects of Arctic Ocean circulations are now included in the validation. The authors also included thorough discussion on the quality of the data source and the advantages of their methodology to previous approaches. I feel the work is now acceptable for publication after some very minor revisions, mostly editorial.

The figures are generally in good quality. But some of the inserts could benefit from enlargement. For example, it is rather difficult to distinguish the different colors of the dots in the insert of Figure 4. There is plenty of space to make the insert larger.
The velocity vectors in Figure 6 are too small. The authors may want to enlarge the vectors and perhaps only plot out every other vectors.
***We improved the readability of the above figures by enlarging the inset panel and enlarging the vector fields.***

Figures:

Figure 7, Caption: " in the small area in the Baffin Bay encircled by a thick black line" There are three areas encircled by thick black line. In the small areas instead of area?
The area shaded in gray north of 82 N The region shaded
***Corrected.***

Figure 5, Label (b) is not properly positioned. In the caption, 4 weeks average 4-week average
***Changed.***

Figure 10, Caption, Red and green circles …. I notice red and black circles, no green circles.
**Changed.**

Figure 13, Missing horizontal axis labels
***Changed, the horizontal time axis is now present in all speed and bearing time series.***

Figure 14, horizontal axis c , month number month
***Changed.***

Main text:

The paper was, overall, well written. There are, however, some typos that should be fixed and sentences that could be improved before publication. Below are a few examples:

L16, basin wide seasonal basin-wide seasonal …
***Changed.***

L21-22, trends …has largely been found trends…have largely been found
***Left as is, as "has" refers to "Evidence … has been found".***

L23-26, "observational studies of ocean currents give a more fragmentary picture of changes and intensification of surface ocean currents: analysis of regional in situ data (e.g., McPhee, 2012), indirect 25 calculation from wind and ice drift observation (Ma et al., 2017) or, only recently, satellite altimetry data (Armitage et al., 2017; Morison et al., 2021)"
Consider rephrase the sentence to:
observational studies of ocean currents, including analysis of regional in situ data (e.g., McPhee, 2012), indirect calculation from wind and ice drift observation (Ma et al., 2017) or, only recently, satellite altimetry data (Armitage et al., 2017; Morison et al., 2021), give a more fragmentary picture of changes and intensification of surface ocean currents.
***Changed.***

L37, altimetry derived altimetry-derived
***Changed.***

L38, velocity which is velocity that is
***Changed.***

L44, Further data Future data?
***We rephrased the sentence to highlight that the amount of data collected over the Arctic Ocean is going to quickly increase thanks to recently launched altimetry missions.***

L45, "methodological developments for the processing of the signal coming from the ocean in ice covered regions have taken much longer to develop"
Development and to develop are redundant. Consider change to methodologies for … have taken much longer to develop Or methodological developments for … have taken much longer
***Changed.***

L50, 'few' should be 'a few'
***Changed.***

L53, differences between them 'them' refers to what here?
***It refers to gridded datasets independently derived. We rephrased the sentence to clarify this point.***

L124, Anderson et al. (2015)) (2015)
*Corrected.*

L151, compared there zonal cross-section Compared zonal cross-section?
*We rephrased the sentence.*

L155, equation 7 Eq. 7 is mentioned above. Be consistent
*Corrected.*

Table 1, Names, position, monthly data… Names, positions … or Name, position, …
*Changed to "Names, locations …".*

Table 2, Caption: first 17 rows, following 4 rows, etc. Add a column to indicate the location or add a horizontal line between two locations
*Changed the table by adding rows with the names of locations before the first mooring at each location.*

L293, The DAC is today conventionally used over The DAC is conventionally used today over
*Changed.*

L322, state of the art ocean tidal state-of-the-art ocean tidal
*Corrected.*

L421, Monthly η maps were obtained as the average of four weekly maps. What happened to the average when a week spans two months?
*The weeks were defined based on the days of the month rather than the actual calendar week. Each month was divided in four parts as follows: day#1-day#8, day#9-day#15, day#16-day#23, day#24-LASTday. We believe that this has a negligible influence on our final results.*

L461, Repeat of the sentence: In the η 0 monthly fields we generally find that there are extended regions of either positive or negative values. In the η 0 monthly fields we generally find that there are extended regions of either positive or negative values.
*Removed one of the repetitions.*